# Targeting de novo lipogenesis and the Lands cycle induces ferroptosis in KRAS-mutant lung cancer

Caterina Bartolacci[1,14], Cristina Andreani [1,14], Gonçalo Vale [2], Stefano Berto [3], Margherita Melegari[1], Anna Colleen Crouch[4], Dodge L. Baluya[5], George Kemble[6], Kurt Hodges[7], Jacqueline Starrett[8], Katerina Politi[8], Sandra L. Starnes[9], Daniele Lorenzini[10], Maria Gabriela Raso [11], Luisa M. Solis Soto [11], Carmen Behrens[12], Humam Kadara[11], Boning Gao [13], Ignacio I. Wistuba[11], John D. Minna [13], Jeffrey G. McDonald [2] & Pier Paolo Scaglioni [1✉]

Mutant *KRAS* (KM), the most common oncogene in lung cancer (LC), regulates fatty acid (FA) metabolism. However, the role of FA in LC tumorigenesis is still not sufficiently characterized. Here, we show that KMLC has a specific lipid profile, with high triacylglycerides and phosphatidylcholines (PC). We demonstrate that FASN, the rate-limiting enzyme in FA synthesis, while being dispensable in EGFR-mutant or wild-type KRAS LC, is required for the viability of KMLC cells. Integrating lipidomic, transcriptomic and functional analyses, we demonstrate that FASN provides saturated and monounsaturated FA to the Lands cycle, the process remodeling oxidized phospholipids, such as PC. Accordingly, blocking either FASN or the Lands cycle in KMLC, promotes ferroptosis, a reactive oxygen species (ROS)- and iron-dependent cell death, characterized by the intracellular accumulation of oxidation-prone PC. Our work indicates that KM dictates a dependency on newly synthesized FA to escape ferroptosis, establishing a targetable vulnerability in KMLC.

[1] Department of Internal Medicine, University of Cincinnati College of Medicine, Cincinnati, OH 45219, USA. [2] Center for Human Nutrition, The University of Texas Southwestern Medical Center, Dallas, TX 75390, USA. [3] Department of Neuroscience, The University of Texas Southwestern Medical Center, Dallas, TX 75390, USA. [4] Department of Interventional Radiology, The University of Texas MD Anderson Cancer Center, Houston, TX, USA. [5] Tissue Imaging and Proteomics Laboratory, Washington State University, Pullman, WA 99164, USA. [6] Sagimet Biosciences, San Mateo, CA 94402, USA. [7] Department of Pathology, University of Cincinnati College of Medicine, Cincinnati, OH 45219, USA. [8] Yale Cancer Center, Yale School of Medicine, New Haven, CT, USA. [9] Department of Surgery, Division of Thoracic Surgery, University of Cincinnati College of Medicine, Cincinnati, OH 45219, USA. [10] Department of Pathology, Fondazione IRCCS Istituto Nazionale dei Tumori di Milano, via Venezian 1, 20133 Milan, Italy. [11] Department of Translational Molecular Pathology, The University of Texas MD Anderson Cancer Center, Houston, TX, USA. [12] Department of Thoracic H&N Medical Oncology, The University of Texas MD Anderson Cancer Center, Houston, TX, USA. [13] Hamon Center for Therapeutic Oncology Research, The University of Texas Southwestern Medical Center, Dallas, TX 75390, USA. [14]These authors contributed equally: Caterina Bartolacci, Cristina Andreani. ✉email: Scaglipr@ucmail.uc.edu

Mutant *KRAS* (KM) lung cancer (LC) is associated with poor prognosis and resistance to therapy. *KM* expression is not only sufficient to initiate LC but also essential for the viability of KMLC[1,2]. Notwithstanding the notable exception of the *KRAS^{G12C}* mutant[3–6], KM remains undruggable. Furthermore, targeting the KM signaling network has proved to be either ineffective or toxic[7]. Finally, cancer immunotherapy benefits only a minority of KMLC patients[8,9]. Therefore, there is an urgent need for novel therapeutic strategies for KMLC.

KM reprograms cellular metabolism, promoting aerobic glycolysis and lipid synthesis/uptake[10–12]. Notably, KM regulates the expression of fatty acid synthase (FASN), the rate-limiting enzyme of de novo lipogenesis[13], which is often deregulated in cancer[14,15,14–18].

We also reported that acyl-CoA synthetase long-chain family member 3 (ACSL3), which synthesizes fatty acyl-CoA esters downstream FASN, is required for the viability of KMLC both in vitro and in vivo[12]. Also, recent studies indicated that lipid metabolism mediates adaptations to oncogenic *KRAS^{G12C}* inhibition[19]. However, the mechanisms governing the interaction of KM with FA metabolism and their functional consequences have not been characterized in sufficient detail to inform cancer therapy.

FA are required for the synthesis of complex lipids, such as phospholipids (PL) or triacylglycerides (TAG), which are used to build cellular membranes, to produce ATP through ß-oxidation[16], or are stored in lipid droplets[17,20]. On the other hand, in the presence of reactive oxygen species (ROS), polyunsaturated FA (PUFA), long-chain FA with multiple double bonds, undergo lipid peroxidation causing ferroptosis, a form of nonapoptotic ROS-dependent programmed cell death[21]. Ferroptosis plays an important role in the regulation of cell survival in several physiologic as well as pathologic processes, including cancer[22].

Three hallmark features define cancer cell sensitivity to ferroptosis: the presence of oxidizable PL containing PUFA (PUFA-PLs), redox-active iron and inefficient lipid peroxide repair (e.g., glutathione peroxidase 4, GPX4)[23–25]. Consequently, the genes and metabolic pathways controlling lipid peroxide repair, antioxidant response, or PUFA metabolism mediate the sensitivity to ferroptosis.

The Lands cycle is the main process through which PL are remodeled to modify their FA composition[26]. This process is conserved in the plant and animal kingdoms allowing the generation of new PL, effectively bypassing de novo synthesis of the entire PL molecule. Importantly, lung tissue synthesizes dipalmitoyl-PC (the major component of pulmonary surfactant) through this pathway, by de-acylating PC at the $sn_2$ position and substituting a palmitic acid (C16:0) moiety at this site. Moreover, the Lands cycle constitutes the major route for incorporation and release of free arachidonic acid (AA, C20:4) and other PUFA into cellular PL, a process that is dependent on intracellular phospholipase A2 (PLA2). Hence, the proper regulation of the Lands cycle is important to control the accumulation of potentially toxic PL and FA, in order to maintain the integrity of cellular membranes[27,28].

Here, we demonstrated that KMLC has high levels of PL decorated with saturated FA (SFA) and monounsaturated FA (MUFA). Moreover, FASN inhibition drastically reduces SFA/MUFA availability, obligating KMLC cells to incorporate the highly reactive PUFA into PL, thus decreasing the threshold for ferroptosis. Notably, silencing the key regulators of the Lands cycle phospholipase A2 group IVC (*PLA2G4C*) and lysophosphatidylcholine acyltransferase 3 (*LPCAT3*), induces ferroptosis in KMLC cells. We demonstrated that these processes are KM-dependent and confirmed our conclusions with metabolic flux analysis. This study provides the rationale for targeting FA synthesis and the Lands cycle as an effective therapeutic strategy for KMLC.

## Results

**Mutant KRAS induces a specific lipid profile in lung cancer.** *CCSP-rtTA/Tet-O-Kras^{G12D}* (hereafter *TetO-Kras^{G12D}*) and *CCSP-rtTA/Tet-O-EGFR^{L858R}* (hereafter *TetO-EGFR^{L858R}*) mice, when fed doxy, invariably develop lung tumors, which recapitulate tumorigenesis and histological features of human LC[1,29]. We performed mass spectrometry (MS) analysis on micro-dissected lung tumors and unaffected parenchyma (Fig. 1a). We found that the two tumor types have different lipidomic signatures. *TetO-Kras^{G12D}* LC has a significant increase in PC and TAG, as well as of sphingomyelins (SM) and phosphatidylethanolamine (PE), and a decrease in lysophosphatylcholines (LysoPC), while *TetO-EGFR^{L858R}* preferentially has high phosphatidylinositol (PI) and low TAG, (Fig. 1b and Supplementary Fig. 1a). Nevertheless, both *TetO-Kras^{G12D}* and *TetO-EGFR^{L858R}* tumors have higher cholesteryl-esters (CE) and lower diacylglycerides (DAG) than healthy lung (Fig. 1b and Supplementary Fig. 1a). In particular, *TetO-Kras^{G12D}* tumors are enriched in PC species with SFA and MUFA acyl chains, but have less PUFA-containing PC (and PE) than healthy lung (Fig. 1c). To determine the spatial distribution of the major lipid species identified by MS, we used Matrix Assisted Laser Desorption/Ionization (MALDI) imaging (Supplementary Data 1 and 2). We confirmed that TAG, SM and PC are mainly localized in *TetO-Kras^{G12D}* tumors rather than in surrounding stroma (Fig. 1d). In particular, we observed that SFA- and MUFA-PC are preferentially localized within the tumors, while PUFA-PC accumulate in the surrounding lung parenchyma (Fig. 1d). Notably, we observed the same lipidomic pattern in KMLC patient-derived xenografts (PDXs) and primary patient specimens (Fig. 1e–g; Supplementary Fig. 1b; Supplementary Data 3 and 4).

These data suggest that KM increases the intratumor availability of SFA and MUFA for the synthesis of TAG and PC (Supplementary Fig. 1c).

**Mutant KRAS induces a dependency on de novo lipogenesis.** In a previous study using the *TetO-Kras^{G12D}* model, we found that *Kras^{G12D}* activation/extinction positively correlate with the expression/repression of several genes involved in lipid synthesis and metabolism[12]. As previously reported by others[13], we found that KM correlates with FASN overexpression in *TetO-Kras^{G12D}* mice (Supplementary Fig. 2a), in human KMLC cell lines (Supplementary Fig. 2b) and in two independent tumor microarrays (Supplementary Data 5 and Supplementary Fig. 2c–f), while there is no significant correlation with EGFR-MUT (Supplementary Fig. 2g, h).

To test whether the lipidome of KMLC depends on de novo lipogenesis, we used TVB-3664 (Sagimet Biosciences), a non-toxic and specific FASN inhibitor (FASNi)[30,31], and *FASN* silencing on a panel of human and mouse LC-derived cell lines[32–34] (Fig. 2a and Supplementary Fig. 3).

KMLC cell lines were significantly more sensitive to FASNi than KRAS-WT cells, and exhibit inhibitory concentration 50 ($IC_{50}$) values in the nanomolar range (Fig. 2a and Supplementary Fig. 3c).

Both FASNi and *FASN* silencing caused a G2/M cell cycle arrest in KMLC, without affecting KRAS-WT and EGFR-MUT LC cells (Fig. 2b and Supplementary Fig. 3d, f). Notably, exogenous palmitate rescues the detrimental effects of FASNi,

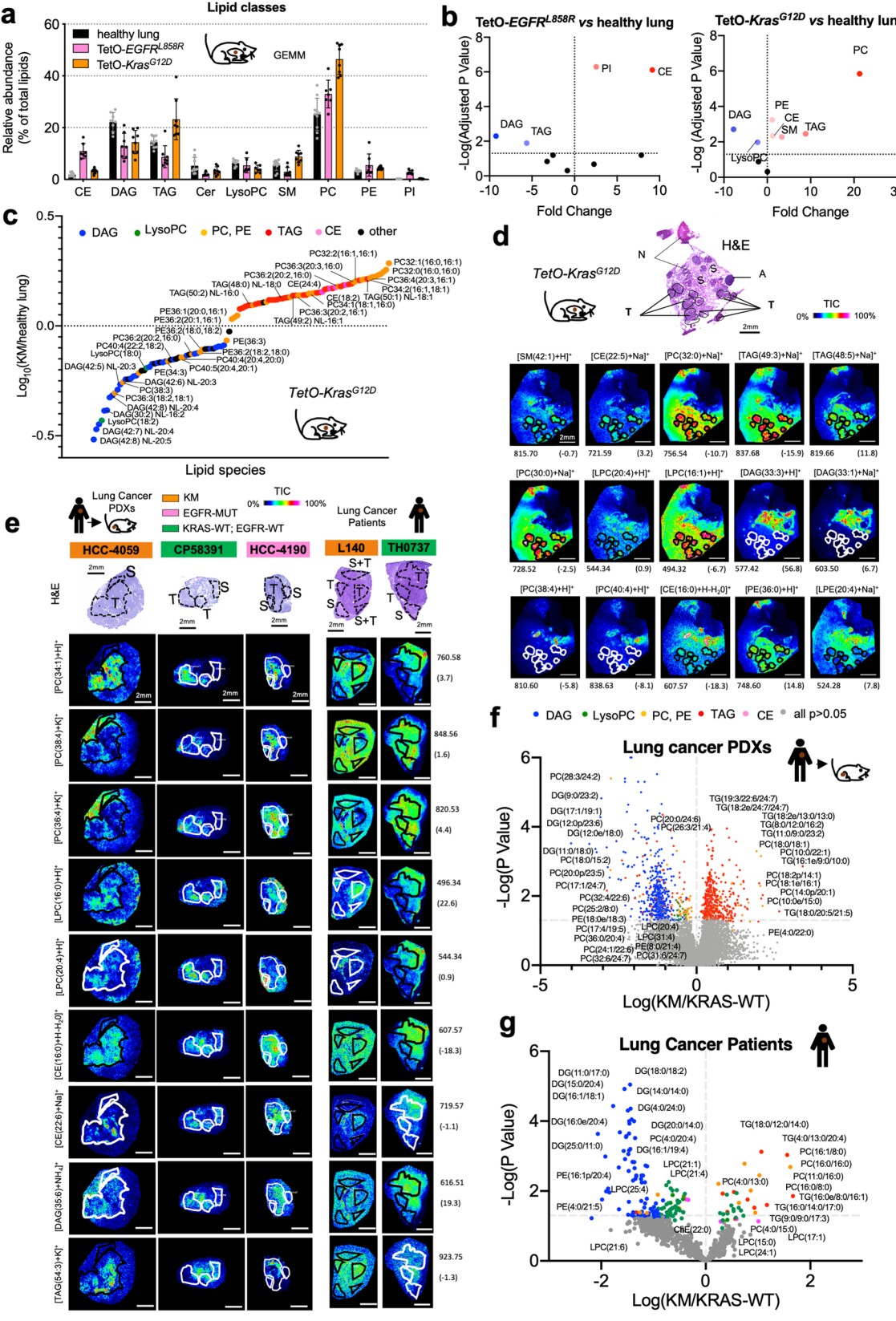

while an inactive FASNi isomer does not affect the viability of KMLC cells (Fig. 2a, b).

To test whether *KM* expression is sufficient to establish a dependency on FASN, we ectopically expressed *KM* in H522 and H661 KRAS-WT LC cells. KM induces upregulation of FASN and

its downstream enzyme, stearoyl-CoA desaturase 1 (SCD1) (Fig. 2c), accumulation of lipid droplets (Fig. 2d and Supplementary Fig. 4a) and palmitate (Fig. 2e), sensitivity to FASNi and *FASN* silencing (Fig. 2f and Supplementary Fig. 4b, c). On the contrary, *KRAS* silencing induces a significant decrease of FASN

**Fig. 1 KMLC has a unique lipidome. a**, **b** MS/MS Lipidomic analysis of murine TetO-$Kras^{G12D}$ tumors ($n = 4$), TetO-$EGFR^{L858R}$ ($n = 3$) and unaffected healthy lung ($n = 3$). Each dot indicates a lung/tumor section. Data are expressed as mean ± SD. PC, phosphatidylcholines; TAG, triglycerides; PE, phosphatidylethanolamines; CE, cholesteryl-esters; PI, phosphatidylinositols; DAG, diacylglycerides; SM, sphingomyelins; Cer, ceramides; LysoPC, lysophosphatidylcholines. Volcano plots in **b** show the lipid classes that are differentially represented for each comparison. The adjusted p-value and difference were calculated using multiple two-tailed t-tests with alpha = 0.05 followed by Benjamini, Krieger and Yekutieli FDR. **c** MS/MS and **d** MALDI imaging analyses showing lipid differentially represented in TetO-$Kras^{G12D}$ tumors as compared to unaffected heathy lung. **e** Representative pictures of MALDI imaging analysis of lung cancer patient-derived xenografts (PDXs) and primary human lung cancer specimens of the indicated genotype. In **d** and **e** rainbow scale represents % ion intensity normalized against the total ion count (TIC). Corresponding H&E and histological annotation are shown. T, tumor; S, stroma; S + T, stroma and tumor mix; N, necrotic area; A, artifact. In (d) and (e) observed m/z and mass error (ppm) values are indicated for each lipid species. Refer to Supplementary Data 1–4 for the complete tentative MALDI lipid annotation and relative quantification. **f**, **g** HPLC-MS/MS analysis of lung cancer PDXs and primary human lung cancer specimens of the indicated genotype. Volcano plots show lipid species identified by HPLC-MS/MS differentially represented in KM versus KRAS-WT samples (PDXs, KM $n = 5$ and KRAS-WT $n = 4$; lung cancer patients, $n = 3$/group). p-values were calculated using multiple two-tailed t-tests followed by Benjamini, Krieger, and Yekutieli FDR.

---

in KMLC cells (Fig. 2g), which also become resistant to FASNi (Fig. 2h). Furthermore, cotreatment with $KRAS^{G12C}$ inhibitor ARS-1620 rescues the effects of FASNi in $KRAS^{G12C}$ LC cell lines (Fig. 2i, j). Using several drug combinations and dosages on H2122 $KRAS^{G12C}$ cells, we confirmed that single treatments always outperformed the combination in viability assays (Supplementary Fig. 4d). All together these data demonstrate that KM is necessary to induce FASN dependency in LC cells.

**FASNi inhibits de novo lipogenesis independently of KRAS mutational status.** To gather mechanistic insights into the differential sensitivity to FASNi, we measured the drug uptake, stability and its activity in LC cells.

Irrespective of the *KRAS* mutational status, FASNi readily accumulates intracellularly (Supplementary Fig. 5a, b), inhibits de novo FA synthesis (Fig. 3a and Supplementary Fig. 5c), causes a concomitant accumulation of malonyl-CoA (FASN substrate) and NADPH (FASN cofactor) (Fig. 3b, c), depletion of lipid droplets (Fig. 3d and Supplementary Fig. 5d–g) and down-regulation of ß-oxidation (Fig. 3e). Moreover, FASNi causes accumulation of AMP (Fig. 3f), triggering the phosphorylation of AMPK and ACC1 in both genotypes (pACC1$^{S79}$) (Fig. 3g). This observation is consistent with the notion that AMPK limits FA synthesis through direct phosphorylation of ACC1, the enzyme that catalyzes the synthesis of malonyl-CoA, the substrate of FASN[35]. Thus, the activation of AMPK/ACC1 enhances the effect of FASNi on FA synthesis (Fig. 3h). However, since these effects occur in both KM and KRAS-WT LC cells, they do not provide an explanation for the specific dependency of KMLC cells on FASN.

**FASN inhibition induces accumulation of PUFA-phospholipids in KMLC cells.** To investigate the metabolic impact of FASN inhibition, we performed MS/MS$^{ALL}$ untargeted lipidomic analysis. We found that FASNi induces accumulation of LysoPC only in KMLC cells. On the contrary, FASNi causes a concomitant decrease of TAG and an increase of DAG irrespectively of *KRAS* status (Fig. 4a, b). No significant changes were found in the activity of either the PC-specific phospholipase C (PLC) (Supplementary Fig. 5h) or the TAG-specific lipase ATGL (Supplementary Fig. 5i), suggesting FASNi does not affect PC and TAG lipolysis[36]. These findings suggested that KMLC uses de novo lipogenesis to fuel the synthesis of TAG and PC. Notably, we found that KMLC cells accumulate PUFA in LysoPC and PC upon FASNi (Fig. 4c–f). In particular, LysoPC containing FA with two and four double bonds account for the LysoPC increase in FASNi-treated KMLC cells (Fig. 4c, d). Moreover, even though we did not detect a significant change in the total amount of PC (Fig. 4a, b), we observed the accumulation of PUFA-PC and

depletion of SFA- and MUFA-PC in FASNi-treated KMLC cells (Fig. 4e, f). This change in the composition of the acyl chains is specific for PC and LysoPC of KMLC, because other lipid classes, such as TAG, do not increase their incorporation of PUFA in response to FASNi (Fig. 4g, h). These data indicate not only that FASN provides SFA and MUFA for the synthesis of TAG and PC in KMLC, but also that its inhibition causes the incorporation of PUFA specifically in PC and LysoPC.

**The Lands cycle prevents the accumulation of PUFA-PC of KMLC.** PC are synthetized via the Kennedy pathway that conjugates phosphocholine to DAG[37]. However, the majority of PL synthesized via the Kennedy pathway are remodeled through the Lands cycle, which consists in the de-acylation of PC and re-acylation of LysoPC[26,28,38]. To test whether FASN inhibition induces uptake and incorporation of exogenous PUFA, we used arachidonic acid (AA, FA 20:4) as PUFA proxy (Fig. 5a–d). In mammals, AA is provided by exogenous dietary sources rich either in AA or its parent molecule linoleic acid (LA, FA 18:2), which is desaturated and elongated in the endoplasmic reticulum to yield AA[39]. Even though FASNi increases the total amount of AA in KMLC cells (Fig. 5a), ethyl acetate-2-$^{13}$C metabolic flux analysis revealed a decreased isotope incorporation in AA of FASNi-treated KMLC cells (Fig. 5b). These data suggest that the intracellular pool of PUFA is dependent on their uptake from the microenvironment. This conclusion is consistent with our MALDI imaging data showing that PUFA-PC are mainly localized in the lung parenchyma surrounding the KM tumors (Fig. 1d, e).

To confirm this conclusion, we used click chemistry to conjugate AA-alkyne to the Alexa-Fluor 488-azide[40–42]. We demonstrated that, even though the baseline AA uptake is lower in KMLC as compared to KRAS-WT cells, FASNi treatment increases the incorporation of AA only in KMLC cells (Fig. 5c, d). Noteworthy, ectopic expression of KM significantly increases AA uptake during FASNi treatment (H552-KM, Fig. 5c, d), indicating that this process is dependent on KM.

To investigate whether this process is influenced by the repertoire of exogenous FA, we developed an in vitro assay of single-choice and competitive uptake of PUFA/SFA in H460 KMLC cells (Supplementary Fig. 6a), using AA-alkyne conjugated to Alexa-Fluor 594 -azide (red) via Click-iT chemistry[40–42], and the C16-BODIPY (green).

We observed that FASNi increases the total FA uptake with respect to vehicle (Supplementary Fig. 6b, c). In particular, in the single-choice settings, we noticed that: (1) the uptake of either AA or C16 is dose-dependent; (2) availability of C16 induces a greater total FA uptake than the same amount of AA in both vehicle and FASNi groups (Supplementary Fig. 6b, c). These two observations

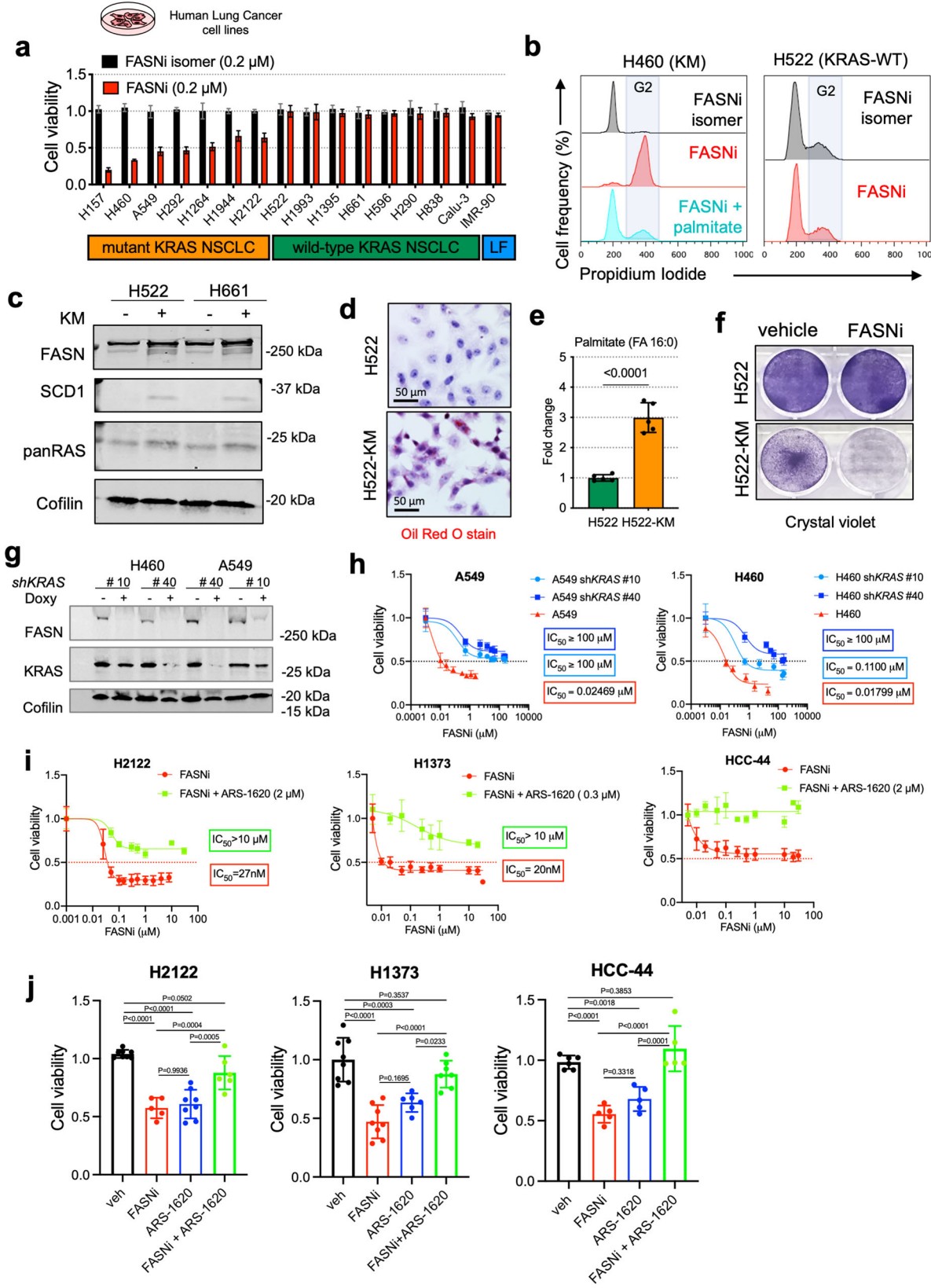

are consistent with our hypothesis that the availability of exogenous PUFA/SFA dictates their uptake and that KMLC might prefer C16 over AA in single-choice conditions. In addition, when we calculated the percentage uptake of C16 and AA in cells subjected to the various FA mixtures, we found that:

(1) in all conditions, C16 accounts for the majority of the total FA uptake; (2) FASNi modestly increases the uptake of AA when cells are incubated with AA:C16 ratio between 1:1-4:1; (3) to produce a comparable percentage uptake of AA and C16, AA:C16 ratio must be between 4:1 and 3:1; (4) when C16 is predominant

**Fig. 2 KM is required to induce dependency on FASN. a** Viability assay of human LC-derived cell lines ($n = 2$ biological independent experiments). Cell line, genotype and treatments are indicated. LF: lung fibroblasts. **b** Cell cycle analysis of H460 and H522 cells, as representative examples of KM and KRAS-WT LC cells, treated as indicated. Cell populations indicate singlets in the FL2-W/FL2-A gate. Refer to Supplementary Fig. 11 for a representative gating strategy. **c** Representative immunoblots of FASN, SCD1 and pan-RAS in H522 and H661 cells transduced as indicated ($n = 2$ independent experiments). **d** Oil red O staining, relative steady-state quantification of palmitate (FA 16:0) (**e**) and crystal violet assay (**f**) of H522 and H522-KM cells treated as indicated. **g** Immunoblot of FASN and KRAS in H460 and A549 cells transduced with doxy-inducible shRNAs targeting *KRAS*. **h** MTT viability assay of H460 and A549 cells treated with FASNi before and after induction of *KRAS* knock-down ($n = 2$ biological independent experiments). **i, j** MTT viability assays of H2122, H1373, and HCC-44 cells (*KRAS^{G12C}* mutant) treated with FASNi alone or in combination with the *KRAS^{G12C}* inhibitor ARS-1620 ($n = 2$ biological independent experiments). Data are expressed as mean ± SD. In **e** Student *t*-test with. In **j** one-way ANOVA followed by Tukey's multiple comparison test.

(AA:C16 ratio 1:3-1:4), FASNi does not increase the uptake of AA in KMLC cells (Supplementary Fig. 6d).

All in all, these results further support our conclusions that: (1) the extracellular availability of PUFA/SFA determines their uptake and the susceptibility to ferroptosis of KMLC cells; (2) KMLC prefers SFA over PUFA; (3) to produce a comparable C16 and AA uptake, PUFA must be present in large excess; (4) excess of exogenous SFA impedes AA uptake, in line with the fact that exogenous palmitate can rescue FASNi phenotype in KMLC.

At transcriptional level, FASNi treatment upregulates several genes involved in the metabolism of lipids and lipoproteins in both KM and KRAS-WT LC cells (Fig. 5e and Supplementary Data 6). However, we found that only FASNi-treated KMLC cells selectively upregulate metabolic genes such as *PLA2G4C*, *LPCAT3* and *ACSL3* (Fig. 5f). These genes are involved in the remodeling of phospholipids through the Lands cycle and in ferroptosis (Fig. 5g), a regulated form of cell death characterized by lipid peroxidation[26,43]. In particular, *PLA2G4C* is a member of the PLA2 family, which hydrolyzes PL to produce free FA and lysophospholipids (LysoPL). LPCAT3 inserts acyl groups into LysoPL, specifically forming PC and PE[38] and it is also required for cells to undergo ferroptosis[44,45]. ACSL3 converts free long-chain FA into fatty acyl-CoA esters, which undergo ß-oxidation or incorporation into PL[46]. ACSL3 is required for KMLC tumorigenesis[12], it plays an important role in AA metabolism in LC[47] and protects cells from ferroptosis[48,49]. To validate these RNA-seq data, we perform a custom siRNA screen targeting 29 genes involved in lipid metabolism. We found that 10 genes, including *PLA2G4C*, *LPCAT3*, *LPCAT1*, *ACSL3* and glutathione peroxidase 4 (*GPX4*), which counteracts ferroptosis by catalyzing the reduction of peroxided PL[21,22,24,25,50], are selectively required for the viability of KM, but not of KRAS-WT LC cells (Fig. 5h, i).

Consistently, KMLC cells are more sensitive than KRAS-WT LC cells to ML162, a specific inhibitor of GPX4 and an inducer of ferroptosis (Supplementary Fig. 7a, b), and we rescued the effect of FASNi and ML162 using the anti-ferroptotic molecule ferrostatin-1 (Fer-1) (Supplementary Fig. 7b–g). Stable knock-down of *GPX4, LPCAT3* and *PLA2G4C* phenocopies the effect of ML162 or FASNi, inducing G2/M cell cycle arrest specifically in KMLC cells (Supplementary Fig. 7h–j), a feature often associated with ferroptosis[51,52].

These results, along with the lipidomic profile of KMLC (Figs. 1 and 4), indicate that the de novo lipogenesis is necessary to repair/prevent lipid peroxidation in KMLC by feeding the Lands cycle with SFA and MUFA. To demonstrate this hypothesis, we performed stable knock-down of *LPCAT3* and *PLA2G4C* in KMLC cells. Then we subjected them to 3PLE extraction[53] coupled with either ethyl acetate-2-^{13}C metabolic flux analysis (Fig. 5j) or steady-state MS/MS^{ALL} lipidomic analysis (Fig. 5k). We determined that silencing either *LPCAT3* or *PLA2G4C* phenocopies FASN inhibition, decreasing the incorporation of de novo synthesized SFA, such as palmitate

(FA 16:0) (Fig. 5j), while increasing the AA (FA 20:4) content of PL (Fig. 5k).

Even though these data are consistent with the role of the Lands cycle in remodeling PL, they are in contrast with the existing literature describing LPCAT3 as an inducer of ferroptosis[44]. Indeed, LPCAT3 is largely reported to have preference for arachidonic acid, AA (20:4)[28]. However, most of the data about LPCAT3 enzymatic activity were obtained in intestine and liver, where it is quite abundant, or in human and mouse hepatoma cell lines[54–58].

To solve this discrepancy, we tested whether, in the context of KMLC, LPCAT3 re-acylates LysoPC with palmitoyl-CoA (16:0-CoA), using an established biochemical assay[59]. We incubated a mixture of acyl-CoAs and LysoPC with microsomal extracts (the cell fraction enriched with LPCAT3), and analyzed the resulting PC species by LC-MS/MS (Supplementary Fig. 8).

We cultured A549 cells stably transduced with an inducible shLPCAT3 either in doxy or in control media for 48 h and incubated them with either arachidonyl-CoA (20:4-CoA) or 16:0-CoA, in presence of the 1-(10Z)heptadecenoyl-2-hydroxy-lyso-phosphatidylcholine (LysoPC 17:1) (Supplementary Fig. 8a). We found that, in this context, LPCAT3 is not only able to use 16:0-CoA to re-acylate LysoPC 17:1, but also that it prefers 16:0-CoA to 20:4-CoA when they are present at the same molar concentration (Supplementary Fig. 8b). This experiment is in agreement with our lipidomic and AA uptake data, suggesting that the cellular context and the abundance of PUFA/MUFA dictate the substrate selection of LPCAT3.

In conclusion, these data demonstrate that both de novo FA synthesis and PL remodeling are necessary to prevent PUFA accumulation and ferroptosis in KMLC cells (Fig. 5l, m).

**FASN and the Lands cycle are required to deflect ferroptosis in KMLC.** Using the lipid peroxidation probe C11-BODIPY (581/591), we confirmed that FASNi causes ferroptosis specifically in KMLC cells, leaving KRAS-WT cells unaffected (Fig. 6a, b). Notably, ectopic expression of *KM* in KRAS-WT cells causes C11-BODIPY oxidation in response to FASN inhibition (Fig. 6c, d), indicating that KM is required for the induction of ferroptosis. *TP53, STK11* and KEAP1/NFE2L2 (also known as NRF2) are frequently mutated in KMLC, deregulating cellular metabolism and oxidative stress[60–64]. We did not find any correlation between their mutational status and sensitivity to FASNi in the LC cell lines we used (Supplementary Data 7). Therefore, these data indicate that the susceptibility of KMLC to ferroptosis depends on KM, rather than on *KEAP1/NFE2L2, TP53*, and *STK11* co-occurring mutations.

Next, we used the 3PLE extraction method coupled with UV absorbance to measure the FA oxidation in the neutral (e.g., TAG) and polar lipid (e.g., PL and LysoPL) fractions of KMLC cells (Supplementary Fig. 9a)[53]. Indeed, as oxidation occurs in lipids containing two or more double bonds, it causes an increase

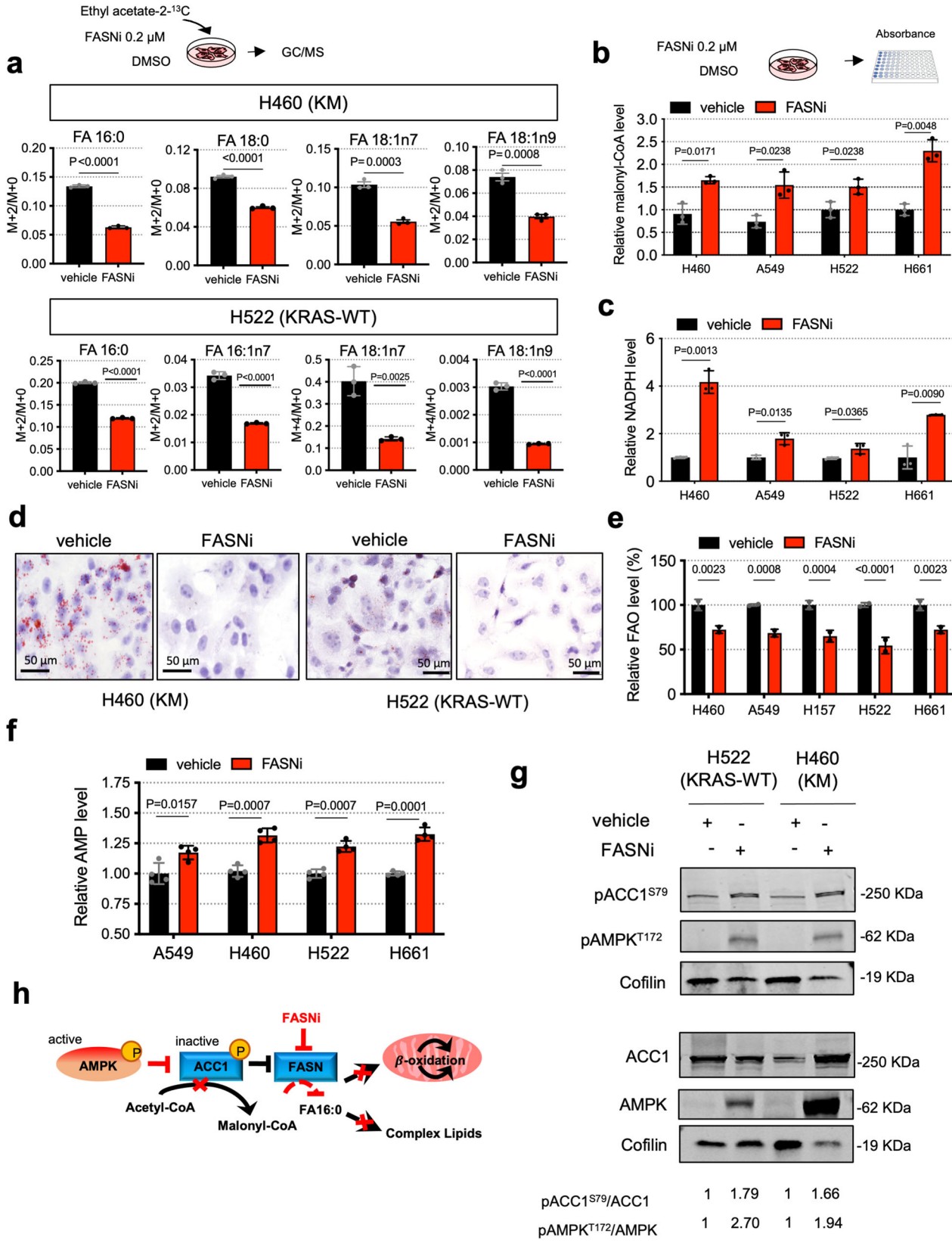

in the UV emission that can be used to quantify the rate of primary lipid oxidation[65]. We found that FASNi induces oxidation specifically of the polar lipid fraction of KMLC cells (Supplementary Fig. 9b), without perturbing neutral lipids (Supplementary Fig. 9c). This finding is consistent with the observation that in KMLC cells, FASNi leads to an enrichment of

PUFA specifically in PC and LysoPC, but not in TAG (Fig. 4). Accordingly, PC supplementation quenches ROS production and propagation in KMLC cell treated with FASNi (Supplementary Fig. 9d and Supplementary Videos).

To further demonstrate that FASN and the Lands cycle are necessary to prevent ferroptosis, we treated KMLC cells with

**Fig. 3 FASNi inhibits fatty acid synthesis and ß-oxidation in both KM and KRAS-WT cells. a** GC/MS quantification of newly synthesized FA in H460 and H522 cells after overnight ethyl acetate-2-$^{13}$C labeling, treated as indicated. Either $M + 2/M + 0$ or $M + 4/M + 0$ ratio is reported. Palmitate, FA 16:0; palmitoleate, FA 16:1n7; vaccenate, FA 18:1n7; oleate, FA 18:1n9 ($n = 3$ independent experiments). **b, c** Relative quantification of malonyl-CoA and NAPH of vehicle- and FASNi-treated LC cells ($n = 3$ independent experiments). **d** Oil red O staining for lipid droplets in H460 and H522 cells. **e, f** Relative quantification of FA ß-oxidation (FAO) and AMP in the indicated cells treated with vehicle or FASNi ($n = 3$ and $n = 4$ biologically independent samples). **g** Immunoblot of phospho-Ser79-ACC1 (pACC1$^{S79}$), ACC1, FASN, phosphor-Thr172-AMPK (pAMPK$^{T172}$) and AMPK. **h** Schematic of the AMPK/ACC1/ FASN axis. FASNi inhibits the synthesis of palmitate (FA 16:0) thereby blocking the synthesis of complex lipids and ß-oxidation (FAO). These events trigger the activation of AMPK, which in turn phosphorylates and deactivates ACC1. ACC1 phosphorylation blocks the synthesis of malonyl-CoA (the substrate for FA 16:0 synthesis) potentiating the inhibitory effects of FASNi. Bars express mean ± SD. Statistical analyses were done using two-tailed unpaired Student's *t*-test.

---

FASNi and/or molecules targeting the cysteine/GSH/ GPX4 system, one of the mainstays restricting ferroptosis (Fig. 6e). Noteworthy, while PC, palmitate, ferrostatin-1 (Fer-1), and N-acetyl-cysteine (NAC) rescue the C11-BODIPY oxidation induced by FASNi, the PUFA linoleic acid (LA) does not (Fig. 6f, g). Furthermore, silencing of *LPCAT3* or *PLA2G4C* induces significant C11-BODIPY oxidation in KMLC cells (Fig. 6h, i).

**Pharmacologic inhibition of FASN suppresses KMLC in vivo.** We explored the preclinical significance of our findings using transgenic *TetO-Kras*$^{G12D}$ mice (Fig. 7a, b) and xenografts of A549 and H460 KMLC cells (Fig. 7c, d). In all the preclinical mouse models, FASNi causes a potent anti-tumor effect without overt systemic toxicities (Fig. 7a–d and Supplementary Fig. 10a–c), and it depletes intratumor and serum palmitate (Supplementary Fig. 10d). Notably, FASNi induces lipid oxidation in both autochthonous and xenograft KMLC, as demonstrated by C11-BODIPY staining (Fig. 7e–g). In addition, lipidomic analysis (Fig. 7h) and MALDI imaging (Supplementary Fig. 10f and Supplementary Data 8) confirmed that inhibiting de novo lipogenesis causes accumulation of PUFA specifically in PC and LysoPC of KMLC tumors. On the other hand, TAG are downregulated and DAG are upregulated, independently of the saturation of their acyl chains (Fig. 7h and Supplementary Fig. 10e). Notably, the ferroptosis inhibitor Liproxstatin-1 (Lip-1) was able to rescue the anti-tumor effect of FASNi in vivo (Fig. 7i, j and Supplementary Fig. 10f).

These data phenocopy the findings obtained in vitro (Figs. 4 and 6f, g; Supplementary Fig. 6) providing further evidence that FASNi induces ferroptosis in KMLC.

## Discussion

KM is the most commonly mutated oncogene in cancer[66]. There is a considerable interest in determining the mechanisms underlying KM-driven tumorigenesis and KM-dependent tumor survival. This knowledge is a prerequisite to develop novel therapies as well as to optimize existing drugs that target either KM or its downstream signaling pathways. It is well known that KM contributes to the regulation of cancer metabolism; however, no metabolic network has been established as a bona fide therapeutic target[10]. Furthermore, it is also emerging that metabolic adaptations mediate resistance to KRAS inhibitors[19].

Here, we report that KMLC depends on de novo FA synthesis and PL remodeling to cope with oxidative stress and to escape ferroptosis. In KMLC, KM upregulates FASN, whose inhibition causes lipid peroxidation and ferroptosis. These effects are specific, since metabolites immediately downstream FASN (i.e., palmitate or PC) rescue this phenotype. We determined that KM, but not EGFR-MUT, is necessary and sufficient to establish the dependency on FASN. On the contrary, we found no correlation with *TP53*, *STK11/LKB1* or *NFE2L2/NRF*2, which are frequently co-mutated with KM and known to influence metabolism and oxidative stress[61,62].

It is a long-standing observation that *FASN* is overexpressed in several cancer types and that KM upregulates its expression[13]. It has been proposed that FA meet the requirements of highly proliferative cells providing building blocks for membranes, or sustaining ATP production through ß-oxidation[16]. However, the inhibition of these processes does not affect KRAS-WT cells and does not explain the exquisite and specific dependency of KMLC on de novo lipogenesis.

Our comprehensive metabolic and functional analysis shows, for the first time, that KM upregulates FASN to increase the synthesis of SFA and MUFA to prevent and repair lipid peroxidation in LC. Our conclusion is supported by metabolic flux analysis showing how FASNi induces the incorporation of PUFA into PL, in particular PC. When de novo lipogenesis is impaired, KMLC increases the uptake of exogenous PUFA. However, even though we demonstrated that cellular lipolysis does not account for this process, we cannot exclude that other mechanisms like lipophagy and macropinocytosis[67,68] might contribute to the enrichment of PUFA observed after FASN inhibition.

Because of their multiple double bonds, PUFA are the major substrate of lipid peroxidation[43,63,69]. Therefore, cells use the Lands cycle to limit the amount of PUFA incorporated in PL, reducing lipid peroxidation and the likelihood to trigger ferroptosis[47,70,71]. Accordingly, we found that silencing key regulators of the Lands cycle induces PUFA accumulation in the side chains of PL, leading KMLC to ferroptosis.

KM expression and/or activation of the RAS/MAPK pathway have been reported to sensitize cells to ferroptosis inducers[72–74]. However, later studies challenged these conclusions[24]. We reason that this incongruency is explained by the fact that activation of KRAS pathway increases the susceptibility to ferroptosis, but that cell context-specific factors contribute to its execution.

Our data indicate that KMLC has a lipidomic profile that is reminiscent of alveolar type II cells (AT2), which are considered the cells of origin of LC[75]. AT2 cells are a major source for surfactant lipids secreted into the alveoli. Surfactant lipids comprise about 90% of pulmonary surfactant, a lipoprotein complex, which decreases surface tension in the post-natal lung and prevents alveolar collapse. PC are the most abundant surfactants lipids[76]. Notably, SFA and MUFA-PC like dipalmitoylphosphatidylcholine PC 32:0 (DPPC, 16:0/16:0), PC 32:1 (16:0/16:1) and PC 30:0 (16:0/14:0) are the most abundant members of surfactant PC both in AT2 cells and in KMLC[77–79]. Thus, it seems reasonable to conclude that KM hijacks lung-specific mechanisms evolved to mitigate oxidative stress and ferroptosis under the high oxygen tension conditions found in the pulmonary alveolus[80]. Indeed, transitioning from the fetal hypoxic environment to air breathing at birth requires the activation of adaptive responses, as demonstrated by the observation that, mouse AT2 not only have high expression of *Kras* soon after birth, but they also exhibit high expression of genes involved in

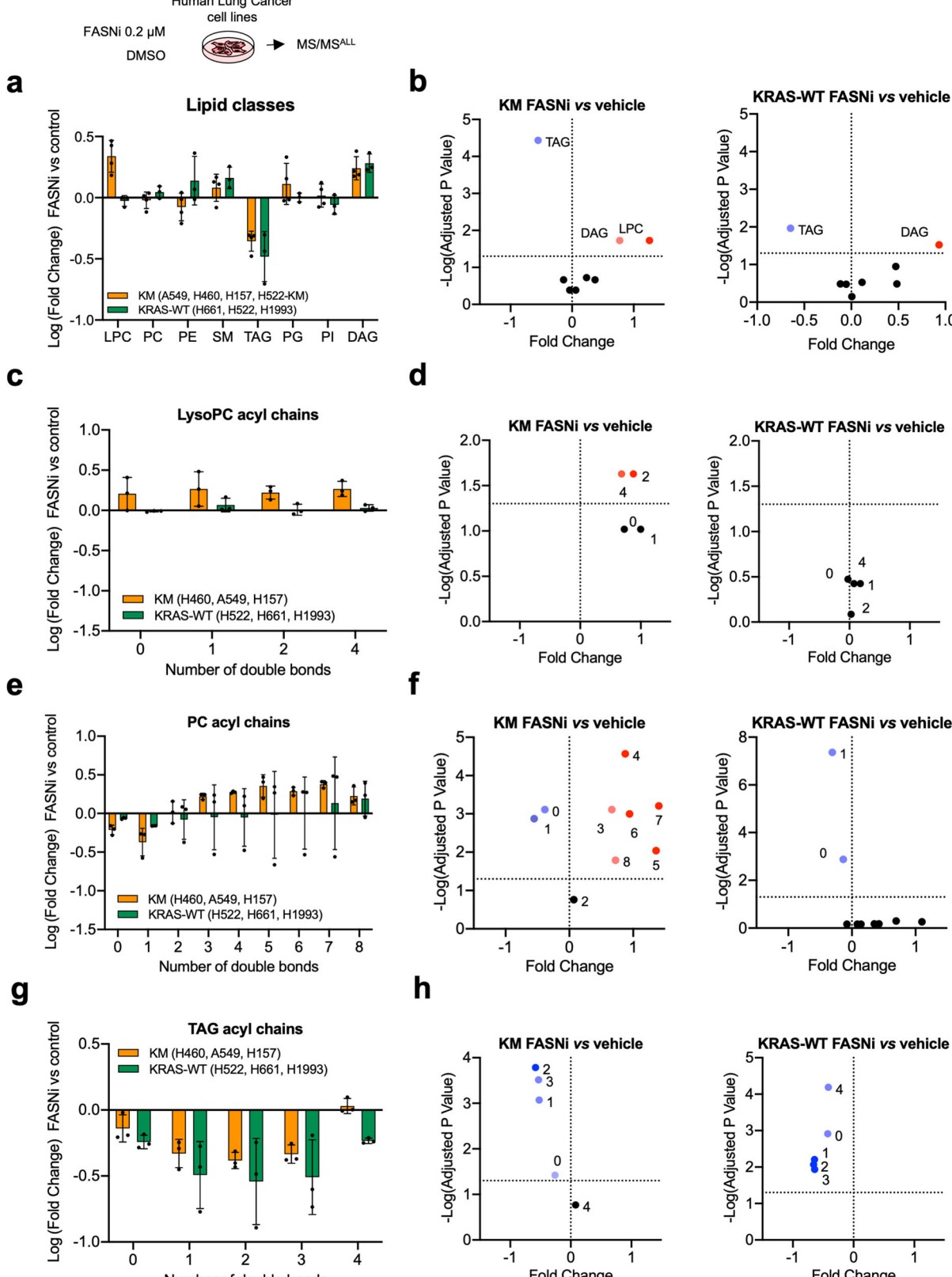

**Fig. 4 FASNi induces accumulation of PUFA-PC and PUFA-LysoPC in KMLC. a** Variation of the lipid classes identified by MS/MS in KM and KRAS-WT LC cells. Bars represent Log (fold-change) of FASNi treatment over vehicle control (*n* = 3 biologically independent cell lines/group). **b** Volcano plots of multiple two-tailed *t*-tests followed by Benjamini, Krieger, and Yekutieli FDR representing the significant changes in lipid classes for the indicated comparisons (cutoff adj *p* < 0.05). **c–h** Relative double bond quantification in the indicated lipid classes (*n* = 3 biologically independent cell lines/group) and volcano plots showing the correspondent multiple two-tailed *t*-tests followed by Benjamini, Krieger, and Yekutieli FDR (cutoff adj *p* < 0.05). Bars express mean ± SD.

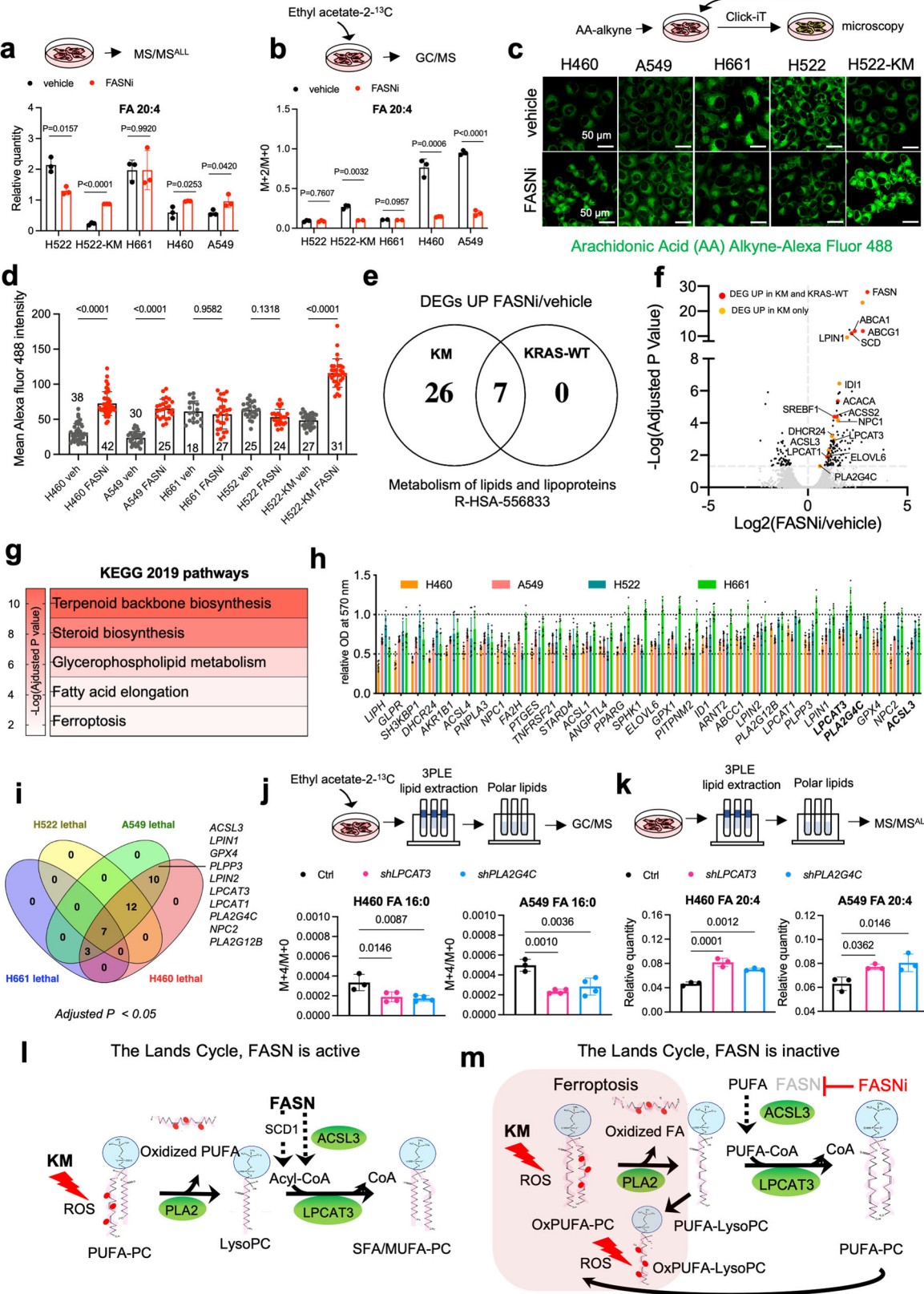

the unfolded protein response (UPR), antioxidant response and lipid synthesis[80]. Hence, FASNi interferes with the PC-dependent antioxidant function, decreasing the threshold to ferroptosis in the context of high ROS in KMLC.

A well-developed literature reported that oncogenic RAS leads to an increase in ROS production via multiple mechanisms[81–85].

In this scenario, we reason that the PUFA-rich lung environment, surrounding KMLC in vivo, may represent a "ticking bomb" for KM tumors. This is in agreement with our observation that exogenous palmitate bypasses FASNi-induced cell death, with previous studies showing that exogenous MUFA confer resistance to ferroptosis[49] and with our competitive uptake assay,

**Fig. 5 FASN and the Lands cycle limit PUFA content of phospholipids of KMLC.** Quantification of total (**a**) and newly synthetized (**b**) arachidonic acid (FA 20:4) in the PL fraction of the indicated cell lines ($n = 3$ biologically independent samples). Tracer incorporation was measured after 7-h incubation with ethyl acetate-1,2 $^{13}C_2$. **c, d** Incorporation of arachidonic acid (AA) alkyne in the indicated cell lines treated as indicated, and its quantification. $n = 20$–42 cells over 2 biologically independent samples. **e** Venn Diagram of the "metabolism of lipids and lipoproteins" (R-HSA-556833) genes upregulated in KM (H460, A549) and KRAS-WT (H661, H522) LC cells treated with FASNi. **f, g** Gene expression volcano plot and top KEGG pathways specifically upregulated in KMLC cells upon FASNi treatment. **h, i** Cell viability after siRNA-mediated knockdown of the indicated genes ($n = 2$ biologically independent experiments). Venn diagram summarizes lethal genes specific for KMLC cells (H460 and A549, Dunnett's multiple comparison test with cutoff adj $p < 0.05$). **j, k** Quantification of newly synthetized- palmitate (FA 16:0) and total arachidonic acid (FA 20:4) in the PL fraction of indicated cell lines after shRNA-mediated knockdown of lysophosphatidylcholine acyltransferase 3 (*LPCAT3*) and phospholipase A2 group IV C (*PLA2G4C*) ($n = 4$ and $n = 3$ biological independent samples). **l, m** Working model explaining the role of FASN in the regulation of the Lands cycle in KMLC. FASN is active: KM induces ROS that oxidize the PUFA acyl chain on PC. PLA2 removes the oxidized fatty acid (FA) on PC synthetizing a LysoPC. FASN and SCD1 produce saturated FA (SFA) and MUFA, respectively. SFA/MUFA are transferred to CoA by ACSL3. These acyl-CoAs are used by LPCAT3 to re-acylate the LysoPC forming again PC. Inhibition of FASN (**m**) causes the depletion of SFA/MUFA and uptake of exogenous PUFA for the re-acylation of LysoPC. This process increases the amount of PUFA-PC and PUFA-LysoPC, which are oxidized under oxidative stress (oxPUFA-PC and oxPUFA-LysoPC). Accumulation of these lipid species leads to cell death via ferroptosis. Bars represent mean ± SD. In **a, b, d, j, k** Statistical analyses were done using two-tailed unpaired Student's *t*-test. In **f** *p*-values were generated using one-way ANOVA and adjusted for multiple comparisons using a Benjamini–Hochberg correction (FDR). In **h** statistical analysis was performed using one-way ANOVA followed by Dunnett's multiple comparisons test.

showing that extracellular availability of SFA/PUFA dictates their uptake in KMLC.

In addition, even though de novo lipogenesis feeds the Lands cycle with newly synthesized even-chain FA, our study indicates that the remodeling of PL might target pre-existent species, such as PC, containing odd-chain FA (Figs. 1f, g and 7h), which derive either from the diet or alternative biosynthetic/metabolic pathways (i.e., α-oxidation of exogenous branched-chain FA)[86–90].

Future studies are needed to determine whether FA deriving from dietary sources or the tumor microenvironment could directly influence KMLC tumorigenesis and response to therapy.

Our data predict not only that FASNi will be effective in KMLC therapy, but also that other inducers of ferroptosis may exert selective anti-tumor effects in KMLC. It is noteworthy that TVB-2640, a FASNi derivative with improved pharmacokinetic properties, is showing promising results in a phase II trial in KMLC (NCT03808558) and that several other FDA-approved drugs (e.g., sorafenib, sulfasalazine, artesunate, lanperisone) are known to induce ferroptosis in certain cancers[91–97]. Therefore, it is likely that cancers other than KMLC rely on FASN and the Lands cycle to overcome ferroptosis.

In this context, our study will not only inform the current and future clinical trials with FASNi, but it will also guide future strategies to harness ferroptosis and lipid metabolism as cancer therapy.

## Methods

**Human LC samples and human LC PDXs.** Patients have been consented under protocols "2013-4722: Prospective Lung Cancer Outcomes Database" and "2013-4520: Investigator Initiated LUN 2007" approved by the Institutional Review Board at University of Cincinnati. Human LC samples ($n = 6$, Supplementary Table 1) were obtained from University of Cincinnati Biorepository and frozen human LC PDXs ($n = 11$, Supplementary Table 2) were obtained from Hamon Cancer Center at UT Southwestern Medical Center under usage agreement (SR ID:TB0109). All human research participants provided informed consent to both be included in the study and also for publishing their information.

**Cell lines.** LC cell lines were obtained from the cell-line repository of the Hamon Center for Therapeutic Oncology Research (UT Southwestern Medical Center (1,2), IMR-90 human lung fibroblasts were from ATCC (CC-186™), Dr. Monte M. Winslow kindly provided mouse KMLC cell lines 238N1, 802T4, 368T1, 593T5 derived from LSL-*Kras*$^{G12D}$ lung tumors[33]. Cells were maintained as previously described[12,98]. Cell lines were authenticated by DNA fingerprints for cell-line individualization using Promega Stem Elite ID system, a short tandem repeat (STR)-based assay, at UT Southwestern Medical Center Genomics Core.

For drug treatments, growth medium was replaced by HyClone RPMI medium (GE, SH30027) supplemented with 5% heat-inactivated dialyzed FBS (SIGMA-ALDRICH, F0394).

**Drugs and enzymatic inhibitors.** TVB-3664 (FASNi) or the inactive isomer (TVB-2632) were from Sagimet Biosciences (San Mateo, CA). ML162 (#20455) or Ferrostatin-1 (#17729) were from Cayman. ARS-1620 (HY-U00418) and Liproxstatin-1 (HY-12726) were from MedChemExpress. 16:0–18:1 PC (hereafter PC #850457P), Sodium palmitate (#P9767) and N-Acetyl-L-cysteine (#A7250) were from Millipore-Sigma. CellROX™ Green Reagent (#C10444) was from Thermo Fischer Scientific.

**Mouse studies.** All animal studies were approved by the Institutional Animal Care and Use Committee (IACUC) at University of Cincinnati (protocols 18-04-16-01 and 20-11-30-02). Both male and female mice were included in the analysis. Power calculation was conducted using ClinCalc.com using mean tumor burden from previous mouse experiments[12,99]. The investigators were blinded during post-study data analyses. Mice were humanely euthanized when they develop signs of severe distress defined as following criteria: (1) more than 20% decrease in body weight compared to age/sex matched control animals; (2) pelvic bones, ribs or spine visible (body condition score of 2 or less); (3) hunched posture, unkempt coat or labored breathing; (4) not moving or staying away from cohorts, not able to obtain food/water. For xenograft models, mice will be euthanized is the tumor reaches 3 cm³ cumulative volume (multiple tumors) or 2 cm-diameter (single tumor) or whenever the animal develops signs of distress. Unless otherwise specified mice were fed Purina LabDiet (PicoLab® Rodent Diet 20) and maintained under constant humidity, temperature and 12 h light/dark cycles.

**Transgenic mice.** *CCSP-rtTA/Tet-op-Kras*$^{G12D}$ (FVB/SV129 mixed background) mice were described previously[1]. We obtained lung-specific *Kras*$^{G12D}$ expression (i.e., KM) by feeding mice with doxycycline (doxy)-implemented food pellets (ENVIGO, TD 2018, 625 Dox, G) for 2 months. At this point, mice were randomly assigned to vehicle or FASNi. Lungs of *CCSP-rtTA/Tet-O-EGFR*$^{L858R}$ mice were kindly provided by Dr. Katerina Politi.

**Xenograft mouse models.** We performed xenograft studies in 6-week-old NOD/SCID mice (Jackson Laboratory), as previously described[12,98]. Briefly, $1 \times 10^6$ cells (A549 or H460) were injected in the right flank of NOD-SCID mice. Tumors were measured twice a week using a digital caliper. Mice were euthanized when xenografts reached 2 cm³ or at the end of the study, whichever came first.

**Detection of FASN in human KMLC.** We obtained data regarding FASN expression in human KM, EGFR-MUT, KRAS-WT and EGFR-WT LC cell lines from the Cancer Cell Line Encyclopedia (CCLE) database[100], www.broadinstitute.org/ccle).

YTMA-310 tumor tissue microarray (TMA), Yale Pathology Tissue Service (YPTS), consists of 30 EGFR-mutant, 43 KRAS-mutant, 66 EGFR and KRAS wild-type (EGFR/KRAS WT) NSCLC patient samples, and 6 cell-line cores (HCC193, A431, HT29, SW480, MCF7, and H2126)[101,102]. TMA-4 was from Department of Translational Molecular Pathology at the University of Texas MD Anderson Cancer Center ($n = 242$ adenocarcinoma specimens, Supplementary Data 5).

Sections were deparaffinized and subjected to antigen retrieval in Citrate buffer. After 1 h antigen block in 2.5% normal horse serum (ImmPRESS™ reagent kit, Vector Laboratories), followed by overnight incubation with mouse FASN monoclonal antibody (sc-55580, SCBT, dilution 1:500 in 2.5% normal horse serum) at 4 °C. Then slides were stained with HRP-conjugated anti-Mouse Ig (cat. MP-7402, ImmPRESS™ reagent kit, Vector Laboratories) and the peroxidase substrate kit, DAB (SK-4100, Vector Laboratories). Scoring was carried out blindly. Immunoreactivity score (0-300) was determined multiplying the staining intensity (0, no staining; 1, weak; 2, moderate; 3, strong staining) and the percentage of positive tumor cells in each core (0-100%).

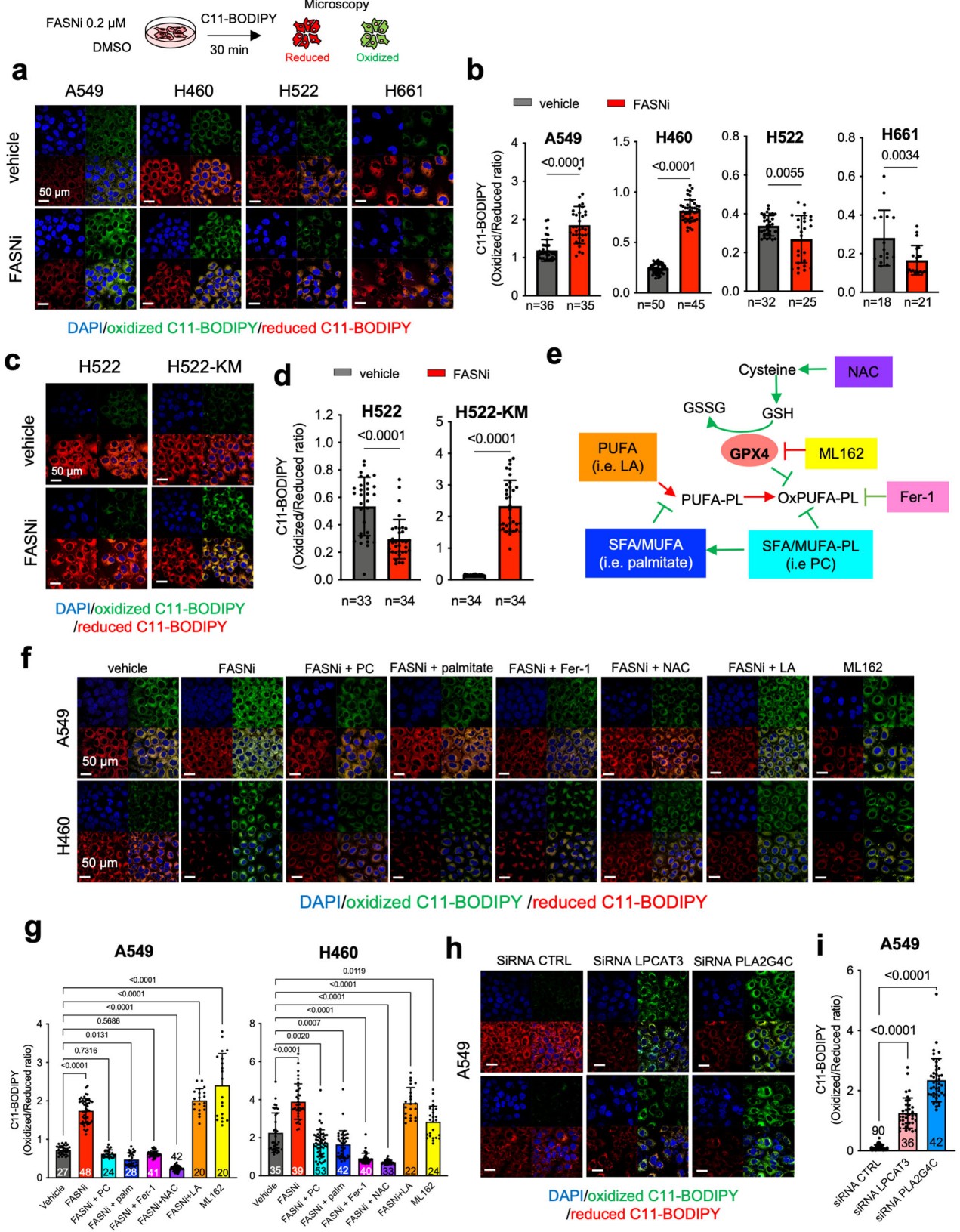

**FASNi in vivo treatment**. TVB-3664 (60 mg/kg/100 μL/mouse) was dissolved in 0.2% DMSO-PBS, sonicated and administered daily via oral gavage. We treated *CCSP-rtTA/Tet-op-Kras^G12D* mice after feeding them with doxy for 2 months. We treated NOD/SCID mice when xenografts reached ~30–50 mm³. At this time, mice were randomly assigned to either vehicle (0.2% DMSO in PBS, control) or FASNi (TVB-3664) group.

**Liproxstatin-1 in vivo rescue experiment**. Mice bearing A549 xenografts (~20–40 mm³) were randomized in three groups: vehicle (0.2% DMSO in PBS by oral gavage and 2% DMSO + 40% PEG300 + 2% Tween 80 in ddH2O i.p.), FASNi (60 mg/kg BW by gavage and 2% DMSO + 40% PEG300 + 2% Tween 80 in ddH2O i.p.) and FASNi + Liproxstatin-1 (60 mg/kg BW FASNi by gavage and 10 mg/kg BW Lip-1 i.p., as previously described[103]).

**Fig. 6 FASN and the Lands cycle are required to deflect ferroptosis in KMLC. a–d** C11-BODIPY staining in KM, KRAS-WT, and H522-KM LC cells. Oxidized and reduced C11-BODIPY are indicated in red and green, respectively. Bars indicate the relative C11-BODIPY oxidation (n, number of cells). **e** Schematic of the GPX4 axis of ferroptosis and some of its regulators. Red = pro-ferroptosis; Green = anti-ferroptosis. NAC, N-acetyl cysteine; GSH, glutathione; GSSG, glutathione disulfide; ML162, GPX4 inhibitor; Fer-1, ferrostatin-1; PUFA, Polyunsaturated fatty acids; OxPUFA, oxidized PUFA; PL, phospholipids; MUFA, monounsaturated fatty acids; SFA, saturated fatty acids; LA, linoleic acid; PC, phosphatidylcholine. **f**, **g** C11-BODIPY rescue experiments in the indicated cell lines and their quantification. **h**, **i** C11-BODIPY stain on A549 cell-line transfected with the indicated siRNAs (48 h post transfection) and their quantification. In **g** and **i** n of cells analyzed over two independent experiments are indicated in the graphs. Bars represent mean ± SD. In **b** and **d** two-tailed unpaired student t-test, in **g** multiple t-tests.

**Blood and tissue collection**. We collected blood (150 μL) from the submandibular vein and left at RT until clot formation. Lung were harvested as previously described[1]. Briefly, we cannulated the right heart and perfused anesthetized mice with PBS lacking calcium and magnesium. Lung lobes were excised, then either fixed overnight in 4% paraformaldehyde at 4 °C, or snap-frozen and stored at −80 °C for further analyses.

**Viability assays**. For MTT assays, we plated 5000 cells/0.1 mL/well in 96-well plates. The day after, we added 0.1 mL of medium containing appropriate concentrations of the compound under study to each well. After 7 days (or 5 days for ARS-1620) of treatment, 10 μL of MTT solution (5 mg/mL) was added into each well. After 4 h incubation, the medium was removed and the formazan salts dissolved in 200 μL DMSO/well for 10 min at 37 °C. The absorbance was read at 570 nm (OD 570 nm). Data are represented as mean ± SD of 3 independent experiments.

Crystal violet assays were performed as previously described[104]. Briefly, 3-4 × 10^4 cells/well were seeded in 12-well plate. At the experimental endpoint, 0.25 mL/well of crystal violet solution (0.5% of crystal violet powder in 20% methanol) was added to the plate and incubated at RT for 20 min on an orbital shaker. After removing the crystal violet solution, the plate was washed with water and let to dry overnight. Pictures were taken before adding 1 mL/well of methanol. After shaking 20 min at RT, the absorbance was read at 570 nm (OD 570 nm).

**Drug stability assay**. We seeded 2000 H460 cells/well in 50 μL complete medium in a 96-well plate at 37 °C overnight. The next day fresh RPMI + 5% FBS dialyzed medium with 200 nM TVB-3664 and added to the wells. Medium was harvested at different time points (t = 0, 60 min, 240 min, 480 min, 24 h). Briefly, 100 μL of media was removed from each well and transferred to a tube containing 125 μL of organic crash solution (Methanol (MeOH), 0.2% formic acid (FA), 50 ng/mL internal standard). Wells were washed with 100 μL ice-cold PBS and PBS was transferred to the corresponding tube containing the crash solution. Cells were trypsinized, trypsin was blocked with 100 μL of cold medium and transferred to a new tube. Wells were rinsed with 125 μL of crash solution and the rinsed solution was transferred to the tube. Tubes were vortexed for 15 s and incubated at room temperature for 10 min, spun at 13,200 rpm for 5 min. 190 μL of supernatant was transferred to an LCMS vial with insert and analyzed by LCMS (QTrap 4000). Ion Source/Gas Parameters: CUR = 310, CAD = Med, IS = 4500, TEM = 650, GS1 = 70, GS2 = 70. Buffer A: Water + 0.1% FA; Buffer B: MeOH + 0.1% FA; flow rate 1.5 mL/min; column Agilent C18 XDB column, 5 micron packing 50 × 4.6 mm size; 0−1.0 min 5% B, 1.0−1.5 min gradient to 100% B, 1.5−2.5 min 100% B, 2.5−2.6 min gradient to 5% B, 2.6−3.5 5% B; IS: tolbutamide (transition 271.2 to 91.2); TVB-3664 transition 469.19 to 437.1.

**Measurement of intracellular drug concentration**. We plated H460 cells on 10 cm dishes at 5 × 10^5/plate and incubated at 37 °C overnight. The next day, media was removed and replaced with fresh RPMI + 5% FBS dialyzed medium containing TVB-3664 (200 nM). Assay time points were 2, 4, 8 and 24 h. At each time-point, media was removed, cells washed twice with ice-cold PBS and trypsinized. Cells were resuspended in PBS at 1 × 10^6 cells/mL. After adding 250 μL of crash solution (MeOH, 0.2% Formic Acid, 0.05 ng/mL internal standard). After incubation at RT for 10 min and centrifugation, the supernatant (160 μL) was analyzed by LCMS (QTrap 4000). QTrap 4000 parameters were the same as set for drug stability assay. To analyze data, blank signal was subtracted from all points on standard curve and then plotted in GraphPad to create a standard curve. The standard curve was used to quantitate samples.

**Preparation of BSA-conjugated palmitate**. 1 mM Sodium Palmitate/0.17 mM BSA solution (6:1 molar ratio Palmitate:BSA) was prepared using ultra-fatty acid-free BSA (SIGMA ALDRICH, A6003-25G). Briefly, BSA was dissolved in 150 mM NaCl solution. Half of the solution was conjugated with Sodium Palmitate (SIGMA ALDRICH, P-9767), stirring 1 h at 37 °C; the remaining was used as vehicle control in subsequent assays.

**Cell cycle analysis**. We performed cell cycle analysis after 4 days of pharmacological treatments, or 72 h after siRNA transfection or doxy-dependent induction of shRNAs, respectively. The cells were fixed in ice-cold 70% ethanol for 30 min. RNA digestion was performed with 100 g/mL RNase A (Sigma-Aldrich, R6513) for

15 min at 37 °C. DNA staining was performed with 50 μg/mL propidium iodide (Sigma-Aldrich, P4170) for 30 min at 37 °C. The cells were analyzed using a BD LSRFortessa™ Flow Cytometer, and the cell cycle distribution was determined with FlowJo v8.7 Software.

**Oil red O staining**. We treated cells either with FASNi, 0.2 μM or with DMSO, 0.2% (i.e., vehicle) for 4 days. Cells were rinsed with 1x PBS twice, fixed with formalin (3%) for 1 h, and washed again. Cells were incubated with isopropanol (60%) for 5 min and then with ORO solution (ORO, 2 mg/mL, Alfa Aesar, A12989) for 20 min. Nuclei were stained with hematoxylin stain solution, Gill 1(RICCA, #3535-16). For quantification, ORO staining was extracted with 100% isopropanol. The absorbance was read at 492 nm.

**Protein extraction and immunoblot**. Tissues or cell lines were lysed in ice-cold RIPA lysis buffer containing cOmplete™ protease inhibitors cocktail and Phos-STOP™ phosphatase inhibitors cocktail (Millipore Sigma). 30 μg of total protein were separated using Midi Criterion TGX Stain-Free precast gels (Bio-Rad, 5678024), transferred onto nitrocellulose membranes (Bio-Rad, 1620112), and then blocked with 5% Blotting-Grade Blocker (Bio-Rad, 1606404) in TBS for 1 h, at RT. The membranes were incubated overnight at 4 °C with the indicated primary antibodies: FASN (C20G5) rabbit mAb (CST #3180, 1:1000 dil); SCD1 (C12H5) rabbit mAb (CST #2794, 1:1000 dil); AMPKα Antibody rabbit pAb (CST #2532, 1:1000 dil); phospho-AMPKα (Thr172) (40H9) rabbit mAb (CST #2535, 1:1000 dil); ACC1 (C83B10) Rabbit mAb (CST #3676, 1:1000 dil); Phospho-ACC1 (Ser79) (D7D11) rabbit mAb (CST #11818, 1:1000 dil); KRAS (234-4.2) mouse mAb (Millipore Sigma #MABS194, 1:700 dil); pan RAS (C-4) mAb (SCBT #SC-166691, 1:500 dil); GAPDH (6C5) mAb (SCBT #SC-32233, 1:500 dil); LPCAT3 rabbit polyclonal (Novus Bio #NBP3-04752, 1:1000 dil); Cofilin (D3F9) rabbit mAb (CST #5175, 1:1000 dil). Cofilin and GAPDH were used as loading controls. Donkey anti-mouse IgG (H + L) crossed-absorbed secondary antibody-DyLight 680 (invitrogen #SA5-10170, dil 1:5000); Goat anti-rabbit IgG (H + L) crossed-absorbed secondary antibody-DyLight 800 (invitrogen #SA5-10036, dil 1:5000); Goat anti-mouse IgG (H + L) crossed-absorbed secondary antibody-DyLight 800 (invitrogen #SA5-10176, dil 1:5000) were used secondary antibody. Blots were analyzed using Odyssey Scanner and Image Studio Software (LI-COR).

**Malonyl-CoA quantification**. Malonyl-CoA concentration was assessed using the human malonyl Coenzyme A ELISA kit (MyBioSource, MBS705079), following manufacturer's instructions. Test samples were prepared in triplicates.

**Measurement of fatty acid oxidation (FAO)**. We used a fatty acid oxidation colorimetric assay kit (Biomedical Research Service Center, State University of New York, E-141) as previously described[105]. Samples were assayed in triplicate.

**NADPH quantification**. NADPH in lysates of cell lines or A549 xenografts was determined using a colorimetric NADPH Assay Kit (Abcam's, ab186031), following the manufacturer's instructions on LC cells and A549 xenografts treated with either FASNi or vehicle for 4 days. Test samples were prepared in triplicates.

**AMP quantification**. We used a colorimetric AMP colorimetric assay kit (Bio-Vision, K229-100) to detect AMP in cell lysates (~1 × 10^7) and A549 xenografts (~10 mg) following the manufacturer's instructions. Samples were assayed in triplicate. 1 mM AMP Standard was used as positive control.

**Lipase enzymatic activity**. We used a Lipase activity colorimetric assay (Abcam, ab102524) that quantifies glycerol deriving from TAG hydrolysis mediated by ATGL. Briefly, 2 × 10^6 cells corresponding to each experimental condition tested were analyzed in triplicate. Lipase was used as positive control, while samples without the reaction mix were used background controls.

**Cu(I)-catalyzed azide-alkyne cycloaddition reaction (Click-iT chemistry)**. Cells were grown on coverslips in a 12-well plate and kept either with FASNi, 0.2 μM or with vehicle (DMSO, 0.2%) for 4 days. Then, we incubated them with 20 μM arachidonic acid alkyne (Cayman Chemical, 10538) in RPMI with 2% ultra-fatty

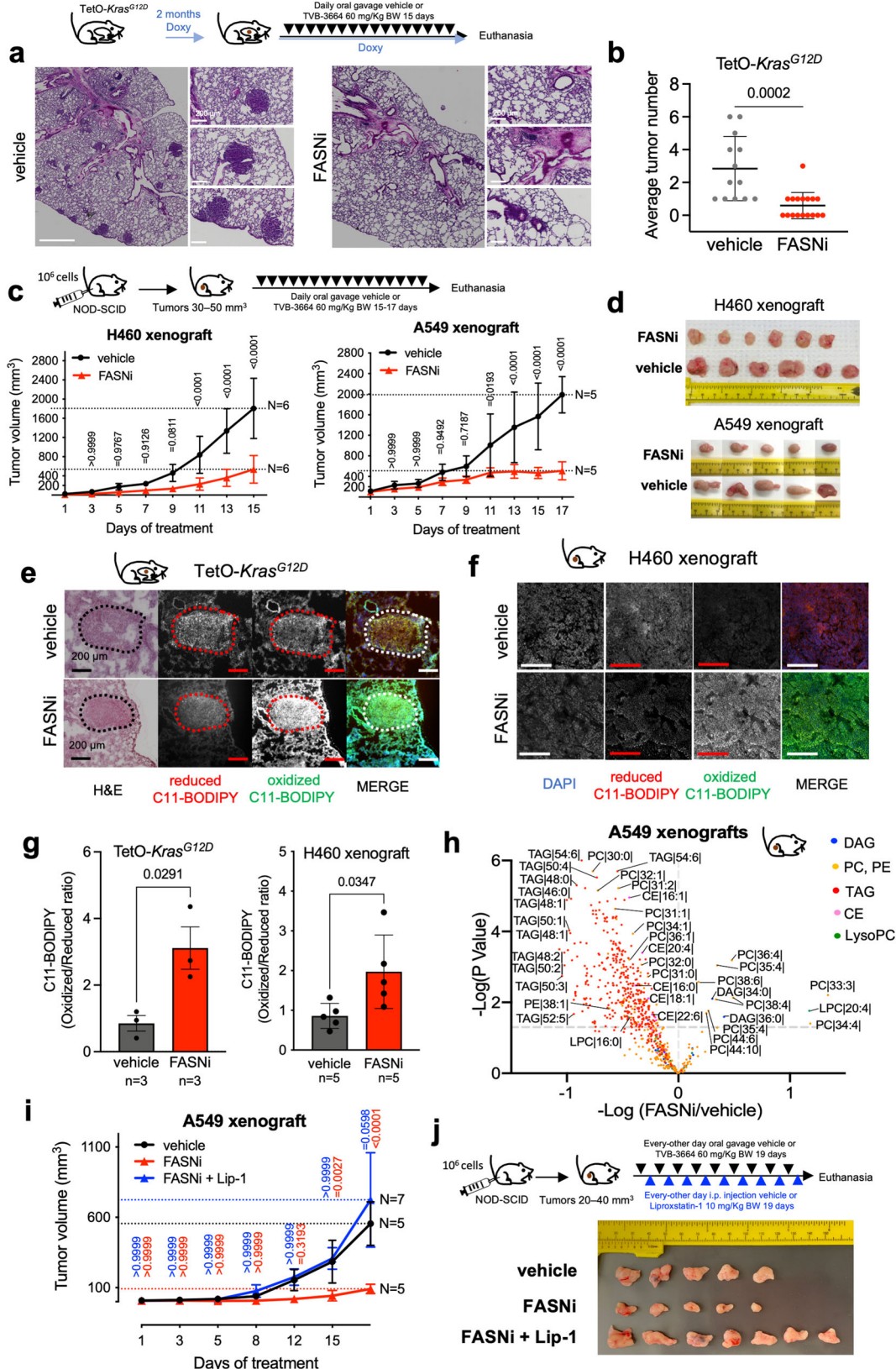

acid-free BSA (SIGMA ALDRICH, A6003-25G) for 6 h. After fixation with 3% paraformaldehyde in PBS for 15 min and permeabilization with 0.25% Triton X100 in PBS for 15 min, cells were washed with 1% BSA in PBS. The Click-iT reaction cocktail was prepared according to manufacturer's instructions[106]. Briefly, Click-iT™ Cell Reaction Buffer Kit (Thermo Scientific, C10269) was mixed with CuSO4 (1 mM final concentration) and Alexa-Fluor 488 azide (10 μM final concentration,

Thermo Scientific, A10266). After staining nuclei with DAPI, cells were washed again, and mounted on a slide using Fluoromount-G medium.

**C11-BODIPY staining**. We stained fixed cells with 2.5 μM C11-BODIPY581/591 (SIGMA ALDRICH, D-3861) for 60 min. The ROS-dependent oxidation of the

**Fig. 7 FASNi is effective in KMLC in vivo. a, b** Representative H&E pictures of the lungs of TetO-*Kras* mice treated as indicated and tumor number quantification. *n* = number of mice. **c, d** In vivo growth curves and post-resection pictures of H460 and A549 xenografts in NOD/SCID mice treated as indicated. **e–g** Representative pictures of C11-BODIPY staining of the lungs of TetO-*Kras* and of H460 xenografts, and their quantification. Dotted circles in **e** indicate lung tumor identified via H&E staining. **h** Volcano plot showing lipid species identified by MS/MS differentially represented in FASNi *versus* vehicle (*n* = 5 mice/group). **i** In vivo growth curves and **j** post-resection pictures of A549 xenografts in NOD/SCID mice treated as indicated in the schematic. Color-coded statistics indicate comparisons to vehicle. Lip-1, Liproxstatin-1. In **b, c, g, i** data are represented as mean ± SD. In **b, g** unpaired two-tailed student *t*-test, in **c**, multiple *t*-tests followed by Benjamini–Hochberg FDR, in **h** generalized linear model followed by Benjamini–Hochberg FDR, and in **i** two-way ANOVA plus Sidak's comparisons.

polyunsaturated butadienyl portion of this lipid probe results in a shift of the fluorescence emission peak from ~590 nm (reduced: red) to ~510 nm (oxidized: green)[107,108]. Briefly, cells were treated with 0.2 µM FASNi or 0.2% DMSO, for 4 days. In rescue experiments, cells were co-treated with FASNi and one of the following: palmitate (100 µM), phosphatidylcholine (PC, 100 µM), or Ferrostatin-1 (Fer-1, 1 µM) for 4 days. 5 mM N-acyl-cysteine (NAC) was added for 60 min after 4 days of FASNi treatment. For snap-frozen samples, we used 10 µm-thick cryo-sections of A549 xenografts (5 mice/group) and lungs of CCSP-rtTA/Tet-op-Kras (3 mice/group). Samples were incubated with C11-BODIPY581/591, washed with PBS, fixed with 10% formalin for 1 h, counterstained with DAPI and mounted using Fluoromount-G medium (Thermo Scientific). For cells, three images per slide were acquired using a Zeiss LSM 710 confocal microscope equipped with a Plan-Apo 63x/1.4 oil DIC M27 objective and ZEN 2010 B SP1 software (Zeiss). For tissues, three images per sample were acquired using a Lionheart FX (BioTek) and Gen5 software (v3.06). We quantified the images with ImageJ software (version 1.46; NIH, Bethesda, MD, USA).

**Single-choice and competitive PUFA/SFA uptake assay.** We used H460 cells, as the representative human KMLC cell-line, AA-alkyne as PUFA proxy that we then conjugated to an Alexa-Fluor 594-azide (red), and C16-BODIPY (green, Thermo Scientific) as SFA representative. Briefly, we treated H460 cells with either vehicle (0.2% DMSO) or FASNi (0.2 µM) for 4 days.

Then, both treatment groups were randomized in ten subgroups receiving the following FA mixtures in RPMI with 2% ultra-fatty acid-free BSA (SIGMA ALDRICH, A6003-25G) for 6 h: (1) 20 µM AA alone; (2) 10 µM AA alone; (3) 20 µM C16 alone; (4) 10 µM C16 alone; (5) 5 µM AA + 5 µM C16 (AA:C16 ratio 1:1); (6) 10 µM AA + 10 µM C16 (AA:C16 ratio 1:1); (7) 16 µM AA + 4 µM C16 (AA:C16 ratio 4:1); (8) 15 µM AA + 5 µM C16 (AA:C16 ratio 3:1); (9) 4 µM AA + 16 µM C16 (AA:C16 ratio 1:4); (10) 5 µM AA + 15 µM C16 (AA:C16 ratio 1:3). As reported for the AA uptake, after fixation with 3% formaldehyde in PBS for 15 min and permeabilization with 0.25% Triton X100 in PBS for 15 min, cells were washed with 1% BSA in PBS. The Click-iT reaction cocktail was prepared according to manufacturer's instructions[106]. Briefly, Click-iT™ Cell Reaction Buffer Kit (Thermo Scientific, C10269) was mixed with CuSO4 (1 mM final concentration) and Alexa-Fluor 594 azide (10 µM final concentration, Thermo Scientific). After staining nuclei with DAPI, cells were washed and mounted on a slide using Fluoromount-G medium. Three images per slide were acquired via Zeiss LSM 710 confocal microscope equipped with a Plan-Apo 63x/1.4 oil DIC M27 objective, setting the same laser intensity for both green and red channels. Single channel and total signal intensities per cell were quantified using ImageJ software (version 1.46; NIH, Bethesda, MD, USA).

**Plasmids and virus production.** pBabe-Kras^G12D (#58902) and LT3GEPIR (#111177) were from Addgene. The insert expressing the shRNA was cloned into the vector *LT3GEPIR* in the XhoI and EcoRI sites. The shRNAs target sequences were selected either from the library described by Feldmann et al.[109] or from the splashRNA database[110] (http://splashrna.mskcc.org/). The hairpin targeting sequences are: *shLPCAT3 #1* 5′-GCCTCTCAATTGCTTATTTTA-3′; *shLPCAT3 #2* 5′-AAGGAAAGAGAAGTTAAA-3′; *shPLA2G4C #1* 5′-CAGAATGAATGT-GATAGTTCA-3′; *shPLA2G4C #2* 5′- ACATGGTTATCTCTAAGCAAA-3′; *shGPX4 #1* 5′-GTGGATGAAGATCCAACCCAA-3′; *shGPX4 #2* 5′-AGGCAA-GACCGAAGTAAACTA-3′. *shKRAS #10* 5′-AAGTTGA-GACCTTCTTAATTGGT-3′; *shKRAS #40* 5′-TCAGGACTTAGCAAGAAGTTA-3′. Production of lentiviruses was performed as previously described[12].

**siRNA.** Predesigned FASN siRNAs were from Millipore Sigma (#1, SASI_Hs01_00057850; #2, SASI_Hs01_00057851; #3, SASI_Hs02_00336920; #4, SASI_Hs01_00057849). A custom siRNA library (Supplementary Data 9), universal MISSION® siRNA Universal Negative Control #1 (cat. SIC001) and #2 (cat. SIC002) were from Millipore Sigma. We used the DharmaFECT 4 Transfection Reagent (Thermo Scientific) for siRNA transfection. RNA was extracted using TRIzol Reagent (cat. 15596, Life Technologies) and retrotranscribed using iScript cDNA Synthesis kit (cat. 170-8891, Bio-rad). Knock-down efficiency was evaluated 48 h after transfection via real-time PCR using the PowerUP™ SYBR® Green Master Mix (cat. A25742, Thermo Fisher Scientific) and custom designed primers (Supplementary Data 9).

**RNA-seq and bioinformatic analysis.** We extracted total RNA with TRIzol. Residual genomic DNA was removed with the Turbo DNA-free kit (AM1907, ThermoFisher). One microgram of total DNase-treated RNA was used for library preparation with the NEBNext Ultra II Directional RNA Library Prep kit. Reads were aligned to the human hg38 reference genome using STAR (v3.7.3a)[111]. Gencode annotation for human (version v37) was used as reference alignment annotation and downstream quantification. Gene level expression was calculated using featureCounts (v2.0.1)[112] using intersection-strict mode by exon. Counts were calculated based on protein-coding genes from the annotation file. Low expressed genes were filtered using a per time-point approach with RPKM >= 0.5 in all samples in one or the other time-point. Differential expression was performed in R using *DESeq2* version 1.34[113]. Surrogates variables were calculated using *sva* version 3.42[114] and included in the modeling. We estimated log2 fold changes and *p*-values. P-values were adjusted for multiple comparisons using a Benjamini–Hochberg correction (FDR). Differentially expressed genes were considered for FDR < 0.05. Gene list enrichment analysis was carried out using Enrichr[115,116] (https://amp.pharm.mssm.edu/Enrichr/#).

**Laser-capture microdissection (LMD) of tumor tissue.** LMD was performed at the Histopathology Core of UT Southwestern Medical Center as previously described[117]. Briefly, 10 µm cryo-sections of tumors were mounted onto PEN membrane glass slides (Thermo Fisher Scientific, LCM05220). Sections were then stained using Histogene™ Staining Solution (Thermo Fisher Scientific, KIT0415), washed with high-performance liquid chromatography (HPLC)-grade water and subjected to LMD using a Leica LMD System. LMD sections were recovered in 0.2 mL tubes, immediately snap-frozen and stored at −80 °C until the analysis. Four replicates per sample were prepared.

**Solvents and reagents.** All the HPLC or liquid chromatography–mass spectrometry (LC/MS) grade solvents were from Sigma-Aldrich (St Louis, MO, USA). SPLASH LipidoMix™ standards, were from Avanti Polar Lipids (Alabaster, AL, USA). Fatty acid (FA) standards (FA(16:0{$^2H_{31}$}), FA(18:1ω9{$^2H_5$}) and FA(20:4 ω6{$^2H_8$}) were from Cayman Chemical (Ann Arbor, MI, USA). An eVol® precision pipette equipped with a glass syringe (Trajan Scientific, Austin TX, USA) was used for the addition of FA standards.

**Sample preparation for MS/MS^ALL and GC/MS.** Laser-captured micro-dissected samples, cell pellets containing $2.5 \times 10^5$ cells or 10 µL of serum were transferred to glass tubes for Liquid-Liquid Lipid Extractions (LLE). For xenografts, 100 mg of tissue were transferred to a 2.0 mL pre-filled Bead Ruptor tube (2.8 mm ceramic beads, Omni International, Kennesaw, GA, USA), and homogenized in 1 mL of methanol/dichloromethane (1:2, v/v) using a Bead Ruptor (Omni International). Aliquots equivalent to 0.5 mg of tissue were used for LLE.

**MALDI imaging mass spectrometry (MALDI-IMS).** Ten micrometer thick sections were mounted on Superfrost Microscope Slides (Fisherbrand) and stored at −80 °C. They were processed by the Chemical Imaging Research Core, UT MD Anderson Cancer Center. A HTX Sprayer M5 matrix applicator (HTX Technologies, LLC., Chapel Hill, NC, USA) was used to apply 2,5-Dihydroxybenzoic acid (DHB) for positive mode, at a flow rate of 100 µL/min with temperature set to 75/30 °C for sprayer/tray. DHB was dissolved to a concentration of 15 mg/mL in 50% acetonitrile with 0.1% TFA. Before being loaded into the mass spectrometer, the slides were placed in a MALDI plate, scanned using an EPSON scanner (Epson, Suwa, Japan), and the sections of tissue were mapped into High Definition Imaging software (HDI 1.4; Waters, Milford, MA, USA). DHB is considered the standard matrix for lipid studies[118]. MALDI-MS imaging was performed using a Water Synapt G2 Si (Waters Corporation, Milford, MA). Data were acquired with a spot of 60 µm with 300 laser shots at 1 kHz, using a pulse energy with an average of 25 µJ. The laser intensity was adjusted to 60%. The mass range was 50–2000 *m/z* (as it is typically done for lipids) and the instrument was calibrated using peak signals from red phosphorus. We converted all files into imzML using the Waters High Definition Imaging (HDI) software. We performed all data image visualization and data analysis using msIQuant[119]. Within msIQuant, Peak option signal to noise ratio (SNR) was set at 3.0 and peak Group Detection within 0.5 Da as for marker selection, Marker mass range was ±0.05 Da and the maximum intensity in range was used as intensity method.

Lipid identification was performed comparing observed peaks (experimental mass) with the theoretical values reported in the Lipid MAPS (http://www.lipidmaps.org/)[120] and Madison Metabolomics Consortium (http://mmcd.nmrfam.wisc.edu/)[121] databases (theoretical mass), using the mass accuracy (0.5 Da) as a tolerance window. In this way, we identified a single candidate for each peak in the spectrum. Using experimental and theoretical values, mass error (ppm) was calculated as $\frac{observed\ m/z - theoretical\ m/z}{observed\ m/z} \times 10^6$. Tentative identification of lipid species is available in Supplementary Data 1.

Under the pathology guidance on corresponding H&E images, we manually annotated regions of interest (ROI) of cancer areas and surrounding stroma to mimic laser microdissection analysis in msIQuant version 2.0[119]. To provide a "semi-quantitative" analysis for each ROI, after total ion count (TIC) normalization in msIQuant, we normalized peak intensity by the area (pixel). Number of pixels, sum of intensities for all pixels, average intensity per mm², average intensity per pixel, standard deviation of intensity per pixel, relative standard deviation of intensity per pixel, median intensity per pixel, lower quartile (Q1) intensity per pixel, higher quartile (Q3) intensity per pixel, and the minimum and maximum intensities are reported in Supplementary Data 2–4 and Supplementary Data 8.

One limitation of our MALDI-MS experiments is that TAG are prone to in-source fragmentation and they do not ionize well with DHB matrix. Therefore, several TAG species might not have been detected in our settings. Another limitation is that MALDI-MS provides a tentative species identification. Accordingly, we employed this technique to complement the HPLC-MS/MS and MS/MS$^{ALL}$ lipidomics that was performed on the same samples in parallel. MALDI-MS data generated in this study have been submitted to the DRYAD database under https://doi.org/10.5061/dryad.gtht76hq1 (ref. [122]).

**Lipidomic experiments**. After 4 days of treatment, with vehicle or FASNi, cells were either washed twice with cold PBS and harvested for MS/MS$^{ALL}$, or incubated with 3 mM Ethyl acetate-1,2 $^{13}C_2$ (SIGMA ALDRICH, 283819) for 7 h (1 h for the time-lapse experiment) to measure de novo palmitate synthesis by GC/MS.

**Lipid profiling by direct-infusion MS/MS$^{ALL}$**. The LLE was performed at RT through a modified Bligh/Dyer extraction technique. Briefly, 3 mL of methanol/dichloromethane/water (1:1:1, v/v) were added to the samples. The mixture was vortexed and centrifuged at $2671 \times g$ for 5 min. The organic phase (bottom phase) was collected and dried under $N_2$. The extracts were resuspended in 600 µL of dichloromethane/methanol/isopropanol (2:1:1, v/v/v) containing 8 mM ammonium fluoride (NH4F) and 33 µL of 3:50 diluted SPLASH LipidoMix™ internal standard. Extracts were infused into a SCIEX quadrupole time-of-flight (QTOF) TripleTOF 6600+ mass spectrometer (Framingham, MA, USA) via a custom configured LEAP InfusePAL HTS-xt autosampler (Morrisville, NC, USA). Electrospray ionization (ESI) source parameters were, GS1 25, GS2 55, curtain gas (Cur) 20, source temperature 300 °C and ion spray voltage 5500 V and −4500 V in positive and negative ionization mode, respectively. GS1 and 2 were zero-grade air, while Cur and CAD gas was nitrogen. Optimal declustering potential and collision energy settings were 120 V and 40 eV for positive ionization mode and −90 V and −50 eV for negative ionization mode. Samples were infused for 3 min at a flow rate of 10 µL/min. MS/MS$^{ALL}$ analysis was performed by collecting product-ion spectra at each unit mass from 200-1200 Da. Analyst® TF 1.7.1 software (SCIEX) was used for TOF MS and MS/MS$^{ALL}$ data acquisition. Data analysis was performed using an in-house script, LipPy. This script provides instrument quality control information, isotopic peak corrections, lipid species identification, data normalization, and basic statistics.

**Fatty acid profiling by GC-MS**. Total fatty acid profiles were generated by a modified GC-MS method previously described[123]. The lipid extract was spiked with 100 µL of 0.5 µg/mL FA standard mixture (FA(16:0{$^2H_{31}$}), FA(20:4 ω6{$^2H_8$}) and FA(22:6 ω3{$^2H_5$}) in methanol, then hydrolyzed in 1 mL of 0.5 M potassium hydroxide solution prepared in methanol at 80 °C for 1 h. Hydrolyzed FA were extracted by adding 2 mL of dichloromethane/water (1:1, v/v) to the sample in hydrolysis solution. The mixture was vortexed and centrifuged at $2671 \times g$ for 5 min. The organic phase (bottom phase) was collected and dried under $N_2$. To analyze FA present in the polar lipid fraction, a three-phase extraction method was performed, as already described[53]. Briefly, lipids were extracted with water, methyl acetate, acetonitrile (ACN) and Hexane (1:1:0.75:1). After centrifugation, polar lipids (upper phase) were collected and dried. Total or polar FA samples were resuspended in 50 µL of 1% triethylamine in acetone, and derivatized with 50 µL of 1% pentafluorobenzyl bromide (PFBBr) in acetone at RT for 25 min in capped glass tubes. Solvents were dried under $N_2$, and samples were resuspended in 500 µL of iso-octane. Samples were analyzed using an Agilent 7890/5975 C (Santa Clara, CA, USA) by electron capture negative ionization (ECNI) equipped with a DB-5MS column (40 m × 0.180 mm with 0.18 µm film thickness) from Agilent. Hydrogen (carrier gas) flow rate was 1.6 mL/min and injection port temperature was set at 300 °C. Sample injection volume was 1 µL. Initial oven temperature was set at 150 °C, and then increased to 200 °C at a 25 °C/min, followed by an increase of 8 °C/min until a temperature of 300 °C was reached and held for 2.2 min, for a total run time of 16.7 min. FA were analyzed in selected ion monitoring (SIM) mode. The FA data were normalized to the internal standards. Fatty acid with carbon

length $C \leq 18$ were normalized to FA(16:0{$^2H_{31}$}), $C = 20$ were normalized to FA(20:4 ω6{$^2H_8$}), and $C = 22$ were normalized to FA(22:6 ω3{$^2H_5$})). Data were processed using MassHunter software (Agilent).

**High-performance liquid chromatography–mass spectrometry (HPLC-MS/MS)**. PDXs and human lung cancer specimens were stored at −80 °C. LLE was performed using a modified Bligh/Dyer extraction technique. The extracts were resuspended in 30 µL of dichloromethane/methanol/isopropanol (IPA) (2:1:1, v/v/v) containing 8 mM NH4F and SPLASH LipidoMix™ internal standards. Reversed-phase chromatographic separation was achieved with the Acclaim C30 column: 3 µm, 2.1 × 150 mm (Thermo Fisher Scientific, Waltham, MA). The column was maintained at 35 °C and tray at 20 °C. Solvent A was composed of 10 mM ammonium formate (AF, LC-MS grade) in 60:40 ACN:water (LC-MS grade) with 0.1% formic acid (FA, LC-MS grade). Solvent B was composed of 10 mM AF with 90:10 IPA:ACN with 0.1% FA. The flow rate was 250 µL/min, and the injection volume was 10 µL. The gradient was 50% solvent A (3 to 50%). The Orbitrap (Thermo) mass spectrometer was operated under heated electrospray ionization (HESI) in positive and negative modes separately for each sample. The spray voltage was 3.5 and 2.4 kV for positive and negative mode, the heated capillary was held at 350 °C and heater at 275 °C. The S-lens radio frequency (RF) level was 45. The sheath gas flow rate was 45 units, and auxiliary gas was 8 units. Full scan (m/z 250–1200) used resolution 30,000 at m/z 200 with automatic gain control (AGC) target of $2 \times 10^5$ ions and maximum ion injection time (IT) of 100 ms. Normalized collision energy (NCE) settings were 25, 30, 35%.

Lipid identification and relative quantification were performed with LipidSearch 4.1 software (Thermo) as previously described[124]. The search criteria were as follows: product search; parent m/z tolerance 5 ppm; product m/z tolerance 10 ppm; product ion intensity threshold 1%; filters: top rank, main isomer peak, FA priority; quantification: m/z tolerance 5 ppm, retention time tolerance 1 min. The following adducts were allowed in positive mode: +H, +NH4, +HH2O, +H2H2O, +2H, and negative: −H, +HCOO, +CH3COO, −2H +Na, +K.

**Determination of phospholipid oxidation by UV/Vis spectroscopy**. For estimation of conjugated dienes, lipids from cell samples were fractioned in the polar (phospholipids) and neutral fractions, using the 3PLE method[53]. Lipid fractions were dried and dissolved in ethanol. For both fractions, the spectra of ultraviolet absorption between 200 nm and 300 nm were measured as previously described[65].

**CellROX™ Green assay and live imaging**. Sub-confluent H460 cells in 6-well plates were treated with either vehicle, TVB-3664 (0.2 µM) or combination of TVB-3664 (0.2 µM) and 16:0–18:1 PC (50 µM). After 72 h, CellROX Green reagent was spiked in at final concentration of 5 µM, and incubated for 30 min at 37 °C. Plates were imaged over 1 h with one image taken every 5 min. Time-laps imaging was conducted using Lionheart FX automated microscope (BioTek) and Gen5 software.

**Microsomal LPCAT3 activity assay**. To measure the substrate specificity for the acyltransferase activity of LPCAT3, we adapted an established method[59] using microsomes purified from A549 cells, stably transduced with the doxy-inducible LT3GEPIR-shLPCAT3#1 (5′-GCCTCTCAATTGCTTATTTTA-3′) after 48 h incubation in RPMI supplemented with 5% heat-inactivated FBS with or without doxy. For microsome preparation, cells were pelleted at $300 \times g$, resuspended in homogenization buffer [50 mM Tris-HCl, pH 7.4, 250 mM sucrose, 1 mM EDTA, 20% (w/v) glycerol, and complete protease inhibitor cocktail (Roche)] and lysed using a Sonics Vibra-Cell probe sonicator (Newton, CT). After centrifugation at $12,000 \times g$ for 20 min at 4 °C, the supernatant was collected and centrifuged at $100,000 \times g$ for 60 min at 4 °C. The microsomal pellet was resuspended in assay buffer (10 mM Tris-HCl, pH 7.4, 150 mM NaCl, 1 mM EDTA) and proteins quantified using Bradford reagent (Bio-Rad). Hexadecanoyl (pamitoyl; 16:0-CoA, cat. 870716P), 9Z,8Z,11Z,14Z-eicosatetraenoyl (arachidonoyl;20:4-CoA cat. 870721P), and 1-(10Z)-heptadecenoyl-2-hydroxy-lysophosphatidylcholine (17:1-LPC, cat. 855677C) were from Avanti Polar Lipids (Alabster, AL). Stock solutions were made as follows: 60 mM 20:4-CoA or 16:0-CoA in 100% methanol; 200 mM 17:1-LPC in assay buffer; while the final concentrations were: 50 ng total protein from microsomes, 3 mM of LysoPC 17:1, 3 mM of each fatty Acyl-CoA esters (either 16:0 or 20:4, alone or in combination) and 12.5 mM fatty acid-free BSA in assay buffer (10 mM Tris-HCl, pH 7.4, 150 mM NaCl, 1 mM EDTA). The acyl-transferase assay was performed at 37 °C for 30 min. The reaction was stopped with methanol-chloroform (2:1, v/v) and lipids were extracted by the Blygh and Dyer method, dried, resuspended in dichloromethane/methanol/isopropanol (2:1:1, v/v/v) containing 8 mM ammonium fluoride (NH4F) and 33 µL of 3:50 diluted SPLASH LipidoMix™ internal standards prior to HPLC-MS/MS. Phospholipid products of the LPCAT3 assay were measured using a Orbitrap (Thermo) mass spectrometer in negative mode. The acetate adducts [M + CH3COO]- ions were used as target ions. Quantitation was performed using LiPy script software and using PC, 15:0–18:1(d7) as an internal standard.

**Statistical analysis**. Data analysis was performed using Microsoft Excel for Mac (version 16.34) and GraphPad Prism version 9 (GraphPad Software, San Diego, CA, USA, www.graphpad.com). All data presented are expressed as mean ± SD of

two or more biological replicates/biologically independent experiments (*n* values in each figure/figure legend). The significance of the results was assessed using two-tailed unpaired Student's *t*-test to compare two groups. When more than two groups were compared, one- or two-way ANOVA was used followed by Dunnett's, Tukey's or Sidak's post-test or multiple two-tailed *t*-tests followed by FDR correction for multiple comparison.

**Reporting summary**. Further information on research design is available in the Nature Research Reporting Summary linked to this article.

## Data availability

RNA-seq data were deposited in GEO under the accession number GSE168782. MALDI-MS data generated in this study have been submitted to the DRYAD database under https://doi.org/10.5061/dryad.gtht76hq1 [122]. The publicly available databases LIPID MAPS (https://www.lipidmaps.org/), METLIN (https://www.metlin.scripps.edu) and Human Metabolite Database (https://www.hmdb.ca) were used for the lipidomic annotation. The Cancer Cell Line Encyclopedia database (CCLE, https://sites.broadinstitute.org/ccle/datasets) was used to retrieve *FASN* expression data. All data supporting the findings of this work are available within this article, Supplementary Information, Source data and the peer review file. Further requests for resources, reagents and data should be directed to and will be fulfilled by the corresponding author. Source data are provided with this paper.

## Code availability

All commercially or publicly available codes and software used are listed throughout the manuscript and in the reporting summary. The custom-made LipPy script[53] can be accessed for academic purposes only and upon request to goncalovale@gmail.com and jeffrey.mcdonald@utsouthwestern.edu.

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

## Acknowledgements

The preliminary part of this work was done at UT Southwestern medical Center. We thank CPRIT RP140672, CPRIT RP160652, The University of Cincinnati College of Medicine, NIH/NCI R01 CA259845-01A1 and The LCS Foundation (Ohio) (P.P.S.). Lung Cancer SPORE (P50CA70907) (J.D.M., P.P.S., J.W.S., K.H., I.I.W.), the Harold C. Simmons Cancer Center through NCI Cancer Center support grant and 2P30CA016672. J.G.M. is supported in part by NIH HL020948. Cancer Prevention and Research Institute of Texas CPRIT RP160652, The University of Texas MD Anderson Cancer Center. Y.T.M.A. 310 was funded in part by the Yale SPORE in Lung Cancer P50 CA 196530 (PI: Roy Herbst). We thank Dr. Monte Winslow for kindly providing the murine LSL-KRAS$^{G12D}$ cell lines, John M. Shelton for helping set up LMD conditions, Dr. Ken Greis and Dr. Robert Ross at UC Proteomics and Metabolomics Laboratory to provide guidance and instrumentation for the HPLC-MS/MS experiments, Dr. Peter Pathrose for performing KRAS mutation analysis on human lung cancer samples, Dr. Maria F. Czyzyk-Krezeska and Dr. David Plas for critically revising the manuscript.

## Author contributions

C. Bartolacci and C.A. equally contributed to this study; C. Bartolacci, C.A., and P.P.S. designed the study; C. Bartolacci, C.A., G.V., M.M., A.C.C., D.L.B. performed experiments; S.B. and C.A. performed bioinformatic analysis; G.V., J.G.M., and D.L.B. gave technical assistance with lipidomic experiments; A.C.C. performed MALDI acquisition; J.G.M., J.D.M., K.P., J.S., B.G. provided some key reagents and resources; G.K. provided TVB-3664 and TVB-2632; S.L.S. provided the human lung cancer sample; K.H., I.I.W. provided pathology assistance with YTMA 310 and TMA 4; D.L. performed immuno-histochemistry analysis on TMA 4; M.G.R. and L.M.S.S. provided tissue acquisition and pathology assistance with TMA 4; C. Behrens clinical database management and H.K. mutational database acquisition and management (TMA 4); C.A., C. Bartolacci, and P.P.S. wrote and revised the manuscript with comments from all authors.

## Competing interests

G.K. is CEO and Chief Scientific Officer at Sagimet Biosciences where TVB-3664 has been developed. The other authors declare no competing interests.
