## [Peer Review File · Nature Communications]

Title: Targeting de novo lipogenesis and the Lands cycle induces ferroptosis in KRAS-mutant lung cancerREVIEWER COMMENTS

Reviewer #1 (Remarks to the Author); expert on MALDI imaging and lipidomics:

The paper by et al. aims at the identification and characterization of deregulated metabolic networks in KRAS mutant lung cancer. The beauty of the paper is that it has deployed various different, but genetically related samples to study lipogenesis and its role in ferroptosis. This multilevel “omics” analysis combined with different imaging technologies is the interdisciplinary approach needed to tackle the complexity of this disease. The outcome that FASN and the lipid cycle are required to circumvent ferroptosis provides new targets in the treatment of this lung cancer subtype. A significant amount of the conclusions are based on the lipid observations using mass spectrometry and spatially resolved MS, looking at the different lipid classes observed. There are some major issues with the interpretation of these lipids that I am concerned with and merit further elaboration. In particular the interpretation of up- and downregulation of the lipids under certain stimuli is based on the observation of peak intensity differences, but a solid quantification is missing. As a result I cannot recommend the paper in its current form, and would suggest some major changes.

The m/z values listed on page 13 seem more precise than the typical resolution of the MS system used normally offers. Can the authors explain how they have come to such a precise parent ion isolation? Also, as no m/z selection window is specified for the MS/MS MSI analysis there is a distinct possibility that other molecules have been selected for MS/MS, Quadrupole based isolation is typically not precise with 3 digits!

Unfortunately, the results in figure 1D are difficult to interpret due to the lack of the H&E stained image and the histological annotation. Additionally, the authors concluded an enrichment of TAG, SM and PC's based on this figure, which implies that the method used is quantitative. However extensive literature on MALDI-MSI indicates that the use of isotopically labelled standards to perform a quantitative study. I suggest the authors add those experiments and their interpretation of the results to substantiate the current claim. On a similar topic, it is unclear what the ion types observed for the lipid species are. It is known that particular apoptotic/necrotic and perhaps ferroptotic processes result in altered Na/K ratios that can result in local efficient cationization of specific lipid species in MSI. These are often mistaken for upregulation/enrichment, but actually reflect local cation availability. Examination of the supplemental figures (1 and 8) also does not tell me which ion types or m/z values have been observed. The figure captions just speak about MALDI pictures. Additionally the supplementary data provided is void of the MSI data or any of its annotation and interpretation.

The interpretation of the imaging data, and hence the conclusions drawn from the MSI figures would be greatly facilitated if the corresponding H&E images of the identical sections would have been added. Currently this is impossible. Appropriate tumor, stromal and necrotic annotation by a trained pathologist could also be related to the different lipid markers commonly found in tumor MSI.

Reviewer #2 (Remarks to the Author); expert on KRAS-mutant lung cancer and metabolism:

It has been known for a long time that FASN is upregulated in several cancer types and that Mutant KRAS (KM) upregulates its expression. In this manuscript, using comprehensive metabolic and functional analysis in vitro, Bartolacci et al found that KM upregulates FASN to increase the synthesis of SFA and MUFA, which can prevent and repair lipid peroxidation in lung cancer cells, thereby inhibiting ferroptosis and supporting the survival of lung cancer cells. This study provides new insight of the mechanism of FASN upregulation in KRas-driven lung tumorigenesis. Most experiments are well designed and the results support the conclusion.

Before accepting for publication, one major concern is that most mechanistic studies and conclusions are generated from in vitro cell culture. However, many studies have demonstrated that cancer metabolism is different between in vitro and in vivo. In Fig. 7, in vivo studies show that FASNi inhibits KM lung tumor growth and xenograft tumor growth, which probably through ferroptosis based on in vitro cell culture study. If this is the case, can ferroptosis inhibitor rescue the tumor growth. Liproxstatin-1 can be used for in vivo animal study. In addition, please also examine the effect of FASNi +/- Liproxstatin-1 treatment in established KRAS WT tumor growth.

There are some other minor concerns:

1. In Fig. 1C-F and S4A, KM overexpression increases the lipid droplet accumulation and C16:0. Can this phenotype be inhibited by FASNi?
2. KM over-expression increases lipid droplet accumulation (Fig. 2D) in KRAS WT cells. Fig. S5D, FASNi inhibits lipid droplet accumulation in both KMLC cells and KRAS WT cells. Both H1395 and H1933 KRAS WT cells already have certain amount of lipid droplet accumulation. Therefore, does KM over-expression in those cells still promote de novo FA synthesis, further increase lipid droplet accumulation and cause them to be sensitive to FASNi? Is the accumulation of lipid droplets related to the sensitivity to FASNi?
3. It's better to provide a higher resolution of H&E in 7A right panel if possible, although it does not affect the results. In addition, Fig. 7B shows tumor number, not tumor burden quantification. Please correct it in figure legend or add tumor burden quantification.
4. Please correct grammar error in method: "We removed residual genomic DNA was removed with the Turbo DNA-free kit (AM1907, ThermoFisher)".
5. Fig. S7B, C, please add legend for red line and black line.
6. Please label cell name in Fig. 7H.

Reviewer #3 (Remarks to the Author); expert on lipid metabolism:

The manuscript by Bartolacci et. al. revealed that inhibition of FASN causes specific lipid remodeling in KRAS-mutant lung cancer and induces ferroptosis. Research in the recent year has demonstrated that

FASN is a promising target for several types of cancers, and FASN inhibition is currently in clinical trial for KRAS mutant cancer. These past studies provided background and scientific premise for the current work. This work contributes significantly to understanding the mechanism that KRAS mutant lung cancer is particularly sensitive to FASN inhibition, and identified the importance of Lands cycle. It is of great scientific interest. The paper utilized a variety of approaches and provided a large amount of data. I find it of high quality in general. Some questions and suggestions to improve the manuscript are detailed below.

1. In many lipidomic result figures (e.g. Fig 1F& G, Fig 7H), a large fraction of highlighted changes are in lipids containing odd- number fatty acyl chains. This is quite surprising, given human de novo FA synthesis is largely unable to produce odd chain. It is important to further validate the lipidomic identification and make sure the peaks are correctly assigned. If the odd-number lipids are real, it is important to discuss where do they come from (microbiome? Diet? Or from branch chain-coA) and why do they change significantly with FASNi.
2. The authors reported FASNi causes increase in AMPK activation, and suggest this may be due to inhibition of beta-oxidization. Why would FASNi cause inhibition of beta-oxidization? Is it because high malonyl-coA inhibits CPT? Some direct evidence that beta-oxidization rate is lower would strengthen the paper. It is also interesting that malonyl-coA increases even though ACC is inhibited upon FASNi treatment. And NADPH increases as well. Would increased NADPH increases the capacity for cells to defend against ROS? Some discussion about the seemingly counterintuitive observation between ACC inhibition and increased malonyl-coA, and increased NADPH and increased lipid oxidation would be helpful.
3. The authors showed that FASN inhibition increases AA uptake. Does it preferably increase the uptake of PUFA or increase fatty acid uptake across the board? How does the increase of fatty acid uptake change the PUFA/ SFA (or MUFA) ratio? Is the change dependent on the exogenous FA distribution or availability?
4. It is interesting that LPCAT3 is particularly increased and required for the survival of KMLC. There is evidence that LPCAT3 has substrate preference for transferring PUFA-coA to the sn-2 position of lipids. This seems to be opposite than expected if the main point is to control PUFA/SFA ratio in phospholipids in KMLC.
5. In Fig 1E, it is interesting that the spatial distribution of CE 22:6 is different than other PUFA containing glycerol-lipids. Discussion about possible reason would be good.

Minor

1. "HPLC" was mis-spelled as "HLPC" in a few places
2. Fig 5J, the authors used $^{13}C_2$ -acetate labeling to assay the FA synthesis and its incorporation into lipid. When acetyl-coA is labeled, FA will have a distribution of many different labeled forms. Why do the authors specifically use M+4/M+0 as an indicator? Same issue with Fig 5B, why specifically M+2/M+0? Is that because of elongation in this case? Providing the full isotopic distribution would be more appropriate.
3. How are the lipidomic data normalized? For instance in Fig 1A, is that normalized to tissue weight or normalized to median signal so it represent relative abundance?

4. In MALDI and BODIPY imaging results (e.g. Fig 1E, 7E), how is the boundary of tumor defined?

Reviewer #4 (Remarks to the Author); expert on ferroptosis, lipidomics and transcriptomics:

In this manuscript, Bartolacci et al. characterized how Kras-G12D and EGFR-L858R mutations affect the lipidome composition and lipid metabolic pathways in various lung cancer models. Using both untargeted lipidomic profiling and MALDI imaging-assisted spatial lipidomic analysis, the authors identified a relative increase in TAG and PC components specifically in Kras-mutant lung cancer (KMLC) models, in particular TAG/PCs that contain saturated or monounsaturated fatty acyl side chains. The authors demonstrated that these metabolic changes are induced by the upregulation of FASN expression and de novo lipogenesis activity. FASN expression is induced in a mutant Kras-dependent manner, and chemical inhibition of FASN specifically inhibits KMLC growth. The authors further characterized the lipidomic and transcriptional changes induced by FASN inhibition in KMLC cells. RNA-seq analysis showed that FASN inhibition induces the upregulation of genes involved in the Lands Cycle (a phospholipid fatty acyl side chain remodeling program). From the lipidomic analysis, the authors identified an increase in the relative abundances of lyso-PC and polyunsaturated PCs in response to FASN inhibition. The authors then showed that inhibiting either FASN or the Lands Cycle enzymes, leads to increased lipid peroxidation and elevated sensitivity to ferroptosis induction. Finally, the authors demonstrated that FASN inhibition is an effective strategy to limit Kras-mutant lung tumor growth both in animal models and in human patients.

Overall, this study is interesting and important for our improved understanding about the metabolic vulnerabilities of Kras-mutant tumors. The experiments are generally well-designed, and systematic approaches including lipidomics, transcriptomics, and functional screening are widely adopted to provide unbiased views on the questions been explored. Publication in Nature Communications is recommended given the following major points are properly addressed. In addition, the writing of the manuscript is hasty in many places, this referee has listed several issues to be addressed as minor points; but to improve the readability of this study, more scrutiny of the manuscript is recommended for the authors.

Major points

1. Page 18 line 11: the rationale for choosing FASN inhibitor to test KMLC dependency on de novo lipogenesis should be explained more clearly. The author might consider to move line 17 "As previously reported, we found that KM correlates with FASN..." to the beginning of this section. In addition, is this correlation significant?
2. Following the previous point, the author may want to comment on whether there is any other protein expression that correlates with KM in TetO-KrasG12D mice.

3. Though not a main focus on this study, the author may discuss potential mechanisms underlying the upregulation of SCD/LPCAT3/ACSL3/PLA2G4C expression in response to FASN inhibition in KMLC to improve the coherence of the logic.

4. A key assumption in this manuscript is that mutant Kras is inducing high levels of reactive oxygen species (ROS) and sensitizing cancer cells to ROS-induced cell death such as ferroptosis. It is under this prominent stress that the activity of FASN is important for producing enough saturated/monounsaturated fatty acids to repair the oxidized and damaged polyunsaturated phospholipids. How does KM induce ROS in lung cancer cells?

5. In Figure 1E, the MALDI imaging data does not truly support that increased TAG and PC contents are unique to Kras-mutant but not EGFR-mutant tumors. The authors may consider toning down their conclusion statement.

6. In Figure 2I-J: The authors observed a growth inhibitory effect after FASN inhibitor or KM inhibitor treatment alone, but not in the FASN inhibitor + KM inhibitor treatment condition in KMLC cell lines. These data suggest that FASN inhibitor can reverse the effect of KM inhibitor. Though these results are aligned with the logic of the epistatic relationship between mutant Kras and FASN, the overall result is quite surprising considering that the mutant-Kras-dependent cancer cells are deprived of both hyperactive Kras signaling and de novo lipogenesis, yet the cells are surviving much better than the single-agent treated cells. The authors should characterize whether the cells that survived dual mutant Kras/FASN inhibition remain tumorigenic in vivo (are these cells senescent or quiescent?). Moreover, what is the best treatment course to combine Kras inhibition and FASN inhibition for tumor suppression? Clarification of this issue is particularly important considering the clinical interest in combining mutant KRAS inhibitors and FASN inhibitors for cancer treatment.

7. The authors identified that LPCAT3 is specifically required for the survival of KMLC cells by participating in the Lands Cycle. LPCAT3, as described in this manuscript, however, is believed to be an enzyme that preferentially incorporates polyunsaturated fatty acyl- (PUFA-) CoA into lyso-PL to synthesize PUFA-containing phospholipids (thus promoting ferroptosis in cells, Dixon SJ et al ACS Chemical Biology, 2015, PMID: 25965523). This role is contradictory to the explanation in the manuscript in which LPCAT3 helps synthesized SFA/MUFA to get onto plasma membrane. Is there any other evidence supporting this claim, such as label-tracking or membrane lipidomic analysis following LPCAT3 inhibition? This experiment may also apply for other Lands Cycle related genes identified in the screening.

8. Page 19 line 20-21: FASNi affects FA synthesis in both KM and KRAS-WT. Have the authors compared the FASNi sensitivity between these cells? Since these cells differ in their lipidomes, it might be worth investigating to what extent this difference contributes to FASNi sensitivity.

9. In this study, all genetic perturbations are achieved using siRNAs and shRNAs, this referee is wondering that precluded the authors from using CRISPR/Cas9-mediated genome editing in parallel, considering that the effects of siRNA (which was used in several key experiments) can be transient?

10. In the abstract, the authors highlighted that “FASN inhibition promotes the intracellular accumulation of lipid peroxides and ferroptosis in KMLC”. While the major part of main Figure 6 demonstrates that FASN inhibition induces lipid peroxidation, but not direct cell death, Supplementary Figure 6B are the only data and contexts that FASN inhibition is sufficient to induce ferrostatin-1 rescuable cell death (in the absence of additional GPX4 inhibitors). Are the results in Supplementary Figure 6B applicable to other KMLC cell lines? Can these data be strengthened and presented in the main figure?

11. Comparing the results in Figure 4A and Figure 1A, in Figure 1A there is an increase in the total PC contents in the KRAS-mutant samples, whereas in Figure 4A there is no difference, what is the explanation for this discrepancy?

12. Page 26 line 19 in Discussion: the authors proposed that cell context specific factors contribute to the effect of KRAS on ferroptosis sensitivity. This is an interesting point that should be further expanded.

Minor points

Text:

1. The abstract is too long and should be revised to highlight the key findings.
2. P18-line 8, intratumor availability “of” SFA and MUFA
3. P17-line22, “then” should be “than”
4. Page 19 line 5: the sentence “KrasG12D inhibitor...reverses the effects...” is confusing. The author should revise this statement.
5. P19-line 14, depletion of lipid droplets is presented by Figure 3D not Figure 3G
6. P21-line 8, Fig not Fid
7. P21-line 5-8, these statements are unnecessary and are not fully supported by the data
8. P22-line 14: Fig. S6G and S6H do not exist. Are the authors referring to Figure S6E and S6F?
9. P42-line 12, HPLC has a typo?

Figures:

1. Figure 1A: The authors should describe the precise data normalization method used in processing the lipidomic analysis results? How is the relative level calculated?
2. Figure 1B: Need x Axis label
3. Figure 1E right panel, whether the second PC 18:1/20:4 was labeled wrong? Should it be labeled as PC 20:4/20:5 ?

4. Figure S1B and 7H: x-axis should be Log, not -Log.
5. Figure 1F-1G, Figure S1B: Is the y-axis P-value or adjusted P-value?
6. Figure 2B, is there a palmitate-only condition as a control, considering the diverse metabolic effects that palmitate treatment could exert on cells?
7. Figure 6H and 6I, did the authors confirm the knockout efficacy of the siRNAs for LPCAT3 and PLA2G4C by qPCR or western blot?
8. Figure 3A: Why are certain fatty acids analyzed using M+2/M+0 whereas others are using M+4/M+0? For FA 18:1n7, where are the analytes different for the same metabolite in two different cell lines?
9. Figures 4B,4D,4F,4H: Need x Axis label
10. Figure S7B: Need to add a legend or label what red and black line represent

We thank the reviewers for the thoughtful comments that we found extremely useful for the improvement of our manuscript. We addressed all the comments and we hope that the manuscript is now suitable for publication.

The revised manuscript has been modified according to the reviewers' suggestions. We reformatted and expanded several Figures, Supplementary Figures and we added new key experiments and data.

We also formatted the text to improve clarity.

Additional evidence, data and comments were included in the rebuttal letter because we think they were beyond the scope of our manuscript. However, we are available to include them in the manuscript should you deem it necessary.

For clarity, throughout the rebuttal text, Figures referring to the revised manuscript will be indicated in **bold blue**, and Figures referring to the rebuttal will be indicated in **bold black**.

All the changes introduced in the manuscript and supplementary information are indicated in **red**.

We hope that the manuscript is now suitable for publication.

Rebuttal_ Table of contents

Answer to Reviewer #1	page 1
Answer to Reviewer #2	page 11
Answer to Reviewer #3	page 15
Answer to Reviewer #4	page 28
References	page 36

Answer to reviewer #1

We thank the reviewer for the positive evaluation and useful comments. Below, we answered to their concerns point by point

Q1: The reviewer commented that MALDI-IMS is not a quantitative method.

A1: We agree that MALDI-IMS is a qualitative rather than a quantitative method. Quantitative analysis by MALDI-IMS is not straightforward, nor is it widely performed because of several factors: i) non-uniform crystallization of analytes and matrix; ii) baseline variability, iii) variability of signal intensity from laser shot to laser shot, sample to sample and run to run; iv) competitive ionization and/or ion suppression.

We would like to clarify that all the quantitative lipidomic data (Fig. 1a, b, c, f, g; Fig. 7h) were obtained with quantitative MS/MS analysis performed on laser-microdissected LC sections. Of note, for MALDI-IMS we used tissue sections that were contiguous to the ones analyzed by MS/MS. Being well-aware that MALDI-IMS, as it stands, is not sufficient to provide quantitative information for the analytes, as stated in lines 16-20 at page 5 of the manuscript, we used MALDI-IMS only to spatially resolve the distribution and the relative abundance of molecules of interest that we had identified by quantitative MS/MS analysis. Accordingly, we did not deem necessary to employ an internal standard correction strategy (such as the use of an isotopically labelled standard), which we will take into consideration for future studies.

Q2: The reviewer noted that the results in Fig. 1d are difficult to interpret due to lack of the corresponding H&E-stained images and their histological annotation.

A2: We agree with this suggestion. Accordingly, we provided H&E-stained sections, including their histological annotation, next to their corresponding MALDI-IMS images (Fig. 1d, Fig. 1e, Supplementary Fig. 1a).

Q3: The reviewer asked for more evidence of the enrichment in TAG, SM and PC in KMLC.

A3: To address this question, we reported the total spectra acquired and some of the lipid species we annotated as triacylglycerols (TAG), sphingomyelins (SM) and phosphatidylcholines (PC) in representative human and PDX lung cancer samples in Fig. 1 and 2, respectively. A complete list of all the annotated lipid species, with their molecular formula and ion adducts is provided in Supplementary Tables 1-4, and Supplementary Table 8

As typically observed in MALDI-IMS in positive mode, PC lipids were the dominant species in the range between m/z 678.5 [PC(30:0)+H]⁺ and m/z 872.56 [PC(40:6)+K]⁺ (Fig. 1B, 2B, Table 1, Supplementary Tables 1-4). For instance, the signal at m/z 772.52 with a mass error of 0.60 ppm corresponds to potassiated PC 32:0, *i.e.* dipalmitoylphosphatidylcholine (DPPC). Indeed, DPPC is the major surface-active component (40-70% of total PC) of the mammalian pulmonary lipidome. Protonated PC 32:0, at m/z 734.56, and its sodiated homologue, at m/z 756.55, were also detected with high mass accuracy. In addition to DPPC, di-saturated (30:0, 34:0) and mono-unsaturated (32:1, 34:1) PC, which are the main PC lipid species in mammalian pulmonary surfactant, were also observed as high peaks with three different charge carriers (protonated, sodiated, potassiated): PC 30:0 ([M+H]⁺, m/z 706.55; [M+Na]⁺, m/z 728.52; [M+K]⁺, m/z 744.50); PC 32:1 ([M+H]⁺, m/z 732.55, [M+Na]⁺, m/z 754.54, [M+K]⁺, m/z 770.50), PC 32:0 ([M+H]⁺, m/z 734.56, [M+Na]⁺, m/z 756.56, [M+K]⁺, m/z 772.52) PC 34:1 ([M+H]⁺, m/z 760.58, [M+Na]⁺, m/z 782.56, [M+K]⁺, m/z 798.55).

Fig. 1 and Fig. 2 are now provided as examples for the reviewer's eye only.

Figure 1. MALDI-IMS spectra of representative human Lung Cancer samples. Reported spectra compare wt KRAS (TH7037, green) and KMLC (L140, yellow.) **(A)** Whole spectra from m/z 50 to 2000 Da. **(B)** Spectra from m/z 700 to 850 Da are dominated by phosphatidylcholines (PC) species. **(C)** LysoPC (LPC) are detected in the range from m/z 460 to 580. **(D)** Triacylglyceride (TAG) adducts are found from m/z 845 to 947 Da. Note how PC 32:0, in the three ionization forms - protonated $[PC(32:0)+H]^+$, sodiated $[PC(32:0)+Na]^+$ and potassiumated $[PC(32:0)+K]^+$, and LysoPC16:0, present as $[LPC(16:0)+H]^+$, $[LPC(16:0)+Na]^+$ and $[LPC(16:0)+K]^+$ ions, are more abundant in KMLC (orange) than in KRAS-WT (green). Similarly, TAG, as $[TAG(50:2)+Na]^+$, are more abundant in KMLC. On the contrary, PC with polyunsaturated fatty acids (PUFA)-PC as PC 40:4 $[PC(40:4)+H]^+$ and PC 38:5 $[PC(38:5)+Na]^+$ are less abundant in KMLC. However, as total signal is the result of both tumor and stroma, we used region of interest (ROI) analysis to better compare tumor and stroma ROIs.

Figure 2. MALDI-IMS spectra of representative PDX Lung Cancer samples. Reported spectra compare wt KRAS LC (CP58391, green), KMLC (HCC_4059, yellow), and mutant EGFR LC (EGFR-MUT, HCC_4190, blue). **(A)** Whole spectra from m/z 50 to 2000 Da. **(B)** Spectra from m/z 700 to 850 Da with major PC species. **(C)** LysoPC (LPC) detected in the range from m/z 460 to 580 Da. **(D)** Triacylglycerides (TAG) between m/z 845 to 947. Note how PC 32:0 and PC 34:1, in the three ionization forms - protonated $[PC(34:1)+H]^+$, sodiated $[PC(34:1)+Na]^+$ and potassiumated $[PC(34:1)+K]^+$ - are more abundant in KMLC (orange) and EGFR-MUT (blue) than in wt KRAS (green). Similarly, LysoPC 16:0 is more abundant in KMLC. On the contrary, KMLC is the least rich in (PUFA)-PC as PC 40:4 $[PC(40:4)+K]^+$.

However, as total signal is the result of both tumor and stroma, we used region of interest (ROI) analysis to better compare tumor and stroma ROIs.

Table 1. List of [M+H]⁺, [M+Na]⁺, and [M+K]⁺ molecular species that have been identified. LPC: Lyso Phosphatidylcholines, PC: Phosphatidylcholines, SM: sphingomyelin.

Precursor m/z (observed)	Diagnostic fragment ions (m/z)	Lipid Annotation	Ion Adduct
496.34	104.10, 184.07, 478.32	LPC 16:0	[M+H] ⁺
518.32	104.10, 459.24, 146.982, 313.27	LPC 16:0	[M+Na] ⁺
534.29	104.10, 475.22	LPC 16:0	[M+K] ⁺
494.32	104.10, 184.12	LPC 16:1	[M+H] ⁺
516.30	457.23	LPC 16:1	[M+Na] ⁺
706.54	184.05	PC(30:0)	[M+H] ⁺
728.52	523.4729, 545.3448, 669.50	PC(30:0)	[M+Na] ⁺
744.49	685.55	PC(30:0)	[M+K] ⁺
730.54	184.12	PC(32:2)	[M+H] ⁺
732.55	184.12	PC(32:1)	[M+H] ⁺
754.54	695.47, 571.47, 549.48, 146.98	PC(32:1)	[M+Na] ⁺
770.51	711.44	PC(32:1)	[M+K] ⁺
734.57	184.12	PC(32:0)	[M+H] ⁺
756.55	551.48, 573.49, 697.48	PC(32:0)	[M+Na] ⁺
772.53	713.45	PC(32:0)	[M+K] ⁺
468.31	184.07	LPC(14:0)	[M+H] ⁺
520.34	184.07, 104.10	LPC(18:2)	[M+H] ⁺
542.32	146.98, 483.24	LPC(18:2)	[M+Na] ⁺
558.28	499.22	LPC(18:2)	[M+K] ⁺
522.36	504.34, 104.10, 184.07	LPC(18:1)	[M+H] ⁺
560.31	501.25	LPC(18:1)	[M+K] ⁺
544.34**	485.26, 146.98	LPC(18:1)	[M+Na] ⁺
544.34**	184.07, 526.33, 104.10	LPC(20:4)	[M+H] ⁺
746.60	184.07	PC(O-34:1)/PC(P-34:0)	[M+H] ⁺
703.57	184.07	SM(34:1)	[M+H] ⁺
725.56	666.48, 542.48, 146.98	SM(34:1)	[M+Na] ⁺
741.53	682.46	SM(34:1)	[M+K] ⁺
790.60	184.07, 772.60	PC(34:2)-OH	[M+H] ⁺

However, looking at the whole spectrum of the tissue section, we can not discriminate between the specific contribution of cancer cells (which we are interested in) from the surrounding stroma.

To address this issue, and to make the interpretation of the MALDI-IMS data easier, as asked by the reviewer, we exploited tools available in the MSiReader software¹ to delineate regions of interest (ROI) of cancer area and surrounding stroma to mimic laser microdissection analysis. Also, as explained in more detail below (in answers **A4**, **A5**), the MSiReader software allows for ROIs in a tissue section to be analyzed according to several parameters, including the number of pixels, average intensity, standard deviation of intensity, and median and quartile intensities.

Thus, to provide a “semi-quantitative” analysis, after total ion count (TIC) normalization, for each ROI, we have now normalized peak intensity by the area (pixel). Then, the following quantities were calculated for each image and ROI: the number of pixels, sum of intensities for all pixels, average intensity per mm², average intensity per pixel, standard deviation of intensity per pixel, relative standard deviation of intensity per pixel, median intensity per pixel, lower quartile (Q1) intensity per pixel, higher quartile (Q3) intensity per pixel, and the minimum and maximum intensities of each pixel. We clarified our analysis in the “methods” section of our manuscript (page 26 lines 18-23 and page 27 lines 1-2) and we included the data in **Supplementary Tables 1-4**.

Noteworthy, PC esterified with mono-unsaturated or saturated Fatty Acids (MUFA, SFA, respectively) were not only found to be very abundant, but mainly present within the tumor ROI of human KMLC and

TH7037 (KRAS-WT)

L140 (KM)

Figure 3. Tissue distribution of MUFA-PC, PUFA-PC and TAG in human LC.

(A) Distribution of representative PC lipids esterified with monounsaturated or saturated fatty acids (MUFA/SFA-PC) and their colocalization within wt KRAS and KM human LC (TH7037 and L140, at the top and bottom, respectively). **(B)** Representative PC lipids esterified with polyunsaturated fatty acids (PUFA-PC) and their colocalization. **(C)** Representative TAG lipids and their colocalization. Protonated PC 34:1, PC 32:1 and PC 30:0 [PC(34:1)+H]⁺, [PC(32:1)+H]⁺ and [PC(30:0)+H]⁺ were identified at *m/z* values 760.58, 732.55, and 706.54, respectively. Potassiated PC 38:4, potassiated PC 36:4 and sodiated PC 40:4 [PC(38:4)+K]⁺, [PC(36:4)+K]⁺ and [PC(40:4)+Na]⁺ were identified at *m/z* values 848.56, 820.52, and 860.62, respectively. Potassiated TAG 52:3, TAG 54:6 and TAG 55:2, [TAG(52:3)+K]⁺, [TAG(54:6)+K]⁺, [TAG(55:2)+K]⁺ were observed at *m/z* values 895.70, 917.70, and 939.70, respectively. Data are displayed as TIC-normalized. Corresponding H&E and histological annotation are shown. T, tumor; S, stroma; S+T, stroma and tumor mix. Scale bar: 2mm.

PDX lung cancer samples (L140 and HCC_4059 in **Fig. 3** and **Fig. 4**, respectively), while stroma and surrounding ROI are devoid of them.

CP58391 (KRAS-WT)

HCC-4059 (KM)

HCC-4190 (EGFR-MUT)

Figure 4. Tissue distribution of MUFA-PC, PUFA-PC and TAG in PDX LC samples. (A) Distribution of representative PC lipids esterified with monounsaturated or saturated fatty acids (MUFA/SFA-PC) and their colocalization within KRAS-WT, KM and EGFR-MUT LC (CP58391, HCC_4059 and HCC_4190, at the top, middle and bottom, respectively). **(B)** Representative PC lipids esterified with polyunsaturated fatty acids (PUFA-PC) and their colocalization. **(C)** Representative TAG lipids and their colocalization. Protonated PC 34:1, PC 32:1 and PC 30:0 [$[PC(34:1)+H]^+$], [$[PC(32:1)+H]^+$] and [$[PC(30:0)+H]^+$] were identified at m/z values 760.58, 732.55, and 706.54, respectively. Potassiated PC 38:4 and PC 36:4 and sodiated PC 40:4 [$[PC(38:4)+K]^+$], [$[PC(36:4)+K]^+$] and [$[PC(40:4)+Na]^+$] were identified at m/z values 848.56, 820.52, and 860.62, respectively. Sodiated TAG 50:1, TAG 54:6 and TAG 55:2, [$[TAG(52:3)+K]^+$], [$[TAG(54:6)+K]^+$], [$[TAG(55:2)+K]^+$] were observed at m/z values 895.70, 917.70, and 939.70, respectively. Data are displayed as TIC-normalized. Corresponding H&E and histological annotation are shown. T, tumor; S, stroma; Scale bar: 2mm.

In **Fig. 3A** and **4A** we show the ion distribution of representative PC esterified with MUFA ($[PC(34:1)+H]^+$, $[PC(32:1)+H]^+$), or SFA ($[PC(30:0)+H]^+$) and their colocalization within primary human and PDX lung cancer samples (**Fig. 3A** and **4A**). Note how in KRAS-WT and EGFR-MUT PDX samples (CP58391 and HCC-4190 in **Fig. 4**) MUFA and SFA-PC species are equally present in both the tumor and stroma ROI, while in KMLC (HCC-4059) they are exclusively present in tumor ROI.

By contrast, PUFA-PC, as PC 38:4, 36:4 and 40:4 ($[PC(38:4)+K]^+$, $[PC(36:4)+K]^+$, $[PC(40:4)+Na]^+$), respectively, were not detected in tumor ROI of KMLC (**Fig. 3B** and **4B**). Indeed, **Fig. 4B** shows that in KMLC PDX (HCC-4059) these PUFA-PC species are excluded from tumor ROI while being localized in stroma ROI. In contrast, in KRAS-WT and EGFR-MUT PDX samples (CP58391 and HCC-4190 in **Fig. 4**) PC 38:4, 36:4 and 40:4 could be detected in both tumor and stroma ROI.

Also, MALDI-IMS confirmed that TAG species mainly localize within tumor ROI in KMLC. **Fig. 3C** and **4C** show the spatial distribution of representative TAG species in human primary and PDX lung cancer samples. While TAG 52:3, 54:5 and 55:2 (as potassiated ions) are low-abundant in wt KRAS human LC (TH7037), they are present at higher levels in KMLC (L140), where they localize within tumor ROI.

Moreover, also in PDX samples (**Fig. 4**), representative TAG species TAG 50:1, 52:2, 56:5 are specifically present in the tumor ROI while being absent in the stroma ROI in KMLC sample (HCC_4059). On the contrary, in KRAS-WT and EGFR-MUTR PDX samples (CP58391 and HCC-4190 in **Fig. 4**) TAG are found in both tumor and stroma ROIs.

Figure 5. Normalized MALDI-IMS data of representative lipid species in human and PDX LC samples. Bars show the normalized intensity in tumor (red) or stroma (black) ROIs of representative ionized lipid species showed in Fig.3-5. For all the indicated ionized lipid species, the sum of ion intensity (a.u.) was normalized by Area (pixel). TH7037 (green) and L140 (orange) are wtKRAS and KM human LC; respectively. CP58391 (green), HCC-4059 (orange) and HCC-4190 (blue) are KRAS-WT, KM and EGFR-MUT PDX LC.

The values obtained by this “semi-quantitative” approach for representative data are listed in **Supplementary Tables 3-4** and plotted in **Fig.5**.

Figure 6. Tissue distribution of representative sphingomyelin (SM) in human and PDX LC samples. (A, B) Protonated SM 42:2, **(C, D)** SM 34:1, **(E, F)** SM 42:3 and **(G, H)** SM 40:1 ($[SM(42:2)+H]^+$, $[SM(34:1)+H]^+$, $[SM(42:3)+H]^+$ and $[SM(40:1)+H]^+$, respectively) were identified at m/z values 813.68, 703.57, 811.69, and 767.67; respectively. Data are displayed as TIC-normalized. T, tumor; S, stroma; S+T, stroma and tumor mix. Scale bar: 2mm.

In addition, we report some of the peaks annotated as SM (**Fig. 6**). In particular, here we show the tissue distribution of protonated SM 42:2 (**Fig. 6A, B**), SM 34:1 (**Fig. 6C, D**), SM 42:3 (**Fig. 6E, F**), and SM 40:1 (**Fig. 6G, H**) in human primary and PDX LC samples. These MALDI-IMS data show how, consistently with the quantitative data obtained from MS/MS lipidomics, SM are mainly found in KMLC tumor areas, while being distributed throughout the lung tissue in KRAS-WT and EGFR-MUT samples.

Some of the relevant species shown in **Fig. 3, 4, 6** are also reported in **Fig. 1e** of the manuscript.

Q4: The reviewer asked for additional details regarding our MALDI-IMS protocols and asked for details about MSI data acquisition and their interpretation.

A4: We provided a more comprehensive description of our approach for MALDI-IMS in the “methods” section of our manuscript (page 26 lines 18-23 and page 27 lines 1-2) and we included the data in Supplementary Tables 1-4. Below, we provide additional details of our analytic approach for the reviewer. First, as metabolic changes can occur rapidly once tissues are harvested, thus altering the lipidomic profile, lung samples were immediately snap-frozen and stored at -80°C until analysis.

Second, we developed a technique to cryo-dissect 10 μm thick tissue sections without using any embedding material. This choice is motivated by the fact that OCT, the most common embedding matrix for cryostat sectioning, contains benzalkonium salts which might form adducts with endogenous lipid species, as polyethylene glycols and polyvinyl alcohol primers which often cause ion suppression and string background signals. This approach provides tissue sections of high quality and of homogeneous thickness that allows us to comprehensively “map” the lipidome without the risk of incurring in artifactual chemical modifications².

In addition to the quality of the tissue sections, matrix deposition is a critical parameter for MSI-IMS. In order to compare signal intensities of analytes in different sections, it is necessary to obtain an equal concentration and homogenous deposition of matrix on tissues being compared. Hence, we applied matrices simultaneously to the tissue sections that were to be compared with equalized analyte extraction and co-crystallization conditions (**Fig. 7**). Here, we provide a representative example of the validity of our procedure to process samples by demonstrating the homogenous distribution of 2,5-Dihydroxybenzoic acid (DHB) matrix clusters, detected as [6DHB-4H₂O+NH₄]⁺ (*m/z* 870.15; **Fig. 7A, B**), [3DHB-2H₂O]⁺ (*m/z* 426.15; **Fig. 7C, D**), [5DHB-3H₂O+K]⁺ (*m/z* 755; **Fig. 7E, F**) within tissue sections of representative primary human (TH7037 and L140) and patient derived xenograft (PDX) (HCC-4059, HCC-4190, CP58391) LC samples.

Figure 7. Distribution of matrix cluster signals in lung cancer tissue sections. Images show the even distribution of 2,5-Dihydroxybenzoic acid (DHB) matrix cluster in representative human (**A, C, E**) and PDX (**B, D, F**) specimens. Selective region of interest (ROI)- tumor (T), or stroma (S)- are indicated. (**G**) and (**H**) show the TIC distribution used to normalize signals. Corresponding histological annotation are shown. T. tumor; S. stroma.

After optimization of tissue sectioning, lipids were characterized by scanning lung tissue sections in the mass range *m/z* 50-2000 Da and spatial resolution (60 μm). Signal-to-noise ratio (SNR) for peak detection was set 3.0 and set peak group detection within 0.5 Da.

AP-SMALDI MSI data sets (.raw) were converted to centroid imzML files, lipid ion MS images were generated using the open source software “MSiReader” version 2.0.1.14¹ with a *m/z* bin width of Δ *m/z* = ±5 ppm. No further (pre or post) data processing steps such as baseline correction, noise removal, smoothing were applied for image generation in order to demonstrate the original data quality.

We adopted a spectrum-normalization approach based on total ion counts (TIC) and all the MALDI-IMS data were displayed as TIC-normalized. Distribution of normalization is showed in **Fig. 7G, H**.

Thus, all mass spectra were divided by their TIC so that all spectra have the same integrated area under the spectrum. This normalization approach assumes that there are comparable numbers of signals present in each spectrum. Hence, since we are comparing similar tissue sections, undergoing the same preparation, and being acquired with the same conditions we reason that TIC normalization can improve the ability to compare amount level of analytes across the tissue sections.

After TIC normalization, lipid annotation was done based on specific diagnostic fragment ions of head groups or neutral losses³, or by matching with tandem MS spectra of standard lipid species available in databases, as LIPID MAPS (www.lipidmaps.org), METLIN (www.metlin.scripps.edu) and Human Metabolite Database (www.hmdb.ca), within $\Delta m/z = \pm 2$ ppm or 0.009 Da as reported in **Table 1** and **Supplementary Table 1**.

Q5: The reviewer asked for clarifications about the resolution of the MALDI-IMS system we utilized.

A5: For data acquisition we used the Synapt G2 (MALDI QTOF). The mass resolution specification for the instrument is 20000 at m/z 180 which translates to ± 0.009 Da.

For data analysis we used the open source MSiReader software¹, which is commonly used in the field. As shown in a representative screenshot of the software interface, MSiReader provides m/z values expressed with three decimal digits (**Fig.8**).

However, we agree that this level of resolution is in the means of variability and have approximated m/z values to two digits

Figure 8. The graphical user interface of the msiQuant software. (A-D) A screenshot of the interface as reported by the software manufactures. (A) The project view; (B) the spectra view displaying the average and the maximum intensity spectra of the image; (C) the mass list view with the selected ions; (D) the image view displaying the distribution of the selected ion, the normalization factor, and the concentration levels. **(E-G)** A representative screenshot of the interface. (E) The project view, (F) the spectra view and the (G) image view.

Q5: The reviewer noted that “apoptotic/necrotic and perhaps ferroptotic processes result in altered Na/K ratios that can cause local efficient cationization of specific lipid species in MSI”, which in turn may result in spot-to-spot variance of signal intensities.

A5: We are cognizant of this issue. To address it, we have relied on the assistance of a trained pathologists (Dr. Hodges and Dr. Wistuba) to perform our analysis on viable tissue. Also, we have provided H&E stained sections-refer to **Fig. 1d, Fig.1e, Supplementary Fig. 1a**

As outlined by the reviewer, salts might interfere with the matrix-analyte crystallization process leading to the development of heterogeneous crystals, which in turn results in spot-to-spot variance of signal intensities. To address this issue and achieve unambiguous lipid annotation we: i) verified the homogenous distribution of DHB matrix clusters as explained previously (**Fig. 7**); ii) used TIC-based normalization to ensure that all the spectra in the data set have the same integrated area under the spectrum; iii) verified

that major cation adducts (protonated, sodiated and potassiated) of candidate lipids have similar tissue distribution.

To further clarify this point, here we provide the reviewer with an example of PC, as they commonly undergo cationization with alkalis to form metal-adduct molecules⁴ and are the main substrate of the Lands cycle and ferroptosis- the main focus of our manuscript.

The PC polar head group is a quaternary ammonium ion and thus always ionized. However, as pointed by the reviewer, because tissue sections are rich in sodium and potassium salts, alkali-metal adduct phospholipids are generated along with protonated molecules (Table 1, Supplementary Tables 1-4). Since multiple ions can form from a single species, we agree that the distribution image of a given PC might not reflect the actual distribution of that PC, but rather the heterogeneous distribution of its adducts. For instance, a protonated PC 36:4 molecule is detected as having the same mass as a sodiated PC 34:1 ion at m/z 782 (Table 1, Supplementary Tables 1-4). To overcome this problem, we compared spatial distribution of different cation adducts. For instance, the two types of PC detected at m/z 782 described above could be separated by looking at distribution of m/z 820.58 and m/z 760.58 (Fig. 9).

Figure 9. MALDI-IMS images for m/z 782. For unambiguous annotation of the peak observed at m/z 782.56 in human (A) and PDX (B) lung cancer samples we compared the ion distribution with that observed at m/z 760.58 (C, D) and at m/z 820.58 (E, F), corresponding to protonated PC 34:1 [PC(34:1)+H]⁺, and potassiated PC 36:4 [PC(36:4)+K]⁺, respectively. Data are displayed as TIC-normalized. T, tumor; S, stroma. Scale bar: 2mm.

Q6: The reviewer asked which ion types or m/z values have been observed. In addition the reviewer asked to provide MSI data, its annotation and interpretation in the supplementary data section.

A6: We performed lipid annotation using either specific diagnostic fragment ions of head groups or neutral losses^{3,5}, or by comparing the spectra we obtained with the tandem MS spectra of standard lipid species available in databases, as LIPID MAPS (www.lipidmaps.org), METLIN (www.metlin.scripps.edu) and Human Metabolite Database (www.hmdb.ca) within $\Delta m/z = \pm 2$ ppm or 0.009 Da.

We amended the “methods” section of our manuscript (page 26 lines 18-23 and page 27 lines 1-2) and we included the data in Supplementary Tables 1-4 and Supplementary Table 8.

Answer to reviewer #2

Q1: The reviewer asked whether the ferroptosis inhibitors Liproxstatin-1 rescues the FASNi effects on tumor growth *in vivo*.

A1: We thank the reviewer for the positive evaluation of our work and for making constructive comments. We demonstrated that FASNi induces ferroptosis in KMLC *in vitro* causing: (i) a specific accumulation of PUFA- PC and PUFA-LysoPC (Fig. 4); (ii) lipid peroxidation and (iii) cell death that can be rescued by ferroptosis inhibitors (Fig.6, Supplementary Fig. 7).

Figure 10. FASNi induces ferroptosis in KMLC *in vivo*. (A, B) Representative H&E pictures of the lungs of TetO-Kras mice treated as indicated and tumor number quantification. n=number of mice. (C, D) *In vivo* growth curves and post-resection pictures of H460 and A549 xenografts in NOD/SCID mice treated as indicated. (E-G) Representative pictures of C11-BODIPY staining of the lungs of TetO-Kras and of H460 xenografts, and their quantification. Dotted circles in (E) indicate lung tumor. (H) Volcano plot showing lipid species identified by MS/MS differentially represented in FASNi versus vehicle (n=5 mice/group). P values and difference were calculated using multiple t tests (p<0.05). (I-J) *In vivo* growth curves and post-resection pictures of A549 xenografts in NOD/SCID mice treated as indicated. Color-coded statistics is comparison versus vehicle. Lip-1, Liproxstatin-1. In (B, G) unpaired two-tailed student t test, in (C) multiple t tests and in (I) two-way ANOVA plus Sidak's comparisons with ns, p>0.05; *p<0.05; **p<0.01; ***p<0.001; ****p<0.0001.

We also demonstrated that FASNi induces ferroptosis *in vivo* because: (i) it inhibits tumor growth in KMLC GEMM and xenograft models (**Fig. 7a-d**); (ii) causes lipid peroxidation in autochthonous tumors and xenografts (**Fig. 7e-g**, and **Fig. 10**); (iii) promotes the accumulation of PUFA-PL in tumors (**Fig. 7h** and **Fig. 10**).

As suggested by reviewer #2, to ultimately demonstrate that ferroptosis is the mechanism of action of FASNi, we performed an *in vivo* rescue experiment using A549 (KMLC) xenografts and the ferroptosis inhibitor Liproxstatin-1 (Lip-1) (included in **Fig. 7i-j** and **Supplementary Fig. 10f** of the revised manuscript).

As shown in **Fig. 10**, Lip-1 administration completely rescued the anti-tumor effect of FASNi, confirming that ferroptosis is the mechanism of action *in vivo* as well. We included these results at page 14, lines 8-10 of the revised manuscript. The correspondent methods section has been inserted at page 19, lines 22-23 and page 20, lines 1-3.

Following the suggestion of reviewer #2, we also performed an additional *in vivo* experiment with KRAS-WT human lung cancer cell line H522. As expected from our *in vitro* data, FASNi did not induce any anti-tumor effect on H522 xenografts (**Fig. 11**). Given the absence of an anti-tumor effect, we reasoned it was unnecessary to perform a rescue experiment and we decided not to include these data in the manuscript. We are available to include these data in supplementary information upon request.

Importantly, we did not observe any overt toxicities during the experiments (**Fig.12** and **Supplementary Fig. 10f**).

Figure 11. KRAS-WT LC is resistant to FASNi treatment *in vivo*. (A) *In vivo* growth curves and (B) post-resection pictures of H522 xenografts (KRAS-WT) in NOD/SCID mice treated as indicated. Multiple t tests with ns, $p > 0.05$.

Figure 12 (A) Body weight of mice A549 xenografts (KM) and **(B)** body weight of mice bearing H522 xenografts (KRAS-WT) treated as indicated. Multiple t tests with ns, $p > 0.05$.

Q2: The reviewer asked whether KM over-expression in H1395 and H1933 KRAS-WT cells still promotes *de novo* FA synthesis, further increases lipid droplet accumulation, and causes them to be sensitive to FASNi.

A2: KM overexpression induces accumulation of lipid droplets in H522, H661 and H1993 KRAS-WT cells (Fig. 2d, Supplementary Fig. 4a and Fig. 13A). FASNi depletes lipid droplets also in KRAS-WT cells (Fig. 3d) and in KRAS-WT cells expressing KM (Fig. 13A). Upon KM expression, H522, H661 and H1993 also become sensitive to FASNi (Fig. 2f, Supplementary Fig. 4b, c e Fig. 13B). This is consistent with KM cell lines displaying significantly lower FASNi IC₅₀ values than KRAS-WT cells (Fig. 13C and included in Supplementary Fig. 3c of the revised manuscript).

To test whether lipid droplets abundance correlates with sensitivity to FASNi, we plotted the FASNi IC₅₀ values *versus* the correspondent Oil Red O (ORO) absorbance values in a principal component analysis (PCA) graph. As shown in Fig. 13D, KM and KRAS-WT cell lines cluster in two distinct groups, which are mainly separated according to the FASNi IC₅₀ (Y axis), but not according to the ORO OD₅₁₀ (X axis), indicating that lipid droplet abundance is not discriminating between the two groups (included in Supplementary Fig. 5e of the revised manuscript). In addition, when we performed a Pearson Correlation analysis on both groups of cell lines individually, even though we observe a trend of inverse correlation between FASNi sensitivity and amount of lipid droplets in both cases, such correlation is not statistically significant (Fig. 13E, included in Supplementary Fig. 5f, g of the revised manuscript).

These data, along with the fact that FASNi depletes lipid droplets in both KM and KRAS-WT LC cell lines (Fig. 3d, Supplementary Fig. 4a and Fig. 13A), but it causes cell death and ferroptosis only in KM cells, indicate that lipid droplets do not determine the dependency to FASNi.

Thus, we reason that lipid droplets can be used as a readout of fatty acid synthesis and its inhibition (just like the other readouts, such as TAG depletion, DAG accumulation, or inhibition of beta-oxidation reported in Fig. 3), but it does not have a causal role in establishing FASNi sensitivity. We included these considerations at page 6 lines 14-16 of the revised manuscript.

Q3: The reviewer asked to provide a higher resolution of H&E in Fig. 7a right panel. In addition, the reviewer noted a mistake in the legend of Fig. 7b.

A3: We agree with the reviewer that those images did not have the necessary resolution. This was due to the PDF conversion process. We ensure to upload a high resolution image with the revised manuscript. We also edited the figure legend.

Q4: Please correct grammar error in method: “We removed residual genomic DNA was removed with the Turbo DNA-free kit (AM1907, ThermoFisher)”.

Q5: Supplementary Fig. 7b, c, (Supplementary Fig. 9b, c in the revised manuscript) please add legend for red line and black line.

Q6: Please label cell name in Fig. 6h.

A4-6: We made all requested corrections.

Answer to reviewer #3

Q1: The reviewer noted that “in many of our lipidomic result figures, a large fraction of highlighted changes is in lipids containing odd- number fatty acyl chains (e.g. Fig. 1f g, Fig. 7h), which appears to be “quite surprising, given human *de novo* FA synthesis is largely unable to produce odd chain”, and asked for a discussion of our results. Also, the reviewer encouraged to discuss where odd-number lipids come from (e.g. microbiome, diet, or branch chain-coA) and why they change significantly with FASNi.

A1: We thank the reviewer for the positive evaluation of our manuscript and for asking this interesting question. As the reviewer noted, *de novo* FA synthesis in humans and mice mainly produces even-number FA by sequentially adding a 2-carbon unit (acetyl-CoA) derived from malonyl-CoA, and historically, odd chain fatty acids have been used as internal standards in MS-based lipidomic methods, as they represent only about 1% of the total plasma FA.

Fig 14 KM expression and FASNi equally affect odd- and even-chain lipids. Fold change comparison between odd- and even-chain lipids identified in the lipidomic data of human PDXs (KM/KRAS-WT) (A) and of A549 xenografts (FASNi/vehicle) (B). P values calculated using unpaired student t test $p > 0.05$, not significant.

influence the level of circulating odd-chain FA, both dietary and not dietary sources, as well as alternative biosynthetic/metabolic pathways (including the α -oxidation of branched-chain FA) may play a role⁸⁻¹².

We discussed this point at page 17, lines 15-18 of the revised manuscript.

Even though our study indicates that *de novo* lipogenesis feeds the Lands cycle with newly synthesized even-chain FA, we have to keep in mind that remodeling of phospholipids can target preexistent species, such as PC, which may contain also odd-chain FA derived from exogenous sources. Once inside the cell, such lipids can be elongated⁶, desaturated⁷ or degraded via oxidation. Accordingly, taking into account the lipid species with adj P<0.05 reported in both Fig.1f and Fig.7h of the manuscript, we did not find a significant difference in fold change between species containing odd- or even-chain FA (Fig. 14). Therefore, we conclude that FASNi causes changes also in lipid species containing odd-chain FA by interfering with their remodeling, as much as it does with the species containing even-chain FA.

The question regarding the possible source of odd-chain FA is very interesting, too. However, it is very difficult to determine the provenance of odd-chain fatty acids in our setting. We think that this investigation is beyond the scope of our study, and it would require a more detailed analysis using specific tracers. We can only speculate that the options suggested by the reviewer are reasonable and possible. In this regard, recent studies using germ-free mice, and dietary response in rats and mouse models, indicate that, while microbiome does not

Q2: The reviewer asked why FASNi would cause inhibition of beta-oxidization and wondered if it is because high malonyl-coA inhibits CPT1. Moreover, the reviewer asked for some more direct evidence that β -oxidization rate is lower upon FASNi treatment. Also, the reviewer asked for some discussion about the seemingly counterintuitive observation about ACC inhibition, increased malonyl-coA, increased NADPH and increased lipid oxidation.

A2: We thank the reviewer for the possibility to better explain our results.

We found that, in addition to inhibiting *de novo* fatty acid synthesis, FASNi leads to a rapid and profound increase in malonyl-CoA (**Fig. 3b**), the predominant substrate for FASN. Consistently, others reported that both C75 and Cerulenin (2,3-epoxy-4-oxo-6-dodecadienoylamide), two potent FASN inhibitors, induce a similarly rapid increase in levels of malonyl-CoA^{13,14}.

In addition to its role as a substrate for FASN, malonyl-CoA is pivotally required for energy regulation through its reversible inhibition of carnitine palmitoyltransferase-1 (CPT1), the enzyme which controls fatty acid entry into mitochondria for β -oxidation. In physiologic conditions, during lipogenesis, high steady-state levels of malonyl-CoA inhibit CPT1 preventing mitochondrial oxidation of newly synthesized fatty acids. During starvation, a high fat diet, or TOFA (a reversible inhibitor of Acetyl CoA carboxylase) treatment, *de novo* fatty acid synthesis is reduced, malonyl-CoA levels fall, and CPT1 is activated allowing entry of fatty acid into the mitochondria for oxidation.

However, according to data reported by others, and consistent with our findings, FASN inhibition leads to the non-physiologic metabolic state of concomitant inhibition of fatty acid synthesis and fatty acid oxidation as a consequence of increased malonyl-CoA¹⁵. Indeed, previous observations have already showed that blocking FA synthesis results in lower oxygen consumption¹⁶ and in a drop of fully labeled citrate and mitochondrial potential to a similar extent as blocking FA oxidation¹⁷. Notably, such changes are characteristic effects of FASN inhibitors, rather than being secondary effects of other cellular responses, including apoptosis, as FA oxidation inhibition occurs during FA synthesis inhibition, but before the onset of cytotoxicity¹⁸. Mechanistically this effect is ascribed to the fact that increased malonyl-CoA levels inhibit CPT1, preventing β -oxidation¹⁵⁻¹⁷.

Thus, even though the FASN inhibitor TVB-3664 does not bind directly to CPT1, it increases cellular malonyl-CoA, and, therefore, it is expected to decrease CPT1 activity indirectly, preventing the oxidation of fatty acids^{19,20}.

However, as stressed in the manuscript, FASNi increases malonyl-CoA and decreases FA oxidation (FAO) (**Fig. 3b** and **Fig. 3e**) in both FASNi sensitive and resistant cells (KM and KRAS-WT LC cells, respectively) indicating that this is not the primary mechanism inducing cytotoxicity during FASN inhibition.

To provide more evidence that FASN inhibition decreases β -oxidation, we investigated the bioenergetic profile of representative KM and KRAS-WT LC cell lines upon treatment with FASNi (**Fig. 15**). Among the methods currently used to study cellular FAO activities, one strategy is to determine oxygen consumption rates (OCR). We used the Seahorse XF extracellular flux analyzer to measure OCR²¹.

This technique is capable of assessing mitochondrial function in terms of several bioenergetic parameters including basal respiration, ATP production, proton leak, maximal respiration, spare respiratory capacity and non-mitochondrial respiration in a highly sensitive manner. For the interpretation of the bioenergetic profile, we referred to²².

Figure 15. Bioenergetic profiles obtained during the mitochondrial respiration assay using the Seahorse XF Mito Stress Kit. Parameters of mitochondrial respiration were derived by the sequential addition of pharmacological agents to respiring cells, after 4 day-treatment with either the vehicle (black) or FASNi (red). For each parameter, three oxygen consumption rate (OCR) measurements were made over 18 minutes. After baseline OCR measurement, oligomycin (1 μ M), carbonyl cyanide-p-trifluoromethoxyphenyl-hydrazone (FCCP, 1 μ M), antimycin A, and rotenone (2 μ M), were sequentially added. OCR was calculated by the Seahorse XF96 software package. KM (H2122, H460, A549) and KRAS-WT (H522, H1993) LC cell lines were assayed. Data are shown as median values \pm SD. n=8 replicates.

With this approach we demonstrated that basal OCR, proton leak, ATP-linked OCR, the maximal OCR and the SRC capacity were all significantly lower in all cells tested, independently of FASNi sensitivity/KM status (**Fig. 16**). This reflects a reduced ability of FASNi-treated cells to use membrane potential, which serves as driving force for ATP synthesis. Consistently with the notion that maximal respiration depends on the supply of fatty acyl CoA as substrate, the maximal OCR was drastically reduced in all LC cells, after FASNi (**Fig. 16**).

Figure 16. Calculated values for mitochondrial respiratory parameters. Basal: basal respiration; ATP: ATP-linked respiration; Capacity: maximal respiratory capacity; Spare: spare respiratory capacity; Non Mito: non-mitochondrial respiration; Proton: Proton leak. FASNi-treated cells are shown in red. KM (H2122, H460, A549) and KRAS-WT (H522, H1993) LC cell lines. Statistical significance calculated using unpaired student-test (****<0.0001, ***<0.0001).

FA are well-known to be a major source for reducing equivalents, such as NADH and FADH₂, and electrons for ATP production through oxidative phosphorylation (OxPhos). On the other hand, FA oxidation is a major system for electron supply. Consistently, we also detected a significant decrease in OxPhos, RCR, and coupling efficiency (*i.e.* the percentage of respiration used for ATP synthesis) in cells treated with FASNi (**Fig. 17**), indicating that FASN inhibition (hence palmitate depletion) affects the efficiency of mitochondrial respiration²³.

Regarding NADPH, it primarily serves to provide reducing equivalents for FASN enzymatic activity (7 NADPH molecules are consumed per fatty acid synthesized). On the other hand, FA oxidation pathway provides NADPH indirectly: at each FA oxidation round, NADH, FADH₂, and acetyl coenzyme A (CoA) are generated²⁴. NADH and FADH₂ enter the electron transport chain (ETC) while the acetyl CoA enters the TCA cycle to produce citrate, which is then exported to the cytosol to engage in NADPH production²⁵.

Thus, inhibiting FASN should lead to an accumulation of NADPH, while inhibition of FA oxidation should lead to an indirect increase in NADPH. Since both FA synthesis and oxidation coexist, and both are inhibited by FASNi, *we reason that, from a stoichiometric point of view, inhibiting FASN is predicted to affect NADPH to a greater extent than FA oxidation inhibition. For instance, 16 moles of NADPH are necessary to condensate 8 moles of malonyl-CoA and 1 mole of acetyl-CoA to synthesize one mole of C18-saturated fatty acyl-CoA (stearoyl-CoA). On the other hand, only 9 moles of NADPH can be generated by the pyruvate-oxaloacetate-malate (POM) cycle during FA oxidation²⁶.*

We did not include these results and considerations in the revised manuscript, but we are available to include them if deemed necessary by the reviewers.

Q3: The reviewer asked whether FASNi preferably increases the uptake of PUFA or increases fatty acid uptake across the board, whether the increase of fatty acid uptake changes the PUFA/ SFA (or MUFA) ratio and whether any change is dependent on exogenous FA distribution or availability.

A3: We thank the reviewer for this important question, as in our manuscript we propose that the lung microenvironment rich in PUFA, surrounding KMLC, contributes to the execution of ferroptosis upon FASNi (**Fig. 1d, e** and **Fig. 5c, d**). To answer the reviewer's question, we developed an *in vitro* assay of single-choice and competitive uptake of PUFA/SFA (**Fig. 18A** and **Supplementary Fig. 6** of the revised manuscript). As PUFA proxy we used the AA-alkyne conjugated to Alexa Fluor 594 -azide (red) via Click-iT chemistry. As SFA representative, we used the C16-BODIPY (green, Thermo Scientific). To briefly explain, we treated H460 cells, as a representative human KMLC cell line, with either vehicle (0.2% DMSO) or FASNi (0.2 μ M) for 4 days. Each treatment group was randomized in 10 subgroups receiving the following FA mixtures in RPMI with 2% ultra-fatty acid free BSA (SIGMA ALDRICH, A6003-25G) for 6 hrs:

- 1) 20 μ M AA;
- 2) 10 μ M AA;
- 3) 20 μ M C16;
- 4) 10 μ M C16;
- 5) 5 μ M AA + 5 μ M C16 (AA:C16 ratio 1:1);
- 6) 10 μ M AA + 10 μ M C16 (AA:C16 ratio 1:1);
- 7) 16 μ M AA + 4 μ M C16 (AA:C16 ratio 4:1);
- 8) 15 μ M AA + 5 μ M C16 (AA:C16 ratio 3:1);
- 9) 4 μ M AA + 16 μ M C16 (AA:C16 ratio 1:4);
- 10) 5 μ M AA + 15 μ M C16 (AA:C16 ratio 1:3)

We observed that FASNi increases the total FA uptake in H460 KMLC cells with respect to the vehicle counterparts. In particular, in the single-choice settings, we noticed that: 1) the uptake of either AA or C16 is dose-dependent; 2) availability of C16 induces a greater total FA uptake than the same amount of AA in both groups treated with vehicle and FASNi. These two observations are consistent with our hypothesis that the availability of exogenous PUFA/SFA dictates their uptake and that KMLC might prefer C16 over AA in single-choice conditions (**Fig. 18B, C** and **Supplementary Fig. 6** of the revised manuscript).

When we analyzed the percentage (%) uptake of C16 (green) and AA (red) in cells subjected to the various FA mixtures, we found that: 1) in all conditions, C16 accounts for the majority of the total FA uptake (green bar); 2) FASNi modestly increases the uptake of AA when cells are incubated with an AA:C16 ratio between 1:1-4:1; 3) to produce a comparable % uptake of AA and C16, AA:C16 ratio must be between 4:1 and 3:1; 4) when C16 is predominant (AA:C16 ratio 1:3-1:4), FASNi does not increase the uptake of AA (**Fig. 18D** and **Supplementary Fig. 6** of the revised manuscript).

All in all, these results further support our conclusion that: 1) the extracellular availability of PUFA/SFA determines their uptake and the susceptibility to ferroptosis of KMLC cells; 2) KMLC prefers SFA over PUFA; 3) to produce a comparable C16 and AA uptake, PUFA must be present in large excess; 4) excess of exogenous SFA impedes AA uptake, in line with the fact that exogenous palmitate can rescue FASNi phenotype in KMLC.

We included these data, methods and conclusions in the revised manuscript (Page 9, lines 15-23; page 10, lines 1-12; Supplementary information page 5, lines 11-22 and page 6, lines 1-9; **Supplementary Fig. 6**).

Figure 18. Extracellular availability of PUFA/SFA dictates their uptake by KMLC cells. (A, B) Schematic and representative images of the FA uptake assay. AA, Arachidonic acid. C16, palmitate. (C) Quantification of the total FA uptake by H460 KMLC cells under the indicated conditions. Each dot represents one cell. (D) Relative quantification of AA (red) and C16 (green) uptake by H460 cells under the indicated conditions. Bars represent % of total mean fluorescence intensity signal (BODIPY + Alexa 594). In (C) multiple t test with ns, $p > 0.05$; * $p < 0.05$; ** $p < 0.01$; *** $p < 0.001$; **** $p < 0.0001$.

Q4: The reviewer commented on the role of LPCAT3 in regulating PUFA/SFA ratio of phospholipids in KMLC, since LPCAT3 has been so far reported to prefer PUFA-coA as substrate.

A4: We thank the reviewer for this question (please refer also to **A7** to **Q7** of Reviewer #4). We are aware that our data are in contrast with most of the literature currently available about LPCAT3 activity. Indeed, LPCAT3 is largely reported to have preference for arachidonic acid, AA (20:4). However, most of the data about LPCAT3 enzymatic activity have been obtained in intestine and liver, where it is quite abundant, either *in vivo* or in human and mouse hepatoma cell lines^{27–31}.

Although the initial enzymatic characterization of the Lands Cycle occurred almost 50 years ago, little is known about its role *in vivo*. To date, most studies on phospholipids (PL) fatty acyl composition have utilized *in vitro* biochemical assays, due to the difficulty of detecting specific changes in membrane composition in living cells. Therefore, there is little understanding of how regulatory pathways control PL fatty acyl composition and how such regulatory pathways could dictate biological responses *in vivo*.

Lysophospholipid Acyltransferase (LPAT) activity assays are generally performed exposing cell extracts containing the enzyme of interest to a pair of pure substrates (one lysophospholipid and an acyl-CoA ester) and measuring the conversion of the substrate into the product.

To test whether, at least in the context of KMLC, LPCAT3 re-acylates LysoPC with palmitate-CoA (16:0-CoA), we adapted an established biochemical assay³² (included in **Supplementary Fig. 8** of the revised manuscript), taking advantage of A549 KMLC cells stably expressing a doxycycline (doxy)-inducible *LPCAT3* shRNA (already used in **Fig. 5j, k** and **Supplementary Fig. 7**). The assay consists in incubating a mixture of acyl-CoAs and LysoPC with microsomal extracts (the fraction enriched in LPCAT3), and then analyzing specific PC species by LC-MS/MS (**Fig. 19A**).

First, microsomes were prepared from A549 shLPCAT3, either kept in doxy or no doxy for 48 hours (**Fig. 19A, B**). Second, the fatty acyl-CoA esters chosen were the ones relevant to our question, hence, 20:4-CoA and 16:0-CoA. Third, as previously described³², we used 1-(10Z)heptadecenoyl-2-hydroxylysophosphatidylcholine (17:1-LysoPC) containing *sn-1* acyl chain 17:1, which reduces the background signal from endogenous phospholipids present in the microsomal preparations.

To identify PC we used the acetate adduct [M+CH₃COO]⁻ ions and data were analyzed with the in-house script software LipPy. Evaluation of the enzymatic reaction was performed by calculating the ratio of the integrated area of PC to that of the corresponding internal standard. The absolute amount of microsomal protein was used to normalize the amount of PC.

As previously described³², PC 16:0/18:1 and 18:1/18:1 were measured for each FA-CoA ester condition to check for possible changes in endogenous microsomal PC during the incubation period. As shown in **Fig. 19C**, the amount of these endogenous PC didn't change regardless to the FA-CoAs added. On the other hand, consistently with the data reported in the manuscript (**Fig. 5j**), *LPCAT3* abrogation induced a significant decrease in these endogenous PC species, which contain SFA and/or MUFA, indicating that LPCAT3 can conjugate these FA-CoAs.

However, as these changes might not entirely reflect a direct consequence of *LPCAT3* knockdown, and 48 hour-*LPCAT3* knockdown might have induced other lipid rearrangements, we followed the incorporation of 16:0-CoA and 20:4-CoA into PC deriving from the exogenously provided LysoPC 17:1.

Of note, in the *LPCAT3*-knockdown microsomes, there was a significant decrease not only in the incorporation of 20:4, but also of 16:0 into PC 17:1/20:4 and PC 17:1/16:0 respectively (**Fig. 19D**).

In addition, when exposing microsomes from A549 cells with active *LPCAT3* to the same molar concentration of 16:0 or 20:4 FA-CoAs, 16:0 incorporation into PC 17:1/16:0 is higher than 20:4 incorporation into PC 17:1/20:4 (**Fig. 19D**).

This assay suggests that, in the context of KMLC, the enzymatic activity of *LPCAT3* is not specific for AA, as it uses also saturated FA-CoAs (e.g. 16:0-CoA) to re-acylate LysoPC. Thus, this finding is in agreement with our data obtained with lipidomics, C11-BODIPY staining and competitive FA uptake assay

(Fig. 5j, k, Fig. 6f-i, Supplementary Fig. 6 and Fig. 19). All in all, these data strengthen the conclusion that the microenvironment and the extracellular abundance of PUFA/MUFA/SFA might dictate the substrate selection for LPCAT3.

We provided this newly obtained evidence in Supplementary Fig. 8, at page 11 and page 12 of the amended manuscript and in Supplementary information (page 6, lines 21-23 and page 7).

Q5: The reviewer noted that the distribution of CE 22:6 showed in Fig. 1e is different than other PUFA-containing glycerol lipids and asked to discuss possible explanation for such a difference.

A5: As remarked by the reviewer, MALDI-IMS experiments in KMLC tissue sections showed that, despite being esterified with a PUFA, CE 22:6 is present not only in the stromal area, but in the tumor area as well. Thus, it has a tissue distribution different than PUFA-containing phospholipids, more specifically PUFA-PC, which are excluded from tumor areas in KMLC.

Here, we provide some discussion to explain this apparently counterintuitive finding.

While within eukaryotic cell membranes, cholesterol is essential for regulating fluidity and permeability of the bilayer, outside the membrane, it is esterified with FA to form cholesterol esters (CE). Intracellular CE are stored in unique organelles called lipid droplets (LDs), which consist of a neutral lipid core (CEs and triacylglycerides) surrounded by an amphipathic monolayer mostly comprised of PC³³.

Availability of intratumoral CE is known to favor membrane biogenesis, lipid raft formation and cell signaling, all essential processes for tumor proliferation, invasiveness and survival³⁴.

There are numerous species of CE present in animals, with cholesteryl esters of linoleate [CE(18:2)], arachidonate [CE(20:4)], and docosahexaenoate [CE(22:6)], being the most abundant. Notably, all these CE are esterified with PUFA³⁵. Accordingly, CE 22:6 was one of the species we were able to detect in our MALDI-IMS experiments in its protonated form, [CE(22:6)+H]⁺, at *m/z* 719.57 (Fig. 1e).

CE are known to undergo recurrent cycles of hydrolysis and esterification, known as the CE cycle (Fig. 21)³⁶. Once re-esterified, CE accumulate in LDs.³⁷

Figure 20. Simplified schematic representation of the proposed interactions between the CE cycle and the Lands cycle acyl remodeling pathways.

The proposed route of oxPC formation through oxCE remodeling is indicated by red arrows. ACAT, acetyl-CoA acyltransferase; LPCAT, lysophosphatidylcholine acyl transferase; nCEH, neutral cholesterol hydrolase; PLA₂, phospholipase A₂. LPO: lipid oxidation, FC: free cholesterol.

PUFA-CE also are oxidized by reactive oxygen species (ROS) and, mainly, by 12/15-lipoxygenase (ALOX12/15) to form oxidized CE (oxCE)^{38,39}.

Studies of LDL oxidation indicated that ALOX15 prefers to oxidize the CE(18:2) rather than free linoleic acid (C18:2) *in vitro*⁴⁰.

The vast repertoire of oxCE exhibit various, context-dependent biological activities⁴¹.

As it occurs for PL, also the oxidized CE (oxCE) can be hydrolyzed releasing oxidized fatty acyl chains (oxFA) that can be incorporated into PL³⁹. Thus, it is tempting to speculate that the CE cycle might be another strategy put in place by KMLC to prevent PUFA incorporation into PL, so to escape ferroptosis. Hence, we speculate that, in KMLC, PUFA-CE are 'trapped' within LDs, in order to prevent PUFA peroxidation and their incorporation into PL, such as PC (Fig. 20).

On the contrary, under FASNi, the depletion of LDs (Fig. 3d, Supplementary Fig. 5d) causes PUFA to be released from CE. These PUFA, along with those

uptaken from extracellular sources, become available for reacylation into PC, thus leading to ferroptosis.

We are actively testing this hypothesis. However, we believe that answering experimentally to this question goes beyond the scope of this manuscript. Accordingly, we provide the following data for the Reviewer's eyes only.

To begin testing this hypothesis, we compared the tissue localization of [CE(22:6)+H]⁺ and LDs, visualized by O Red Oil (ORO) staining in PDX sections (Fig. 21C, D). Notably, the tissue distribution of [CE(22:6)+H]⁺ mirrors the areas stained by ORO, which are enriched in LDs (Fig. 21).

To further support our hypothesis, we reassessed the quantitative MS/MS data from A549 KMLC xenografts (**Fig. 7h** and **Fig. 22**). CE were identified using the $[M+NH_4]^+$ fragment at m/z 369.31 (Table 2). Among CE species, we focused on PUFA-containing CE. In accordance with the hypothesis of CE 22:6 being “trapped” intracellularly within the LDs, we found that CE 22:6 and other PUFA-CE, e.g. CE 20:3, 20:4, 20:5, were significantly decreased upon FASNi treatment (**Fig. 7h** and **Fig. 22**).

Figure 21. Tissue distribution of CE 22:6 and LDs in representative sections from PDXs. (A) H&E staining; MALDI-IMS data for $[CE\ 22:6+H]^+$; **(C)** ORO staining and **(D)** their magnification are shown. Scale bar: 2mm in A-C, 1mm in D. Tumor (T) and stroma (S) ROI are indicated. HCC-4059: KMLC PDX, HCC-4190: EGFR-MUT PDX, CP58391: KRAS-WT, EGFR-WT PDX.

Figure 22. FASNi treatment depletes KMLC of PUFA-CE *in vivo*. Volcano Plot shows CE esterified with PUFA that are differentially represented in FASNi treated A549 xenograft vs control. The adjusted P value and difference were calculated using multiple t tests with $\alpha = 0.05$. CE 20:4, CE 20:3, CE 20:5 and CE 22:6 were identified by MS/MS lipidomics using the $[M+NH_4]^+$ fragment at m/z 369.31.

Lipid Annotation	MS1 Molecular Formula	MS2 Molecular Formula	Precursor	Fragment	Neutral Loss	Adduct
CE(20:3)	C47H82O2N	C27H45	692.54	369.3123	323.2277	$[M+NH_4]^+$
CE(20:4)	C47H80O2N	C27H45	690.54	369.3146	321.2254	$[M+NH_4]^+$
CE(20:5)	C47H78O2N	C27H45	688.54	369.3127	319.2273	$[M+NH_4]^+$
CE(22:6)	C49H80O2N	C27H45	714.57	369.3128	345.2572	$[M+NH_4]^+$

Table2. PUFA-CE in A549 xenograft tissue.

Minor Q6: HPLC mis-spelled

Minor A6: We edited this typo.

Minor Q7: The reviewer asked why we used M+4/M0 or M+2/M0 in Fig. 5b and asked for all the isotopic enrichment.

Minor A7: In Fig. 5b, as in all the other labeling experiments, we used 1,2-ethyl acetate- $^{13}\text{C}_2$ as a tracer. Briefly, as explained in the “Methods” section of the manuscript, after 4 days of treatment, with either vehicle or FASNi, cells were incubated with the tracer overnight. The polar fraction (containing PC) was obtained, the FA hydrolyzed, derivatized and their profiles analyzed by a modified GC-MS method previously described⁴².

As the tracer (1,2-ethyl acetate- $^{13}\text{C}_2$) contains two (^{13}C) labeled carbons, molar enrichment of FA, calculated as ratio over unlabeled mass (M0), with two ^{13}C carbons (M+2) and four ^{13}C carbons (M+4) were determined (Fig. 23A).

Also, we report in Fig. 23B the isotopic enrichment (M0, M+2, M+4) relative to data showed in Fig. 5b.

Figure 23. 1,2-ethyl acetate- $^{13}\text{C}_2$ labeling. (A) 1,2-ethyl acetate- $^{13}\text{C}_2$ incorporation produces double-labeled C fatty acids. (B) Mass isotopolog distribution for Arachidonic Acid (FA 20:4) for the indicated cell lines. Cells were treated with either vehicle or FASNi and incubated with the tracer (1,2-ethyl acetate- $^{13}\text{C}_2$) before performing polar lipid extraction, FA derivation and GC/MS analysis. The isotope distribution is expressed as unlabeled mass (M0) and up to 4 mass units heavier (M+4) than the unlabeled compound. There were three biological replicates for each data point. Vertical bars

Minor Q8: The reviewer asked how the MALDI lipidomic data were normalized.

Minor A8: In the MSiReader software¹, which we used for MALDI-IMS data analysis, several normalization strategies can be selected: median, total ion count (TIC), root mean square (RMS), internal standard normalization, or no normalization as shown in Fig. 24A. Normalization factors for each pixel are calculated during the creation of the mslQuant data for the different normalization methods, and hence no further calculation is needed during data evaluation. The MSI results can be displayed using either the m/z distribution or the normalization factor distribution.

TIC is one of the most commonly applied normalization procedures in mass spectrometry. Here, all mass spectra are divided by their TIC (*i.e.* the sum of intensities of all the peaks) so that all spectra in the data set have the same integrated area under the spectrum

Distribution of normalization is shown in Fig. 24 (showed only to the reviewer). In addition to the matrix signal, we also detected endogenous molecules that were distributed uniformly in lung tissue sections in all the ROIs selected (both area classified as tumors and stroma). For instance, Fig. 24A and Fig. 24B show even distribution of positive-ion MALDI-IMS images of m/z 790.53, corresponding to sodiated PE 38:4

([PE 38:4+Na]⁺). Uniform distribution of this endogenous molecule was confirmed for all three charge carriers, H⁺, Na⁺ and K⁺ in positive-ion mode (**Fig. 24B**).

When we compared TIC, median, RMS normalization methods and no normalization we found that the signal distribution of PE 38:4 appears more consistent with the H&E staining after TIC than with the other normalizations/no normalization (**Fig. 24C**).

Hence, all the MALDI-IMS data were displayed as TIC-normalized. To better clarify this point, we edited the manuscript (page 26, lines 11-20), figures and their legends (**Fig. 1, Supplementary Fig.1, Supplementary Fig. 10**), and provided the lipid annotation in **Supplementary Table 1**.

Moreover, in an attempt to provide a “semi-quantitative” analysis, we have now calculated for each image and ROI: the number of pixels, sum of intensities for all pixels, average intensity per mm², average intensity per pixel, standard deviation of intensity per pixel, relative standard deviation of intensity per pixel, median intensity per pixel, lower quartile (Q1) intensity per pixel, higher quartile (Q3) intensity per pixel, and the minimum and maximum intensities of each pixel (in **Supplementary Tables 2-4, and Supplementary Table 8**).

Figure 24. MALDI-IMS images for PE 38:4. PE 38:4 (*m/z* 790.54) was chosen because of its homogenous distribution among selected ROIs. **(A)** The peak observed at *m/z* 790.54 is displayed using different normalization methods: no normalization, TIC, Median or RMS normalization. Human (TH7037 and L140) and PDX (HCC_4059, HCC_4190, CP58391) are shown at the top and bottom respectively. **(B)** Distribution of *m/z* PE 38:4 in the three carriers is displayed using the TIC normalization: [PE(38:4)+Na]⁺ at *m/z* 790.54, [PE(38:4)+H]⁺ at *m/z* 768.55, [PE(38:4)+K]⁺ at *m/z* 806.50). **(C)** H&E staining performed on tissue sections contiguous to the ones used for MALDI-IMS. T, tumor; S, stroma; S+T, stroma and tumor mix. PE: Phosphatidylethanolamine, TIC: Total Ion Count, RMS: Root Mean Square. Scale bar: 2mm.

Answer to Reviewer #4

Q1, Q2: The reviewer asked to explain with better clarity the rationale “for choosing FASN inhibitor to test KMLC dependency on *de novo* lipogenesis” and they asked whether this correlation is significant. In addition, the reviewer asked to comment on whether there is any other protein expression that correlates with KM in TetO-Kras^{G12D} mice. Also, the reviewer asked to move line 17 to the beginning.

A1, A2: As suggested, we moved the sentence at the beginning of the paragraph, we added the reference of a previous report from our lab, in which we performed dox-withdrawal in TetO-Kras^{G12D} mice⁴³, and we further discussed our rationale for choosing FASNi at page 3 lines 8-10 and page 6 lines 4-10 of the revised manuscript.

Expression profiling revealed that upon Kras^{G12D} extinction, several lipid synthesis/metabolism genes (such as *Acs13*, *Acs14*, *Fasn*, *Elovl1*, *Srebf1*) were significantly downregulated⁴³. A year later, D. Felsner's lab, showed that KRAS activates a lipogenesis gene signature and specific transcriptional induction of FASN⁴⁴. All in all, this background along with the lipidomic results in this manuscript, pointed toward a role of fatty acid synthesis in KMLC tumorigenesis.

Therefore, we decided to target FASN because: 1) it is the rate-limiting enzyme of the process and 2) second-generation FASN inhibitors were available and non-toxic in preclinical/clinical trials^{45,46}. Then, to further validate the association between FASN and KM, we analyzed FASN expression and protein level in TetO-Kras^{G12D} mice, human LC cell lines and two human lung tumor microarrays (**Supplementary Fig. 2**), and we found a significant correlation confirming that KMLC has a high FASN level.

Q3: The reviewer asked to “discuss potential mechanisms underlying the upregulation of SCD/LPCAT3/ACSL3/PLA2G4C expression in response to FASN inhibition in KMLC, even though this topic is not a main focus on this study”.

A3: By analyzing the list of genes upregulated by FASNi with EnrichR^{47,48}, we found an enrichment in transcriptional targets of the sterol regulatory element binding factor 1 (*SREBF1*) and of the retinoid-X receptor alpha (*RXRA*) (**Fig. 25**). This evidence is in agreement with our data and previous literature^{44,49}, showing that inhibition of *de novo* lipid synthesis in KMLC causes: 1) SFA/MUFA scarcity and 2) scavenging of exogenous FA (that in KMLC are mainly PUFA).

SREBF1, the master regulator of lipid synthesis, senses lipid deprivation (especially MUFA deprivation), translocates into the nucleus and activates the transcription of *de novo* lipogenesis genes⁵⁰.

ENCODE TF ChIP-seq 2015

Bar Graph **Table** Grid Network Clustergram Appyter  
Hover each row to see the overlapping genes.

10 entries per page

Search:

Index	Name	P-value	Adjusted p-value	Odds Ratio	Combined score
1	SREBF1 HepG2 hg19	0.000001922	0.0004265	156.60	2061.20
2	RXRA H1-hESC hg19	0.000001934	0.0004265	261.59	3441.42
3	SREBF2 GM12878 hg19	0.00007997	0.01176	73.11	689.70
4	RFX5 K562 hg19	0.0004588	0.03808	36.07	277.27
5	RXRA SK-N-SH hg19	0.0004730	0.03808	96.63	739.85
6	RXRA HepG2 hg19	0.0005181	0.03808	38.07	288.00
7	SREBF1 GM12878 hg19	0.002132	0.1343	22.86	140.63
8	REST A549 hg19	0.003950	0.1714	18.18	100.61
9	PRDM1 HeLa-53 hg19	0.004724	0.1714	16.99	90.99
10	POU2F2 GM12891 hg19	0.005233	0.1714	16.34	85.84

Figure 25. List of transcription factors binding genes upregulated by FASNi in KMLC. Analysis done in EnrichR.

Similarly, RXRs (*i.e.* RXRA) function as obligate heterodimeric partners with of the peroxisome proliferator-activated receptors (PPARs) which, once activated by PUFA, translocate into the nucleus and activate the transcription of *de novo* lipogenesis and lipid metabolism genes⁵¹⁻⁵³. Therefore, it seems reasonable to hypothesize that the lipidomic reprogramming caused by FASNi might reshape the transcriptional activity of these nuclear receptors.

In this regard, we are completing a manuscript characterizing these events in KMLC.

Q4: The reviewer asked how KM induces ROS in lung cancer cells.

A4: A well-developed literature reported that oncogenic RAS can lead to an increase in ROS production via multiple mechanisms^{54,55}. ROS can exert a dual role, both mediating the oncogene-induced senescence, but also promoting RAS-induced tumorigenicity. Oncogenic RAS can orchestrate the ROS production via several ways, by transcriptional regulation of pro-oxidant genes such as *NOX1*⁵⁶⁻⁵⁸ and *COX2*⁵⁹, by regulating the mitochondrial ROS production⁶⁰, and repressing anti-oxidant enzymes, such as sestrins and peroxiredoxins⁶¹. We added a sentence about this topic at page 17, lines 8-9.

Q5: The reviewer asked to “tone down” our statement about PC and TAG uniquely increased in KMLC with respect to EGFR-MUT.

A5: We edited the text accordingly (page 5 of the revised manuscript).

Q6: The reviewer asked to determine the best combination of KRAS inhibition (KMi) and FASNi for tumor suppression and to better characterize whether the survivor cells are senescent or quiescent.

A6: Following the reviewer suggestion, we performed viability assays subjecting the human *KRAS*^{G12C} LC cell line H2122 to several co-treatment conditions with FASNi and ARS-1620 (KMi). We chose H2122 cells because these cells are not only representative of our studies, but also because they have been shown to be representative of the response to KM^{G12C} inhibitors⁶². As shown in **Fig. 26A (Supplementary Fig. 4d)** of the revised manuscript, in all the conditions, single treatments with either FASNi and ARS-1620 outperformed the correspondent combinations, regardless of the order of the treatments. This is particularly evident comparing FASNi 72h (red bars) to FASNi 72h followed by 24h co-treatment with ARS-1620 (burgundy bars). Even though these findings are disappointing, given the increasing relevance of KM^{G12C} inhibitors, they are consistent with our conclusion that KM is necessary to induce FASN dependency.

We discussed these results at page 7 lines 5-9 of the revised manuscript.

We agree with the reviewer that it is surprising that KMLC cells survive better when treated with both inhibitors. We hypothesize that ARS-1620, by inhibiting KM, alleviates the KM-dependent oxidative stress that is required for ferroptosis execution. Of note, neither ARS-1620 nor FASNi induce cell death rapidly. Thus, another possible explanation is that the double selective pressure caused in presence of both inhibitors, might accelerate the selection and outgrowth of persisters which are independent of both FASN and KM.

Moreover, following the reviewer's suggestion, we examined senescence-associated beta-galactosidase (SA-β gal) in cells treated for 48h with either FASNi or ARS-1620 alone, or treated for additional 48h with combination treatment⁶³. We found that all treatments (with the exception of FASNi 0.3 μM) induce a significant increase in the percentage of SA-β gal positive cells. Interestingly, we found that: 1) FASNi followed by ARS-1620 treatment significantly increases SA-β gal positive cells with respect to FASNi treatment alone; 2) ARS-1620 produces a higher % of SA-β gal positive cells than FASNi followed by co-treatment with ARS-1620 0.5 μM. With these preliminary data we can speculate that KM inhibition might be the main determinant of a senescence-like phenotype, at least in cell culture.

Finally, the reviewer asked whether these cells are still tumorigenic *in vivo*. We were not able to address this point, since these cells have failed to grow as xenografts in a reasonable time frame (2 months).

This finding is consistent with preliminary evidence indicating that these cells have a long doubling time.

We are currently addressing potential mechanisms that lead to resistance to FASNi and G12Ci treatments, but we think that these investigations are beyond the scope of this manuscript. Therefore we provided the data relative to senescence for the reviewers only.

Figure 26. Cell viability and senescence in H2122 cells upon ARS-1620 and FASNi treatments. (A) MTT cell viability assay in KM-G12C cell line H2122 treated as indicated. (B, C) Representative picture and relative quantification of SA senescence-associated beta-galactosidase (SA-β gal) after 48h with FASNi/ARS-1620 alone or followed by another 48h of combination. n=2 independent experiments. In (B, C) multiple t test with ns, p>0.05; *p<0.05; **p<0.01; ***p<0.001; ****p<0.0001.

regulatory pathways could dictate biological responses *in vivo*.

Lysophospholipid Acyltransferase (LPAT) activity assays are generally performed exposing cell extracts containing the enzyme of interest to a pair of pure substrates (one lysophospholipid and an acyl-CoA ester) and measuring the conversion of the substrate into the product.

To test whether, at least in the context of KMLC, LPCAT3 re-acylates LysoPC with palmitate-CoA (16:0-CoA), we adapted an established biochemical assay³² (included in Supplementary Fig. 8 of the revised manuscript), taking advantage of A549 KMLC cells stably expressing a doxycycline (doxy)-inducible shRNA for the human LPCAT3 (already used in Fig. 5j, k and Supplementary Fig. 7). The assay consists in

Q7: The reviewer noted that our findings are apparently contradictory with LPCAT3 being “believed to be an enzyme that preferentially incorporates polyunsaturated fatty acyl- (PUFA) CoA into lyso-PL”. Thus, the reviewer asked for other evidence supporting our claim.

A7: We thank the reviewer for this question. Since reviewer #3 asked the same question in (Q4), we pasted below *verbatim* our answer **A4**. We are aware that our data are in contrast with most of the literature currently available about LPCAT3 activity. Indeed, LPCAT3 is largely reported to have preference for arachidonic acid, AA (20:4). However, most of the data about LPCAT3 enzymatic activity have been obtained in intestine and liver, where it is quite abundant, either *in vivo* or in human and mouse hepatoma cell lines²⁷⁻³¹.

Although the initial enzymatic characterization of the Lands Cycle occurred almost 50 years ago, little is known about its role *in vivo*. To date, most studies on phospholipids (PL) fatty acyl composition have utilized *in vitro* biochemical assays, due to the difficulty of detecting specific changes in membrane composition in living cells. Therefore, there is little understanding of how regulatory pathways control PL fatty acyl composition and how such

incubating a mixture of acyl-CoAs and LysoPC with microsomal extracts (the fraction enriched in LPCAT3), and then analyzing specific PC species by LC-MS/MS (**Fig. 19A**).

First, microsomes were prepared from A549 shLPCAT3, either kept in doxy or no doxy for 48 hours (**Fig. 19A, B**). Second, the fatty acyl-CoA esters chosen were the ones relevant to our question, hence, 20:4-CoA and 16:0-CoA. Third, as previously described³², we used 1-(10Z)heptadecenoyl-2-hydroxy-lysophosphatidylcholine (17:1-LysoPC) containing *sn*-1 acyl chain 17:1, which reduces the background signal from endogenous phospholipids present in the microsomal preparations.

To identify PC we used the acetate adduct $[M+CH_3COO]^-$ ions and data were analyzed with the in-house script software LipPy. Evaluation of the enzymatic reaction was performed by calculating the ratio of the integrated area of PC to that of the corresponding internal standard. The absolute amount of microsomal protein was used to normalize the amount of PC.

As previously described³², PC 16:0/18:1 and 18:1/18:1 were measured for each FA-CoA ester condition to check for possible changes in endogenous microsomal PC during the incubation period. As shown in Fig. 20C, the amount of these endogenous PC didn't change regardless to the FA-CoAs added.

On the other hand, consistently with the data reported in the manuscript (**Fig. 5j**), *LPCAT3* abrogation induced a significant decrease in these endogenous PC species, which contain SFA and/or MUFA, indicating that *LPCAT3* can conjugate these FA-CoAs.

However, as these changes might not entirely reflect a direct consequence of *LPCAT3* knockdown, and 48 hour-*LPCAT3* knockdown might have induced other lipid rearrangements, we followed the incorporation of 16:0-CoA and 20:4-CoA into PC deriving from the exogenously provided LysoPC 17:1.

Of note, in the *LPCAT3*-knockdown microsomes there was a significant decrease not only in the incorporation of 20:4, but also of 16:0 into PC 17:1/20:4 and PC 17:1/16:0 respectively (**Fig. 19D**).

In addition, when exposing microsomes from A549 cells with active *LPCAT3* to the same molar concentration of 16:0 or 20:4 FA-CoAs, 16:0 incorporation into PC 17:1/16:0 is higher than 20:4 incorporation into PC 17:1/20:4 (**Fig. 19D**).

This assay suggests that, in the context of KMLC, the enzymatic activity of *LPCAT3* is not specific to AA, as it uses also saturated FA-CoAs (e.g. 16:0-CoA) to re-acylate LysoPC. Thus, this finding is in agreement with our data obtained with lipidomics, C11-BODIPY staining and competitive FA uptake assay (**Fig. 5j, k, Fig. 6f-i, Supplementary Fig. 6** and **Fig. 19**). All in all, these data strengthen the conclusion that the microenvironment and the extracellular abundance of PUFA/MUFA/SFA might dictate the substrate selection for *LPCAT3*.

We provided this newly obtained evidence in **Supplementary Fig. 8**, in the amended manuscript (page 11 lines 20-22 and page 12 lines 1-14) and in Supplementary information (page 6 lines 21-23 and page 7).

Figure 19. LPCAT3 enzymatic activity. (A) Schematic representation of the experimental approach. A549 cells were transfected with doxy-inducible shRNA targeting LPCAT3. Lipids were extracted and analyzed by HPLC-MS/MS. (B) WB analysis showing enrichment of LPCAT3 in purified microsomes (M) and the efficiency of LPCAT3 knockdown. (C) The most abundant endogenous PC species PC 16:0/18:1, PC 18:1/18:1, were detected. Upon doxy-induction their levels were significantly reduced but did not change according to the CoA added. (D) PC deriving from reacylation of exogenously provided LysoPC 17:1 with either 16:0-CoA or 20:4 CoA.

In (C, D) multiple t test with ns, $p > 0.05$; * $p < 0.05$; ** $p < 0.01$; *** $p < 0.001$; **** $p < 0.0001$.

Q8: The reviewer asked whether we “compared the FASNi sensitivity between KM and KRAS-WT cells”.

A8: Indeed, we performed these experiments and presented them in our initial manuscript in which we showed the sensitivity to FASNi in both cell line groups in **Fig. 2** (one point concentration was shown just for clarity) and **Fig. 27** (now **Supplementary Fig. 3c** of the revised manuscript). We edited the text of the amended manuscript to increase its clarity (page 6, lines 14-16).

Q9: The reviewer noted that all our genetic perturbations were achieved using siRNAs or shRNAs. Thus, they wondered what precluded us from using CRISPR/Cas9 technology.

A9: As the reviewer noticed, in all the genetic perturbation experiments we took advantage of either siRNAs or shRNAs (**Fig. 2g, h; Fig. 5h, j, k; Fig. 6h, i; Supplementary Fig. 2d, e; Supplementary Fig 4c; Supplementary Fig 7i, j**).

We adopted such a strategy because we reasoned that the selection process required by CRISPR/Cas9 technology would potentially allow for selection of FASNi resistant subclones. As we carried out our *in vitro* experiments in a short-term timeframe (4 days), we preferred to use shRNA vectors to provide high cell-to-cell uniformity within the pool of treated cells and to avoid the selection of resistant subclones.

Q10: The reviewer asked to provide Ferrostatin-1 (Fer-1) rescue experiments in additional KMLC cell lines.

A10: As suggested by the reviewer, we performed viability rescue experiments with Fer-1 in three additional KMLC cell lines (**Fig. 28**). These data are now presented in **Supplementary Fig 7** of the revised manuscript.

Q11: The reviewer asked to explain the discrepancy in the total PC contents in the KRAS-mutant samples between Figure 4a and Figure 1a.

A11: We thank the reviewer for this important question, as this is a crucial point of our study.

In Fig. 1a we demonstrated the contribution of KM to *de novo* lipogenesis and the lipidomic profile of LC. On the other hand, in Fig. 4a we demonstrated the effects of FASNi. We observed no changes in the total amount of PC.

The most likely explanation of this observation is that under FA deprivation KMLC compensates for the lack of synthesized acyl chains by scavenging exogenous FA (which in the lung are mainly PUFA, thus prone to oxidation) for the remodeling of PC. This result is in agreement with data in Fig. 4e, f; Fig. 5 a-d, Fig. 7h, and the FA-uptake experiment (please refer to reviewer #3 Q3, A3, Supplementary Fig 6).

We discussed this conclusion at page 17 of the revised manuscript.

Q12: Text

- 1. The abstract is too long and should be revised to highlight the key findings.**
- 2. P18-line 8, intratumor availability “of” SFA and MUFA**
- 3. P17-line22, “then” should be “than”**
- 4. Page 19 line 5: the sentence “KrasG12D inhibitor...reverses the effects...” is confusing. The author should revise this statement.**
- 5. P19-line 14, depletion of lipid droplets is presented by Figure 3D not Figure 3G**
- 6. P21-line 8, Fig not Fid**
- 7. P21-line 5-8, these statements are unnecessary and are not fully supported by the data**
- 8. P22-line 14: Fig. S6G and S6H do not exist. Are the authors referring to Figure S6E and S6F? 9. P42-line 12, HPLC has a typo?**

A12: All points have been assessed and modified according to the reviewer's comments.

Q13: Figures

Q13.1: Fig. 1a: The reviewer asked to describe the precise data normalization method used in processing the lipidomic analysis results.

A13.1: Fig. 1a refers to lipidomics experiments performed in laser-captured microdissected samples from GEEMs TetO-KrasG12D tumors, TetO-EGFRL858R tumors and unaffected healthy lung (*i.e.*, before Doxycycline induction).

We explained our procedures in detail in the “methods” section (pages 27-28). After sample preparation and analysis via a SCIEX quadrupole time-of-flight (QTOF) TripleTOF 6600+ mass spectrometer, data were acquired at each unit mass from 200-1200 Da using Analyst TF 1.7.1 software (SCIEX) and analyzed using an in-house script, LipPy.

This software allows for lipid species identification, data normalization, and basic statistics. In particular, it allows to normalize the identified lipid species to the SPLASH LipidoMix™ internal standard. Specifically, according to the lipid class they belong to, each lipid is normalized over the corresponding IS: 15:0-18:1(d7) PC, 15:0-18:1(d7) PE, 15:0-18:1(d7) PG, 15:0-18:1(d7) PS 15:0-18:1(d7) PI, 15:0-18:1-d7-PA, 18:1(d7) LPC, 18:1(d7) LPE, 18:1(d7) Chol Ester, 18:1(d7) MG, 15:0-18:1(d7) DG, 15:0-18:1(d7)-15:0 TG, 18:1(d9) SM or Cholesterol (d7).

For Fig. 1a, after normalization, all the identified lipid species were clustered into the major lipid species families and plotted as bars indicating mean ± SD.

Q13.2: Fig. 1a: Need x Axis label

A13.2: Done

Q13.3: Fig. 1e right panel, the reviewer noticed the second PC 18:1/20:4 was labeled wrong and it should be labeled as PC 20:4/20:5/

A13.3: Done

Q13.4: Fig. S1b and 7h: x-axis should be Log, not -Log.

A13.4: Done

Q13.5: **Fig. 1f-g, Supplementary Fig. 1b:** The reviewer asked whether the y-axis is P-value or adjusted P-value.

A13.5: It's adjusted P-value.

Q13.6: **Fig. 2b,** the reviewer asked if we included a palmitate-only condition as a control.

A13.6: We did not include a palmitate-only condition because, in absence of FASNi, palmitate supplementation induces toxicity.

Q13.7: **Fig. 6h and 6i,** the reviewer asked if we confirmed the knockout efficacy of the siRNAs for LPCAT3 and PLA2G4C by qPCR or western blot.

A13.7: Yes, we performed this control (**Fig. 29**).

Q13.8: In **Fig. 3a** the reviewer asked why certain fatty acids are analyzed using M+2/M+0 whereas others are using M+4/M+0?

A13.8: In **Fig. 3a**, as in all the other labeling experiments, we used 1,2-ethyl acetate-¹³C₂ as a tracer.

As the tracer (1,2-ethyl acetate-¹³C₂) contains two (¹³C) labeled Carbons, molar enrichment of FA with two, four, six and eight ¹³C carbons (M+2; M+4; M+6; M+8, respectively) were calculated as the ratio of labelled/ unlabeled mass (M0). We showed just an isotopomer in **Fig. 3a**. Hence, we hereafter reported the whole isotope enrichment distribution for representative KRAS-WT (H522) and KM (H460) human cell lines upon FASNi treatment (**Fig. 30**).

Figure 29. Western Blot for LPACT3 and PLA2G4C in lysates of A549 transfected with the indicated siRNAs. The same cells were used in **Fig. 6h** and **6i**.

Q13.9: In **Figures 4B,4D,4F,4H:** Need x Axis label

A13.9: Done

Q13.10: **Figure S7B (now Supplementary Fig. 9):** Need to add a legend or label what red and black line represent

A13.10: Done

Figure 30. 1,2-ethyl acetate-¹³C₂ labeling.

1,2-ethyl acetate-¹³C₂ incorporation into H522 KRAS-WT LC cells. **(A)** and H460 KM LC cells **(B)**. Isotope fractional enrichment for Palmitate (FA 16:0), Oleic Acid (FA 18:1n9) and vaccenic acid (FA 18:1n7) was calculated as ratio labelled/unlabeled mass (M0). Cells were treated with either vehicle (black) or FASNi (red) and incubated with the tracer (1,2-ethyl acetate-¹³C₂) before performing lipid extraction, FA derivation and GS/MS analysis. N=3, bars mean ± SD. Unpaired t test with ***, p<0.001, ****, p<0.0001, **, p<0.01.

REFERENCES

1. Källback, P., Nilsson, A., Shariatgorji, M. & Andrén, P. E. MslQuant - Quantitation Software for Mass Spectrometry Imaging Enabling Fast Access, Visualization, and Analysis of Large Data Sets. *Anal. Chem.* (2016) doi:10.1021/acs.analchem.5b04603.
2. Schwartz, S. A., Reyzer, M. L. & Caprioli, R. M. Direct tissue analysis using matrix-assisted laser desorption/ionization mass spectrometry: Practical aspects of sample preparation. *J. Mass Spectrom.* (2003) doi:10.1002/jms.505.
3. Murphy, R. C. *Tandem Mass Spectrometry of Lipids: molecular analysis of complex lipids. Tandem mass spectrometry of lipids* (2014).
4. Jackson, S. N., Wang, H. Y. J. & Woods, A. S. In situ structural characterization of phosphatidylcholines in brain tissue using MALDI-MS/MS. *J. Am. Soc. Mass Spectrom.* (2005) doi:10.1016/j.jasms.2005.08.014.
5. Wolf, S., Schmidt, S., Müller-Hannemann, M. & Neumann, S. In silico fragmentation for computer assisted identification of metabolite mass spectra. *BMC Bioinformatics* (2010) doi:10.1186/1471-2105-11-148.
6. Wang, Z. *et al.* The elongation of very long-chain fatty acid 6 gene product catalyses elongation of n-13 : 0 and n-15 : 0 odd-chain SFA in human cells. *Br. J. Nutr.* **121**, 241–248 (2019).
7. Wang, Z. *et al.* Fatty acid desaturase 2 (FADS2) but not FADS1 desaturates branched chain and odd chain saturated fatty acids. *Biochim. Biophys. Acta - Mol. Cell Biol. Lipids* **1865**, 158572 (2020).
8. Foulon, V. *et al.* Breakdown of 2-Hydroxylated Straight Chain Fatty Acids via Peroxisomal 2-Hydroxyphytanoyl-CoA Lyase: A REVISED PATHWAY FOR THE α -OXIDATION OF STRAIGHT CHAIN FATTY ACIDS*. *J. Biol. Chem.* **280**, 9802–9812 (2005).
9. Croes, K., Foulon, V., Casteels, M., Van Veldhoven, P. P. & Mannaerts, G. P. Phytanoyl-CoA hydroxylase: recognition of 3-methyl-branched acyl-CoAs and requirement for GTP or ATP and Mg²⁺ in addition to its known hydroxylation cofactors. *J. Lipid Res.* **41**, 629–636 (2000).
10. Guo, L., Zhou, D., Pryse, K. M., Okunade, A. L. & Su, X. Fatty Acid 2-Hydroxylase Mediates Diffusional Mobility of Raft-associated Lipids, GLUT4 Level, and Lipogenesis in 3T3-L1 Adipocytes*. *J. Biol. Chem.* **285**, 25438–25447 (2010).
11. Weitkunat, K. *et al.* Effects of dietary inulin on bacterial growth, short-chain fatty acid production and hepatic lipid metabolism in gnotobiotic mice. *J. Nutr. Biochem.* **26**, 929–937 (2015).
12. Jenkins, B. J. *et al.* Odd Chain Fatty Acids; New Insights of the Relationship Between the Gut Microbiota, Dietary Intake, Biosynthesis and Glucose Intolerance. *Sci. Rep.* **7**, 44845 (2017).
13. Jin, Y. J. *et al.* Carnitine palmitoyltransferase-1 (CPT-1) activity stimulation by cerulenin via sympathetic nervous system activation overrides cerulenin's peripheral effect. *Endocrinology* (2004) doi:10.1210/en.2004-0039.
14. Pizer, E. S. *et al.* Malonyl-coenzyme-A is a potential mediator of cytotoxicity induced by fatty-acid synthase inhibition in human breast cancer cells and xenografts. *Cancer Res.*

- (2000).
15. Thupari, J. N., Pinn, M. L. & Kuhajda, F. P. Fatty acid synthase inhibition in human breast cancer cells leads to malonyl-CoA-induced inhibition of fatty acid oxidation and cytotoxicity. *Biochem. Biophys. Res. Commun.* (2001) doi:10.1006/bbrc.2001.5146.
 16. Samudio, I. *et al.* Pharmacologic inhibition of fatty acid oxidation sensitizes human leukemia cells to apoptosis induction. *J. Clin. Invest.* (2010) doi:10.1172/JCI38942.
 17. Mikalayeva, V. *et al.* Fatty acid synthesis and degradation interplay to regulate the oxidative stress in cancer cells. *Int. J. Mol. Sci.* (2019) doi:10.3390/ijms20061348.
 18. Pizer, E. S., Chrest, F. J., DiGiuseppe, J. A. & Han, W. F. Pharmacological inhibitors of mammalian fatty acid synthase suppress DNA replication and induce apoptosis in tumor cell lines. *Cancer Res.* (1998).
 19. Menendez, J. A. & Lupu, R. Fatty acid synthase (FASN) as a therapeutic target in breast cancer. *Expert Opinion on Therapeutic Targets* (2017) doi:10.1080/14728222.2017.1381087.
 20. Zaytseva, Y. Y. *et al.* Preclinical evaluation of novel fatty acid synthase inhibitors in primary colorectal cancer cells and a patient-derived xenograft model of colorectal cancer. *Oncotarget* (2018) doi:10.18632/oncotarget.25361.
 21. Ferrick, D. A., Neilson, A. & Beeson, C. Advances in measuring cellular bioenergetics using extracellular flux. *Drug Discovery Today* (2008) doi:10.1016/j.drudis.2007.12.008.
 22. Dott, W., Mistry, P., Wright, J., Cain, K. & Herbert, K. E. Modulation of mitochondrial bioenergetics in a skeletal muscle cell line model of mitochondrial toxicity. *Redox Biol.* (2014) doi:10.1016/j.redox.2013.12.028.
 23. Carracedo, A., Cantley, L. C. & Pandolfi, P. P. Cancer metabolism: fatty acid oxidation in the limelight. *Nat Rev Cancer* **13**, 227–232 (2013).
 24. Ying, W. NAD⁺/NADH and NADP⁺/NADPH in cellular functions and cell death: Regulation and biological consequences. *Antioxidants and Redox Signaling* (2008) doi:10.1089/ars.2007.1672.
 25. Shao, C. *et al.* Cytosolic ME1 integrated with mitochondrial IDH2 supports tumor growth and metastasis. *Redox Biol.* (2020) doi:10.1016/j.redox.2020.101685.
 26. Ratledge, C. The role of malic enzyme as the provider of NADPH in oleaginous microorganisms: A reappraisal and unsolved problems. *Biotechnology Letters* (2014) doi:10.1007/s10529-014-1532-3.
 27. Hashidate-Yoshida, T. *et al.* Fatty acid remodeling by LPCAT3 enriches arachidonate in phospholipid membranes and regulates triglyceride transport. *Elife* (2015) doi:10.7554/eLife.06328.
 28. Rong, X. *et al.* Lpcat3-dependent production of arachidonoyl phospholipids is a key determinant of triglyceride secretion. *Elife* (2015) doi:10.7554/eLife.06557.
 29. Zhao, Y. *et al.* Identification and characterization of a major liver lysophosphatidylcholine acyltransferase. *J. Biol. Chem.* (2008) doi:10.1074/jbc.M710422200.
 30. Li, Z. *et al.* Lysophosphatidylcholine acyltransferase 3 knockdown-mediated liver lysophosphatidylcholine accumulation promotes very low density lipoprotein production

- by enhancing microsomal triglyceride transfer protein expression. *J. Biol. Chem.* (2012) doi:10.1074/jbc.M111.334664.
31. Rong, X. *et al.* LXRs regulate ER stress and inflammation through dynamic modulation of membrane phospholipid composition. *Cell Metab.* (2013) doi:10.1016/j.cmet.2013.10.002.
 32. Martin, S. A., Gijón, M. A., Voelker, D. R. & Murphy, R. C. Measurement of lysophospholipid acyltransferase activities using substrate competition. *J. Lipid Res.* (2014) doi:10.1194/jlr.D044636.
 33. Bartz, R. *et al.* Lipidomics reveals that adiposomes store ether lipids and mediate phospholipid traffic. *J. Lipid Res.* (2007) doi:10.1194/jlr.M600413-JLR200.
 34. Tosi, M. R. & Tugnoli, V. Cholesteryl esters in malignancy. *Clinica Chimica Acta* (2005) doi:10.1016/j.cccn.2005.04.003.
 35. Quehenberger, O. *et al.* Lipidomics reveals a remarkable diversity of lipids in human plasma¹. *J. Lipid Res.* (2010) doi:10.1194/jlr.M009449.
 36. Brown, M. S., Goldstein, J. L., Krieger, M., Ho, Y. K. & Anderson, R. G. Reversible accumulation of cholesteryl esters in macrophages incubated with acetylated lipoproteins. *J. Cell Biol.* (1979) doi:10.1083/jcb.82.3.597.
 37. Ho, Y. K., Brown, M. S. & Goldstein, J. L. Hydrolysis and excretion of cytoplasmic cholesteryl esters by macrophages: Stimulation by high density lipoprotein and other agents. *J. Lipid Res.* (1980) doi:10.1016/s0022-2275(20)39788-1.
 38. Rankin, S. M., Parthasarathy, S. & Steinberg, D. Evidence for a dominant role of lipoxygenase(s) in the oxidation of LDL by mouse peritoneal macrophages. *J. Lipid Res.* (1991) doi:10.1016/s0022-2275(20)42068-1.
 39. Hutchins, P. M. & Murphy, R. C. Cholesteryl ester acyl oxidation and remodeling in murine macrophages: Formation of oxidized phosphatidylcholine. *J. Lipid Res.* (2012) doi:10.1194/jlr.M026799.
 40. Belkner, J., Stender, H. & Kühn, H. The rabbit 15-lipoxygenase preferentially oxygenates LDL cholesterol esters, and this reaction does not require vitamin E. *J. Biol. Chem.* (1998) doi:10.1074/jbc.273.36.23225.
 41. Gonen, A. & Miller, Y. I. From Inert Storage to Biological Activity—In Search of Identity for Oxidized Cholesteryl Esters. *Frontiers in Endocrinology* (2020) doi:10.3389/fendo.2020.602252.
 42. Quehenberger, O., Armando, A. M. & Dennis, E. A. High sensitivity quantitative lipidomics analysis of fatty acids in biological samples by gas chromatography-mass spectrometry. *Biochimica et Biophysica Acta - Molecular and Cell Biology of Lipids* (2011) doi:10.1016/j.bbalip.2011.07.006.
 43. Padanad, M. S. *et al.* Fatty Acid Oxidation Mediated by Acyl-CoA Synthetase Long Chain 3 Is Required for Mutant KRAS Lung Tumorigenesis. *Cell Rep.* (2016) doi:10.1016/j.celrep.2016.07.009.
 44. Gouw, A. M. *et al.* Oncogene KRAS activates fatty acid synthase, resulting in specific ERK and lipid signatures associated with lung adenocarcinoma. *Proc. Natl. Acad. Sci. U. S. A.* (2017) doi:10.1073/pnas.1617709114.

45. Ventura, R. *et al.* Inhibition of de novo Palmitate Synthesis by Fatty Acid Synthase Induces Apoptosis in Tumor Cells by Remodeling Cell Membranes, Inhibiting Signaling Pathways, and Reprogramming Gene Expression. *EBioMedicine* **2**, 808–824 (2015).
46. Syed-Abdul, M. M. *et al.* Fatty Acid Synthase Inhibitor TVB-2640 Reduces Hepatic de Novo Lipogenesis in Males With Metabolic Abnormalities. *Hepatology* **72**, 103–118 (2020).
47. Chen, E. Y. *et al.* Enrichr: Interactive and collaborative HTML5 gene list enrichment analysis tool. *BMC Bioinformatics* (2013) doi:10.1186/1471-2105-14-128.
48. Kuleshov, M. V. *et al.* Enrichr: a comprehensive gene set enrichment analysis web server 2016 update. *Nucleic Acids Res.* (2016) doi:10.1093/nar/gkw377.
49. Kamphorst, J. J. *et al.* Hypoxic and Ras-transformed cells support growth by scavenging unsaturated fatty acids from lysophospholipids. *Proc Natl Acad Sci U S A* **110**, 8882–8887 (2013).
50. Hagen, R. M., Rodriguez-Cuenca, S. & Vidal-Puig, A. An allostatic control of membrane lipid composition by SREBP1. *FEBS Lett.* **584**, 2689–2698 (2010).
51. Grygiel-Górniak, B. Peroxisome proliferator-activated receptors and their ligands: nutritional and clinical implications - a review. *Nutr. J.* **13**, 17 (2014).
52. Zúñiga, J. *et al.* N-3 PUFA Supplementation Triggers PPAR- α Activation and PPAR- α /NF- κ B Interaction: Anti-Inflammatory Implications in Liver Ischemia-Reperfusion Injury. *PLoS One* **6**, 1–9 (2011).
53. Kliewer, S. A. & Willson, T. M. The nuclear receptor PPAR γ -bigger than fat. *Curr. Opin. Genet. Dev.* **8**, 576–581 (1998).
54. Bartolacci, C., Andreani, C., El-Gammal, Y. & Scaglioni, P. P. Lipid Metabolism Regulates Oxidative Stress and Ferroptosis in RAS-Driven Cancers: A Perspective on Cancer Progression and Therapy. *Front. Mol. Biosci.* **8**, 791 (2021).
55. Lim, J. K. M. & Leprivier, G. The impact of oncogenic RAS on redox balance and implications for cancer development. *Cell Death Dis.* **10**, 955 (2019).
56. Mitsushita, J., Lambeth, J. D. & Kamata, T. The superoxide-generating oxidase Nox1 is functionally required for Ras oncogene transformation. *Cancer Res.* **64**, 3580–3585 (2004).
57. Irani, K. *et al.* Mitogenic signaling mediated by oxidants in Ras-transformed fibroblasts. *Science* (80-.). **275**, 1649–1652 (1997).
58. Park, M.-T. *et al.* Novel signaling axis for ROS generation during K-Ras-induced cellular transformation. *Cell Death Differ.* **21**, 1185–1197 (2014).
59. Maciag, A., Sithanandam, G. & Anderson, L. M. Mutant K-rasV12 increases COX-2, peroxides and DNA damage in lung cells. *Carcinogenesis* **25**, 2231–2237 (2004).
60. Liou, G. Y. *et al.* Mutant KRas-Induced Mitochondrial Oxidative Stress in Acinar Cells Upregulates EGFR Signaling to Drive Formation of Pancreatic Precancerous Lesions. *Cell Rep.* **14**, 2325–2336 (2016).
61. Kopnin, P. B., Agapova, L. S., Kopnin, B. P. & Chumakov, P. M. Repression of sestrin family genes contributes to oncogenic Ras-induced reactive oxygen species up-

- regulation and genetic instability. *Cancer Res.* **67**, 4671–4678 (2007).
62. Janes, M. R. *et al.* Targeting KRAS Mutant Cancers with a Covalent G12C-Specific Inhibitor. *Cell* (2018) doi:10.1016/j.cell.2018.01.006.
 63. Debacq-Chainiaux, F., Erusalimsky, J. D., Campisi, J. & Toussaint, O. Protocols to detect senescence-associated beta-galactosidase (SA- β gal) activity, a biomarker of senescent cells in culture and in vivo. *Nat. Protoc.* **4**, 1798–1806 (2009).

REVIEWER COMMENTS

Reviewer #2 (Remarks to the Author):

The revised manuscript addressed all my previous concerns.

Reviewer #3 (Remarks to the Author):

The authors have addressed all my comments.

Reviewer #4 (Remarks to the Author):

The authors have addressed all my concerns satisfactorily. Congratulations to the authors for the interesting study!

Reviewer #5 (Remarks to the Author):

The authors clearly show that mutant Kras in lung cancer is enriched in triacylglycerides and phosphatidylcholine because fatty acid synthase (FASN) is activated in Kras mutant lung cancer (KMLC). FASN inhibition results in ferroptosis. High FASN and Lands cycle activities in KMLC create a dependence on newly synthesized fatty acids to repair lipid peroxidation of polyunsaturated fatty acids (PUFA) in the high oxidative stress environment of KMLC to prevent KMLC cells from undergoing ferroptosis. This is an interesting paper that shows these new mechanistic links for the first time by providing solid evidence at the cellular, animal model, and patient levels.

The authors answered the questions and comment from the initial four reviewers well and significantly improved their manuscript. Since this is my first assessment of this manuscript, I do have additional comments that were not raised in the initial review but are very important to address to make this manuscript technically sound.

Comment 1: It is not clear why PEN (short for polyethylene naphthalate) membrane slides were used for MALDI imaging. PEN membrane slides are typically used for laser capture microdissection, but not MALDI imaging. The Waters Waters Synapt G2 Si typically requires glass slides for MALDI imaging. The use of substrates deposited on glass slides on which tissue sections are placed is important as they could potentially lead to delocalization of analytes as for example describe the following paper.

<https://pubs.acs.org/doi/10.1021/acs.analchem.9b05665>

Comment 2: Please provide in the MALDI imaging methods section what lipid classes were detected with what matrix and in what ion mode so that other scientists can attempt to replicate your results.

More details are needed in the MALDI imaging section. Please describe how exactly you have acquired on-tissue MS/MS spectra and what were your selection windows (in Dalton) when you selected molecular ion peak for fragmentation.

Comment 3: How were triacylglycerides (TAGs) ionized by MALDI imaging in your manuscript? Please specifically give detailed matrix conditions and acquisition parameters for each lipid class detected by MALDI imaging either in the main MALDI imaging methods section or in the supplementary methods section if there is not enough space in main methods. Also, you may have missed out on several important TAGs in your study as TAGs do not ionize well in MALDI imaging with DHB matrix. Several papers have shown more effective methods to ionize TAGs including <https://pubs.acs.org/doi/10.1021/acs.analchem.6b01141>. It may be beyond the scope of this manuscript to rerun MALDI imaging experiments with TAG-optimized matrices. However, it would be helpful to just mention the possibility that additional TAGs which were not detected within the limitations of the current study, could be involved in the reported phenomenon.

Comment 4: The identification of lipid species is insufficient in the current manuscript. Supplementary Table 1 is a good start, but it is far from complete in its current state as it only contains a small fraction of the m/z's described and shown in the manuscript. Also, only very limited fragmentation data are listed for on-tissue MS/MS in Supplementary Table 1. No TAG fragmentation data are given at all. Please provide MS/MS fragmentation data for every single MALDI-imaged lipid species in this manuscript including for all figures and supplementary figures alike. Otherwise, your assignments/identifications provided for the MALDI images are meaningless. Please also provide all MS/MS fragmentation spectra in your supplementary information including interpretations of fragments.

Comment 5: It is standard in the mass spectrometry imaging community to report the mass error for each detected ion, which is a reflection of the mass accuracy at the detected mass. An example would be m/z 795.4 +/- 0.1 Da. You need to provide the mass error for every m/z ion shown in your manuscript for all figures in the main manuscript and supplementary figures and tables alike. Figure 8 in your rebuttal, i.e., the mslQuant display of the mass list view, shows what you need to provide for each reported m/z ion in your manuscript. Lastly, since you are using a time-of-flight instrument with limited mass accuracy, it would be advisable to supplement your MALDI imaging data with a few limited MALDI imaging runs on an instrument with higher mass resolution if you have access to one. At the minimum, please comprehensively provide MS/MS data for each m/z shown and reported from your MALDI imaging data as mentioned above.

Comment 6: Supplementary Figure 10e does not show the histology of tumors corresponding to MALDI imaging data. Please add the histology to these figures as well, like all other figures.

Comment 7: You did a good job on your relative quantitation of MALDI imaging data in response to the previous reviewer #1. Well done.

We acknowledge that reviewer #5 made several constructive comments and asked for important clarifications: we are grateful for their contribution.

For sake of clarity, throughout the text of this rebuttal, we will indicate the Figures referring to the previous rebuttal in **bold blue**, Figures referring to this rebuttal in **bold orange** and Figures/tables in the actual manuscript in **bold black**. Text edits within the main manuscript and supplementary information are highlighted in red.

Point by point answer to the reviewer's comments:

Q1: “The authors answered the questions and comment from the initial four reviewers well and significantly improved their manuscript...I do have additional comments that were not raised in the initial review but are very important to address to make this manuscript technically sound”.

We thank the reviewer for acknowledging our efforts in answering to the previous reviewer. We agree with the reviewer that their additional comments were instrumental to improve our manuscript, especially from a technical point of view.

Q 2: ‘It is not clear why PEN (polyethylene naphthalate) membrane slides were used for MALDI imaging. PEN membrane slides are typically used for laser capture microdissection, but not MALDI imaging’.

A1: We acknowledge that due to a clerical error, we reported erroneously that we used PEN membrane slides. In fact, we used PEN-membranes (Leica, 2 µm) for Laser capture Microdissection (LMD) of tumor samples that underwent lipid extraction and Lipidomics Analysis by HPLC-MS/MS. Instead, we mounted contiguous tissue sections on Superfrost Microscope Slides (Fisherbrand) and stored at -80°C until being analyzed by MALDI-IMS.

We edited the manuscript accordingly in the method section of the revised version of our manuscript (page 26, Line 5).

Q3: “Please provide in the MALDI imaging methods section what lipid classes were detected with what matrix and in what ion mode so that other scientists can attempt to replicate your results. More details are needed in the MALDI imaging section. Please describe how exactly you have acquired on-tissue MS/MS spectra and what were your selection windows (in Dalton) when you selected molecular ion peak for fragmentation.

A3: We acknowledge that we mistakenly referred to the MALDI imaging technique used in the section method as “MALDI-MS/MS”. Instead, MALDI imaging data were generated via MALDI-MS. We edited the method section of the revised version of our manuscript accordingly (pages 26 and 27).

We decided to use of MALDI-MS imaging as this technique has been extensively employed to map the distribution of lipids in a wide range of organs¹. As shown by *Benabdellah et al.* and many other groups, MALDI-MS imaging is a robust and reproducible technique, provided that care is taken during sample preparation and matrix application²⁻¹³. We are well-aware that without the employment of MS/MS, it is not possible to discriminate chemical variants of lipids with identical numbers of acyl carbons and double bonds, that is, with identical masses. Therefore, this study could not specify the identity of the alkyl chains and the position of their double bonds. However, we would like to stress that in our manuscript MALDI-MS imaging served the role of confirming lipidomic data obtained, from the same samples, by HPLC-MS/MS and to provide spatial distribution.

Recognizant of such limitations, we now use the terms “tentatively identified as/assigned to” whenever referring to species identified with MALDI-MS data.

As for the detailed methods, a HTX Sprayer M5 (HTX Technologies, LLC., Chapel Hill, NC, USA), an automated sprayer matrix applicator, was used to apply 2,5-Dihydroxybenzoic acid (DHB) for positive mode, at a flow rate of 100 µL/min with temperature set to 75/30 °C for sprayer/tray. DHB was dissolved to a concentration of 15 mg/mL in 50% acetonitrile with 0.1% TFA. Before being loaded into the mass spectrometer, the slides were placed in a MALDI plate, scanned using an EPSON scanner

(Epson, Suwa, Japan), and the sections of tissue were mapped into High Definition Imaging software (HDI 1.4; Waters, Milford, MA, USA). DHB is considered the standard matrix for lipid studies¹⁴. MALDI-MS imaging was performed using a Water Synapt G2 Si (Waters Corporation, Milford, MA). Data were acquired with a spot of 60 μm with 300 laser shots at 1 kHz using a pulse energy with an average of 25 μJ . The laser intensity was adjusted to 60%. The mass range was 50-2000 m/z (as it is typically done for lipids) and the instrument was calibrated using peak signals from red phosphorus. We performed all data image visualization and data analysis using msIQuant¹⁵. Within msIQuant, Peak option signal to noise ratio (SNR) was set at 3.0 and peak Group Detection within 0.5 Da as for marker selection, Marker Mass Range was ± 0.05 Da and the Maximum Intensity in range was used as intensity Method. We converted all files into imzML using the Waters High Definition Imaging (HDI) software. Lipid identification was performed comparing observed peaks (experimental mass) with the theoretical values reported in the Lipid MAPS (<http://www.lipidmaps.org/>)¹⁶ and Madison Metabolomics Consortium (<http://mmcd.nmrfam.wisc.edu/>)¹⁷ databases (theoretical mass), using the mass accuracy (0.5 Da) as a tolerance window. In this way, we identified a single candidate for each peak in the spectrum in most cases. Using experimental and theoretical values, mass error was calculated $\frac{(\text{observed } m/z - \text{theoretical } m/z)}{\text{observed } m/z} \times 10^6$ as explained also in A4. We now provide this information in the Method Section of the manuscript (pages 26-27) and in the relative figure legends.

We reported the tentative molecular assignments of the peaks we detected in **Table 1**: “Tentative Assignment of Ions” (and now **Supplementary Table S1** of the manuscript). In most cases, signals were isotopically resolved, i.e. they were accompanied by peaks of lower intensity at higher masses. m/z values given in this study refer to the monoisotopic molecular weight, since the MS analyses were recorded with about unit mass resolution over the complete mass range. Most spectra exhibit, apart from the protonated molecular ion (M+1), also the corresponding sodium (M+23) and potassium adducts (M+39).

Based on their monoisotopic m/z value, the major ion signals in the spectra (**Figure 1.A** and **Figure 2.A** of answer to reviewer #1) can be

Figure 1. Tissue distribution of ions tentatively assigned as fragments derived from [PC+H]⁺. Mass Peak and tissue distribution of peaks tentatively assigned to phosphocholine (A) and choline (B) fragments detected at m/z 478, m/z 504. Mass error was 0.01 ppm. Scale bar: 2mm

correctly assigned to individual lipid species with a certain carbon number in their fatty acid chain and number of double bonds. The glycerolipid species numbers (x:y) denote the total length and number of double bonds of acyl chains, while the sphingolipid species numbers correspond to the length and number of double bonds of the acyl chain added to those of the attached sphing-4-enine (d18:1) or sphinganine (d18:0) base.

Together with the molecular ions, we observed several non-matrix peaks (previously reported in **Supplementary Table S1** of the previous version of the manuscript). This observation suggests that in-source prompt fragmentation may have occurred. As elsewhere reported, fragments produced in the desorption/ionization process are generally similar to those produced in the course of ESI/MS¹⁸⁻²⁰. For instance, it is well known that phosphatidylcholine (PC), LysoPC and sphingomyelin (SM) are characterized by the loss of the

TH7037 L140 HCC-4059 HCC-4190 CP58391

A [LysoPC (16:0)+H-H₂O]⁺

B [LysoPC (18:1)+ H-H₂O]⁺

Figure 2. Tissue distribution of ions tentatively assigned as fragments derived from LysoPC 16:0 and 18:1. Mass Peak and tissue distribution of peaks

and LysoPC 18:1; respectively (**Figure 2.A and B**).

Similarly, signals observed at m/z 459 and m/z 475 are consistent with neutral loss (NL) of trimethylamine (M-59) from the sodiated and potassiated adducts of LysoPC 16:0; respectively (**Figure 3.A and 3.B**).

Figure 3. Colocalization of ions assigned as LysoPC 16:0 and PC 32:1 and their minor fragments. Tissue distribution of peaks tentatively identified as sodiated (**A**) and potassiated (**B**) LysoPC 16:0. Tissue distribution of peak assigned to sodiated PC 32:1 (**C**). Parent molecule is in red, the minor fragment in green, their co-localization is in yellow. Mass error is expressed as ppm. Scale bar: 2mm. [LPC (16:0)+Na]⁺ parent m/z 518.33, mass error (7.7), fragment m/z 429.24, mass error (-17.4). [LPC (16:0)+K]⁺ parent m/z 534.33, mass error (-7.3), fragment m/z 474.22, mass error (-5.1). [PC (32:1)+Na]⁺ parent m/z 754.53, mass error (2.5), fragment m/z 695.47, mass error (11.1).

Intense ion peaks were observed at m/z 669 and m/z 523. We assigned them to a NL of 59 Da and 205 Da from the highly abundant sodiated PC 30:0.

Similarly, we tentatively assigned signals at m/z 697, m/z 573, m/z 551 as generated from NL of 59, 183 and 205 Da from sodiated PC 32:1; signals at m/z 695, m/z 571, m/z 549 from NL of 59, 183 and

quaternary ammonium group under conditions of MALDI-MS^{18–20}. Thus, we surmise that the relatively small signals at m/z 184 and m/z 104 match the molecular weight of fragments yielded by successive breakdown of the [M+H]⁺ ions of both PC and SM²⁰.

More specifically, fragments at m/z 184 and m/z 104 indicate most likely the formation of the phosphocholine and the choline fragment, respectively (**Figure 1. A and B**).

We speculate that these fragments reached the detector later as compared to those fragment ions of the same m/z but generated upon laser desorption/ionization. Furthermore, the choline header group carries a stable charge after fragmentation, so fragments of PC yield detectable peaks in the MS spectra.

Also, the formation of ion m/z 478 and m/z 504 likely corresponds to the loss of water of protonated LysoPC 16:0

205 Da from sodiated PC 32:1; while signals at m/z 642 and m/z 542 from NL of 59 and 183 Da from sodiated SM 34:1.

We recognize that peaks at m/z 697.47 and 695.45 could as well be assigned to sodiated phosphatidic acid (PA) 34:1 and PA 32:1; respectively. However, we considered this possibility unlikely since: 1. these peaks are quite intense, and PA makes up only around 1% of total cellular lipid content²¹; 2. the “fragment” peaks colocalize with the “parent molecules” (Figure 3.C).

As we recognize that these are not diagnostic fragments generated by MS/MS fragmentation, Table1 “Tentative assignment of observed ions” is now Supplementary Table S1 of the manuscript. In this table we reported all the ion adducts, theoretical and experimental mass values, and error mass for all the peaks observed and references from other published literature.

Q4: How were triacylglycerides (TAGs) ionized by MALDI imaging in your manuscript? Please specifically give detailed matrix conditions and acquisition parameters for each lipid class detected by MALDI imaging either in the main MALDI imaging methods section or in the supplementary methods section if there is not enough space in main methods. Also, you may have missed out on several important TAGs in your study as TAGs do not ionize well in MALDI imaging with DHB matrix. However, it would be helpful to just mention the possibility that additional TAGs which were not detected within the limitations of the current study, could be involved in the reported phenomenon.

A4: We agree with the reviewer in that the mass spectra were dominated by polar lipids, *i.e.* PC, SM, and phosphatidylethanolamine (PE) (Figure 1.B and 2.B of answer to reviewer #1). However, we were able to detect signals that we tentatively identified as TAGs between m/z 816 and m/z 947 in the mass spectra (Table 1 in red).

Using the experimental conditions described in A3 and consistently with the MALDI-MS literature, TAGs were mainly found as ammonium, sodiated or potassiated (these latter ones at minor intensities) adducts, while proton adducts were absent^{22,23}. Importantly the ion adducts co-localized. For instance, Figure 4 shows the spatial distribution of sodiated and potassiated TAG 56:5.

Of note, our findings are consistent with other manuscripts where TAGs were detected (mainly as sodium adducts) using 2,5-dihydroxybenzoic acid (DHB) as matrix^{14,24,25}. Importantly, DHB, when compared to other matrices, for instance α -cyano-hydroxy cinnamic acid or 1,8-dihydroxy-9,10-dihydroanthracen-9-one (dithranol), had the highest reproducibility and the sensitivity for TAGs analysis²⁵.

We are well-aware that MALDI-MS has some limitations in TAG analysis because of their in-source prompt fragmentation. Also, we recognize that we could have missed several TAGs species which do not ionize well with DHB matrix. However, since in this manuscript MS/MS lipidomics prompted us to focus on PC and LysoPC, 1) we excluded that TAG played a relevant role in the phenomenon and 2) we didn't deem necessary to further test which matrix works the best for TAG ionization/detection in our experimental conditions. However, for the sake of clarity, we include a brief statement about these limitations in our manuscript (page 27, lines 16-20)

Figure 4. Colocalization of TAG 56:5 adducts. (A) Tissue distribution of peaks tentatively identified as sodiated and potassiated TAG 56:5. The sodiated adduct is in red (m/z 931.77, mass error (ppm) -5.9), the potassiated adduct in green (m/z 947.80, mass error (ppm) -4.6), while their co-localization is in yellow. Mass error is expressed as ppm. Scale bar: 2mm.

Q5: The identification of lipid species is insufficient in the current manuscript. Supplementary Table 1 is a good start, but it is far from complete in its current state as it only contains a small fraction of the m/z's described and shown in the manuscript. Also, only very limited fragmentation data are listed for on-tissue MS/MS in Supplementary Table 1. No TAG fragmentation data are given at all. Please provide MS/MS fragmentation data for every single MALDI-imaged lipid species in this manuscript including for all figures and supplementary figures alike.

A5: As stated in A1, we used MALDI MS, not MALDI-MS/MS imaging. Thus, we did not obtain fragmentation data, other than the minor fragments derived from in-source decay as explained in A2. We provided in **Table 1 (Supplementary Table S1)** of the manuscript) all the ion adducts found together with the masses used for tentative peak assignment. The species reported in **fig 1d and 1e** of the manuscript are highlighted in grey. Again, we recognize that MALDI-MS provides a tentative species identification; and, accordingly, we employed this technique to complement the HPLC-MS/MS lipidomic data. Note that MALDI-MS imaging and HPLC-MS/MS lipidomic analysis were performed on the same samples. We edited our manuscript accordingly (page 27, lines 16-20).

Q6: You need to provide the mass error for every m/z ion shown in your manuscript for all figures in the main manuscript and supplementary figures and tables alike.

A6. We now edited the manuscript to add the mass error (ppm) for every m/z ion calculated as $\frac{(\text{observed } m/z - \text{theoretical } m/z)}{\text{observed } m/z} \times 10^6$ (**Fig 1; Supplementary Fig. 1 and 10; Supplementary Table S1**).

Q7: Supplementary Figure 10e does not show the histology of tumors corresponding to MALDI imaging data. Please add the histology to these figures as well, like all other figures.

A7: As requested, we added H&E staining images of mouse xenografts specimens corresponding to MALDI images.

Q8: You did a good job on your relative quantitation of MALDI imaging data in response to the previous reviewer #1. Well done.

A8 We thank the reviewer for recognizing our efforts.

Table 1: Tentative Assignment of Ions

Experimental mass	Theoretical mass	Mass error (ppm)	Tentative Assignment			Associated Literature
			Class	Alkyl composition	ionization product	
369.35	369.35	-2.4	Cholesterol		[M+H-H ₂ O] ⁺	26
490.28	490.29	-17.5	LysoPC	{14:0}	[M+Na] ⁺	27
494.32	494.32	-6.7	LysoPC	{16:1}	[M+H] ⁺	27
506.27	506.26	19.6	LysoPC	{14:0}	[M+K] ⁺	27
496.34	496.33	22.6	LysoPC	{16:0}	[M+H] ⁺	27
516.31	516.31	-1.4	LysoPC	{16:1}	[M+Na] ⁺	27
518.33	518.32	7.7	LysoPC	{16:0}	[M+Na] ⁺	3,27
520.34	520.34	-6.0	LysoPC	{18:2}	[M+H] ⁺	3,27
522.36	522.36	3.3	LysoPC	{18:1}	[M+H] ⁺	3,27
524.28	524.27	7.8	LysoPE	{20:4}	[M+Na] ⁺	3,27
524.36	524.37	-14.6	LysoPC	{18:0}	[M+H] ⁺	3,27
526.29	526.29	-8.9	LysoPE	{22:6}	[M+H] ⁺	27
532.29	532.28	12.8	LysoPC	{16:1}	[M+K] ⁺	27
534.29	534.30	-7.3	LysoPC	{16:0}	[M+K] ⁺	27
535.31	535.30	10.6	LysoPG	{18:0}	[M+Na] ⁺	27
540.25	540.27	-31.1	LysoPE	{20:4}	[M+K] ⁺	27
542.29	542.32	-67.7	LysoPC	{18:2}	[M+Na] ⁺	27
544.34	544.34	0.6	LysoPC	{18:1}	[M+Na] ⁺	3,27
544.34	544.34	0.9	LysoPC	{20:4}	[M+H] ⁺	3,27
546.35	546.35	-7.9	LysoPC	{18:0}	[M+Na] ⁺	27
548.28	548.27	0.9	LysoPE	{22:6}	[M+K] ⁺	27
558.30	558.30	5.4	LysoPC	{18:2}	[M+Na] ⁺	27
560.31	560.31	2.3	LysoPC	{18:1}	[M+K] ⁺	27
562.33	562.33	-0.7	LysoPC	{18:0}	[M+K] ⁺	27
564.27	564.25	32.4	LysoPE	{22:6}	[M+K] ⁺	27
566.33	566.32	10.1	LysoPC	{20:4}	[M+Na] ⁺	27
568.27	568.28	-14.6	LysoPE	{22:4}	[M+K] ⁺	27
568.33	568.34	-10.9	LysoPC	{22:6}	[M+H] ⁺	27
577.52	577.48	56.8	DAG	{33:3}	[M+H] ⁺	
582.30	582.30	6.9	LysoPC	{20:4}	[M+K] ⁺	27
590.32	590.32	3.4	LysoPC	{22:6}	[M+Na] ⁺	27
603.54	603.50	67.1	DAG	{33:1}	[M+Na] ⁺	

606.29	606.30	-3.8	LysoPC	{22:6}	[M+K] ⁺	27
607.57	607.58	-18.3	CE	{16:0}	[M+H-H ₂ O] ⁺	
616.51	616.49	19.3	DAG	{35:6}	[M+NH ₄] ⁺	
617.51	617.51	-3.9	DAG	{36:4}	[M+H] ⁺	
621.47	621.48	-22.2	DAG	{33:0}	[M+K] ⁺	
626.59	626.58	10.1	DAG	{35:1}	[M+NH ₄] ⁺	
645.54	645.54	-5.6	DAG	{38:4}	[M+H] ⁺	
650.51	650.51	-0.6	PE-O-	{30:0}	[M+H] ⁺	
650.59	650.57	22.1	DAG	{37:3}	[M+NH ₄] ⁺	
650.64	650.64	-9.2	Cer	{42:1}	[M+H] ⁺	27
651.53	651.53	-2.3	DAG	{40:6}	[M+H-H ₂ O] ⁺	
652.60	652.60	-8.1	CE	{17:1}	[M+NH ₄] ⁺	
652.60	652.59	15.3	DAG	{37:2}	[M+NH ₄] ⁺	
653.61	653.61	-1.7	DAG	{38:0}	[M+H] ⁺	
654.61	654.62	-7.8	CE	{17:1}	[M+NH ₄] ⁺	
654.61	654.60	15.4	DAG	{37:1}	[M+NH ₄] ⁺	
655.56	655.57	-4.6	DAG	{40:4}	[M+H-H ₂ O] ⁺	
656.62	656.62	5.3	Cer	{40:0(2OH)}	[M+H] ⁺	
656.62	656.63	-18.1	CE	{17:0}	[M+NH ₄] ⁺	
657.64	657.70	-92.6	DAG	{40:2}	[M+H-H ₂ O] ⁺	
659.60	659.58	20.6	DAG	{40:3}	[M+H-H ₂ O] ⁺	
659.60	659.57	32.6	CE	{17:1}	[M+Na] ⁺	
661.61	661.62	-3.6	DAG	{40:1}	[M+H-H ₂ O] ⁺	
664.46	664.42	62.2	PS	{27:1}	[M+H] ⁺	
669.44	669.45	-6.3	PA	{32:1}	[M+Na] ⁺	27
667.56	667.57	-18.9	DAG	{41:5}	[M+H-H ₂ O] ⁺	
670.60	670.65	-66.4	CE	{18:0}	[M+NH ₄] ⁺	
670.60	670.61	-10.1	Cer	d{42:2}	[M+NH ₄] ⁺	
675.53	675.53	-0.3	DAG	{42:8}	[M+H-H ₂ O] ⁺	
675.53	675.54	-13.3	SM	d{32:1}	[M+H] ⁺	
677.54	677.55	-11.7	DAG	{42:7}	[M+H-H ₂ O] ⁺	
678.48	678.51	-45.2	PC	{28:0}	[M+H] ⁺	27
682.46	682.45	19.5	PE	{32:5}	[M+H] ⁺	
685.48	685.42	81.8	PA	{32:1}	[M+K] ⁺	27
685.57	685.61	-57.9	DAG	{42:3}	[M+H-H ₂ O] ⁺	
688.59	688.6	-9.7	Cer	{42:1}	[M+K] ⁺	
689.64	689.64	-6.4	DAG	{42:1}	[M+H-H ₂ O] ⁺	

690.46	690.51	-63.1	PE	{32:1}	[M+H] ⁺	27
692.43	692.45	-30.5	PS	{29:1}	[M+H] ⁺	
693.45	693.45	0.6	PA	{34:3}	[M+Na] ⁺	27
693.56	693.56	-1.4	CE	{20:5}	[M+Na] ⁺	
694.65	694.63	18.3	DAG	{40:2}	[M+NH ₄] ⁺	
695.57	695.60	-41.0	DAG	{43:5}	[M+H-H ₂ O] ⁺	
696.64	696.67	-33.7	CE	{20:1}	[M+NH ₄] ⁺	27
697.47	697.48	-12.2	PA	{34:1}	[M+Na] ⁺	27
700.46	700.49	-50.0	PC	{30:3}	[M+H] ⁺	
701.56	701.55	14.3	DAG	{44:9}	[M+H-H ₂ O] ⁺	
701.56	701.56	1.6	SM	d{34:2}	[M+H] ⁺	27
702.58	702.60	-32.9	DAG	{41:5}	[M+NH ₄] ⁺	27
703.57	703.57	-10.1	SM	d{34:1}	[M+H] ⁺	27
705.58	705.58	3.8	DAG	{44:7}	[M+H-H ₂ O] ⁺	
706.54	706.54	-0.3	PC	{30:0}	[M+H] ⁺	27
709.42	709.42	0.8	PA	{34:3}	[M+K] ⁺	27
711.44	711.44	4.4	PA	{34:2}	[M+K] ⁺	27
712.44	712.43	16.0	PC	{28:2}	[M+K] ⁺	27
713.45	713.45	-2.2	PA	{34:1}	[M+K] ⁺	27
716.43	716.46	-45.1	PC	{28:0}	[M+K] ⁺	27
716.52	716.52	-2.8	PE	{34:2}	[M+H] ⁺	27
718.57	718.54	38.0	PE	{34:1}	[M+H] ⁺	3,27
718.57	718.57	-12.9	PC-P-	{32:0}	[M+H] ⁺	27
718.57	718.57	-12.7	PC-O-	{32:1}	[M+H] ⁺	27
719.57	719.57	-1.1	CE	{22:6}	[M+Na] ⁺	
720.57	720.55	20.1	PE	{34:0}	[M+H] ⁺	27
720.57	720.59	-30.4	PC-O-	{32:0}	[M+H] ⁺	27
720.65	720.61	55.5	DAG	{42:3}	M+Na] ⁺	27
721.48	721.48	5.4	PA	{36:3}	[M+Na] ⁺	27
721.59	721.59	8.7	SM	d{34:0 (OH)}	[M+H] ⁺	27
721.59	721.59	3.2	CE	{22:5}	[M+Na] ⁺	
723.49	723.49	-8.3	PA	{36:2}	[M+Na] ⁺	27
725.55	725.56	-8.1	SM	d{34:1}	[M+Na] ⁺	27
728.52	728.52	-2.5	PC	{30:0}	[M+Na] ⁺	
730.54	730.54	-4.2	PC	{32:2}	[M+H] ⁺	
731.60	731.61	-3.1	SM	d{36:1}	[M+H] ⁺	3,26,28,29

732.55	732.55	-2.9	PC	{32:1}	[M+H] ⁺	3,27
734.57	734.57	-1.4	PC	{32:0}	[M+H] ⁺	3,26–29
737.45	737.45	-5.4	PA	{36:3}	[M+K] ⁺	27
738.53	738.52	15.2	PE	{34:2}	[M+Na] ⁺	27
739.47	739.47	6.2	PA	{36:2}	[M+K] ⁺	27
740.55	740.56	-6.8	PC-O-	{32:1}	[M+Na] ⁺	27
740.55	740.56	-6.8	PC-P-	{32:0}	[M+Na] ⁺	27
740.55	740.56	-10.0	PC-O-	{34:4}	[M+H] ⁺	27
741.54	741.53	15.4	SM	d{34:1}	[M+K] ⁺	27
744.49	744.49	-2.0	PC	{30:0}	[M+K] ⁺	27
744.61	744.59	24.2	PC-O-	{34:2}	[M+H] ⁺	27
746.61	746.57	52.5	PE	{36:1}	[M+H] ⁺	3,27
746.61	746.61	3.8	PC-O-	{34:1}	[M+H] ⁺	27
746.61	746.61	3.8	PC-P-	{34:0}	[M+H] ⁺	27
748.50	748.51	-11.2	PS	{33:1}	[M+H] ⁺	27
748.60	748.59	14.8	PE	{36:0}	[M+H] ⁺	27
748.60	748.62	-33.8	PC-O-	{34:0}	[M+H] ⁺	27
749.51	749.51	-2.7	PA	{38:3}	[M+Na] ⁺	27
750.50	750.52	-40.6	PS	{33:0}	[M+H] ⁺	
752.54	752.52	21.7	PC	{32:2}	[M+Na] ⁺	27
753.59	753.59	-1.6	SM	d{36:1}	[M+Na] ⁺	3,27,28
754.54	754.54	2.5	PC	{32:1}	[M+Na] ⁺	3,27
754.54	754.54	-0.7	PC	{34:4}	[M+H] ⁺	3,27
756.54	756.55	-10.7	PC	{32:0}	[M+Na] ⁺	3,26–29
758.57	758.57	5.7	PC	{34:2}	[M+H] ⁺	3,26,29
760.58	760.59	-3.7	PC	{34:1}	[M+H] ⁺	3,26,27
761.45	761.45	2.5	PA	{38:5}	[M+K] ⁺	27
762.50	762.51	-4.9	PE	{38:7}	[M+H] ⁺	
762.50	762.50	-1.7	PE	{36:4}	[M+Na] ⁺	27
762.50	762.48	25.8	PE-P-	{36:4}	[M+K] ⁺	27
762.50	762.48	25.8	PE-O-	{36:5}	[M+K] ⁺	27
763.48	763.47	13.9	PA	{38:4}	[M+K] ⁺	27
764.53	764.52	4.2	PE	{38:6}	[M+H] ⁺	27
765.47	765.48	-12.7	PA	{38:3}	[M+K] ⁺	27
766.55	766.54	21.7	PE	{38:5}	[M+H] ⁺	27

767.50	767.48	20.2	PA	{38:2}	[M+K] ⁺	27
768.49	768.49	0.8	PC	{32:2}	[M+K] ⁺	27
768.49	768.48	17.7	PS	{35:5}	[M+H] ⁺	
768.58	768.59	-5.3	PC-P-	{34:0}	[M+Na] ⁺	27
768.58	768.59	-5.3	PC-O-	{34:1}	[M+Na] ⁺	27
768.58	768.59	-8.5	PC-O-	{36:4}	[M+H] ⁺	27
770.50	770.51	-6.7	PC	{32:1}	[M+K] ⁺	27
772.52	772.53	-6.0	PC	{32:0}	[M+K] ⁺	27
774.61	774.60	8.0	PE	{38:1}	[M+H] ⁺	27
775.61	775.60	16.4	PC	{34:2}	[M+NH ₄] ⁺	
776.60	776.62	-24.1	PE	{38:0}	[M+H] ⁺	
778.46	778.48	-17.9	PE	{36:4}	[M+K] ⁺	27
778.54	778.54	2.4	PC	{34:3}	[M+Na] ⁺	27
778.54	778.54	-0.6	PC	{36:6}	[M+H] ⁺	
778.61	778.60	19.5	PG-O	{36:2}	[M+NH ₄] ⁺	
778.61	778.60	19.5	PG-P	{36:1}	[M+NH ₄] ⁺	
778.61	778.62	-7.6	GalCer	{d38:1}	[M+Na] ⁺	27
780.55	780.55	0.1	PC	{36:5}	[M+H] ⁺	
780.55	780.55	3.2	PC	{34:2}	[M+Na] ⁺	27
782.57	782.57	0.9	PC	{36:4}	[M+H] ⁺	3
782.57	782.61	-45.6	PC-O-	{37:4}	[M+H] ⁺	
782.57	782.57	0.1	PC	{34:1}	[M+Na] ⁺	3,27,28
784.58	784.57	20.6	PC	{36:3}	[M+H] ⁺	3
786.51	786.50	11.7	PE	{38:6}	[M+Na] ⁺	27
786.61	786.60	6.5	PC	{36:2}	[M+H] ⁺	3,27
787.48	787.49	-14.5	PG	{34:1}	[M+K] ⁺	27
788.52	788.52	0.3	PE	{38:5}	[M+Na] ⁺	27
788.62	788.62	3.0	PC	{36:1}	[M+H] ⁺	3,26,27,29
789.50	789.48	17.1	PA	{40:5}	[M+K] ⁺	27
790.55	790.56	-16.8	PS	{36:1}	[M+H] ⁺	
790.55	790.54	13.0	PE	{38:4}	[M+Na] ⁺	27
792.58	792.57	0.1	PS	{36:0}	[M+H] ⁺	
794.51	794.51	6.5	PC	{34:3}	[M+K] ⁺	27
794.59	794.57	27.4	PE	{40:5}	[M+H] ⁺	27
794.59	794.61	-18.4	PC-P-	{38:4}	[M+H] ⁺	27
794.59	794.61	-18.4	PC-O-	{38:5}	[M+H] ⁺	27

796.53	796.53	3.5	PC	{34:2}	[M+K] ⁺	3,27
796.59	796.59	2.4	PE	{40:4}	[M+H] ⁺	3,27
798.54	798.54	1.0	PC	{34:1}	[M+K] ⁺	3,27
799.66	799.67	-5.0	SM	d{41:1}	[M+H] ⁺	
799.66	799.68	-17.4	TAG	{46:1}	[M+Na] ⁺	
800.61	800.62	-4.6	PE	{40:2}	[M+H] ⁺	
801.68	801.69	-12.1	TAG	{46:0}	[M+Na] ⁺	
801.68	801.68	0.2	SM	d{41:0}	[M+H] ⁺	
802.47	802.48	-14.5	PE	{38:6}	[M+K] ⁺	27
802.54	802.54	8.5	PC	{36:5}	[M+Na] ⁺	27
802.54	802.54	5.0	PC	{38:8}	[M+H] ⁺	
804.56	804.55	6.5	PC	{36:4}	[M+Na] ⁺	3,27
804.56	804.55	3.5	PC	{38:7}	[M+H] ⁺	
806.56	806.57	-2.9	PC	{36:3}	[M+Na] ⁺	3
806.56	806.57	-5.8	PC	{38:6}	[M+H] ⁺	3,27–29
808.58	808.59	-4.0	PC	{38:5}	[M+H] ⁺	3,27
808.58	808.58	-1.0	PC	{36:2}	[M+Na] ⁺	3
810.60	810.60	-5.8	PC	{38:4}	[M+H] ⁺	3,27–29
810.60	810.60	-2.8	PC	{36:1}	[M+Na] ⁺	3
811.67	811.67	-0.7	SM	d{42:3}	[M+H] ⁺	27
812.53	812.54	-22.5	PS	{38:4}	[M+H] ⁺	
812.53	812.52	3.4	PE	{42:10}	[M+H] ⁺	
812.60	812.62	-15.3	PC	{38:3}	[M+H] ⁺	3
812.67	812.68	-3.2	TAG	{48:6}	[M+NH ₄] ⁺	
813.68	813.68	-2.6	SM	d{42:2}	[M+H] ⁺	27
813.68	813.69	-12.7	GlcCer	d{31:2}	[M+Na] ⁺	
814.54	814.56	-24.4	PS	{38:3}	[M+H] ⁺	3
815.70	815.70	-0.7	SM	d{42:1}	[M+H] ⁺	3,27
816.58	816.59	-4.9	PC-P-	{38:4}	[M+Na] ⁺	27
816.57	816.59	-17.6	PC-O-	{38:5}	[M+Na] ⁺	27
816.57	816.57	4.2	PS	{39:8}	[M+H] ⁺	
816.70	816.71	-13.2	TAG	{48:4}	[M+NH ₄] ⁺	
818.50	818.50	5.9	PC	{36:5}	[M+K] ⁺	27
818.50	818.50	9.9	PS	{38:1}	[M+H] ⁺	
818.58	818.57	21.3	PE	{42:7}	[M+H] ⁺	
818.73	818.72	3.7	TAG	{48:3}	[M+NH ₄] ⁺	

819.66	819.65	11.8	TAG	{48:5}	[M+Na] ⁺	
820.53	820.53	4.4	PC	{36:4}	[M+K] ⁺	27
821.65	821.66	-13.3	TAG	{48:4}	[M+Na] ⁺	
823.67	823.68	-10.1	TAG	{48:3}	[M+Na] ⁺	
824.56	824.56	-0.4	PC	{36:2}	[M+K] ⁺	3,27
824.64	824.64	8.1	PG	{38:0}	[M+NH ₄] ⁺	
825.68	825.69	-12.5	TAG	{48:2}	[M+Na] ⁺	
826.57	826.57	-2.8	PC	{36:1}	[M+K] ⁺	27
827.71	827.70	13.4	SM	d{43:2}	[M+H] ⁺	
827.71	827.71	1.4	TAG	{48:1}	[M+Na] ⁺	
828.56	828.55	6.5	PC	{38:6}	[M+Na] ⁺	
829.72	829.72	5.5	SM	d{43:1}	[M+H] ⁺	
829.72	829.73	-6.4	TAG	{48:0}	[M+Na] ⁺	
830.57	830.57	-1.2	PC	{38:5}	[M+Na] ⁺	3
832.58	832.56	27.0	PC-P-	{38:4}	[M+K] ⁺	3,27
832.58	832.56	27.0	PC-O-	{38:5}	[M+K] ⁺	27
832.58	832.58	1.8	PC	{38:4}	[M+Na] ⁺	
833.64	833.65	-7.2	SM	d{42:3}	[M+Na] ⁺	27
833.64	833.66	-22.0	TAG	{49:5}	[M+Na] ⁺	
834.52	834.54	-23.0	PE	{40:4}	[M+K] ⁺	27
834.59	834.60	-8.7	PC	{40:6}	[M+H] ⁺	3,
834.59	834.60	-5.9	PC	{38:3}	[M+Na] ⁺	3,26,28,29
835.66	835.68	-18.8	TAG	{49:4}	[M+Na] ⁺	
836.54	836.54	-9.7	PS	{40:6}	[M+H] ⁺	
836.61	836.61	-2.9	PC	{38:2}	[M+Na] ⁺	3
836.61	836.62	-6.5	PC	{40:5}	[M+H] ⁺	3,27
837.68	837.69	-15.9	TAG	{49:3}	[M+Na] ⁺	
838.63	838.63	-8.1	PC	{40:4}	[M+H] ⁺	27
840.70	840.71	-7.3	TAG	{50:6}	[M+NH ₄] ⁺	
842.67	842.66	3.2	PC	{40:2}	[M+H] ⁺	
843.67	843.68	-14.2	TAG	{48:1}	[M+K] ⁺	27
844.52	844.53	-4.0	PC	{38:6}	[M+K] ⁺	27
845.67	845.70	-37.4	TAG	{48:0}	[M+K] ⁺	27
846.55	846.54	9.9	PC	{38:5}	[M+K] ⁺	
847.68	847.68	5.0	TAG	{50:5}	[M+Na] ⁺	
848.56	848.56	1.6	PC	{38:4}	[M+K] ⁺	27, 3

849.62	849.62	-2.9	SM	d{42:3}	[M+K] ⁺	27
849.69	849.69	-3.4	TAG	{50:4}	[M+Na] ⁺	
850.58	850.57	8.1	PC	{38:3}	[M+K] ⁺	27
851.64	851.64	3.5	SM	d{42:2}	[M+K] ⁺	27
851.64	851.65	-10.8	TAG	{49:4}	[M+K] ⁺	
851.71	851.72	-16.8	TAG	{50:3}	[M+Na] ⁺	27
853.66	853.66	10.1	SM	d{42:1}	[M+K] ⁺	27
853.66	853.67	-4.3	TAG	{49:3}	[M+K] ⁺	
853.72	853.73	-5.2	TAG	{50:2}	[M+Na] ⁺	27
855.74	855.74	1.3	TAG	{50:1}	[M+Na] ⁺	27
856.52	856.53	-2.6	PE	{42:7}	[M+K] ⁺	
856.52	856.53	-2.9	PS	{40:6}	[M+Na] ⁺	
856.58	856.58	-3.4	PC	{40:6}	[M+Na] ⁺	27
858.60	858.60	3.1	PC	{40:5}	[M+Na] ⁺	27
865.72	865.73	-3.7	TAG	{51:3}	[M+Na] ⁺	
867.69	867.68	3.6	TAG	{50:3}	[M+K] ⁺	27
869.70	869.70	2.6	TAG	{50:2}	[M+K] ⁺	27
871.70	871.72	-12.6	TAG	{50:1}	[M+K] ⁺	27
872.55	872.56	-2.8	PC	{40:6}	[M+K] ⁺	27
873.69	873.69	-0.6	TAG	{52:6}	[M+Na] ⁺	
874.57	874.57	-3.8	PC	{40:5}	[M+K] ⁺	27
875.71	875.72	-13.7	TAG	{52:5}	[M+Na] ⁺	
876.58	876.52	73.8	PS	{40:5}	[M+K] ⁺	27
877.72	877.73	-2.3	TAG	{52:4}	[M+Na] ⁺	
879.74	879.74	-3.1	TAG	{52:3}	[M+Na] ⁺	
881.75	881.76	-4.0	TAG	{52:2}	[M+Na] ⁺	27
893.70	893.70	5.4	TAG	{52:4}	[M+K] ⁺	27
895.55	895.53	20.7	PI	{35:4}	[M+Na] ⁺	
895.72	895.72	4.6	TAG	{52:3}	[M+K] ⁺	27
897.73	897.73	-3.3	TAG	{52:2}	[M+K] ⁺	27
899.72	899.75	-25.1	TAG	{52:1}	[M+K] ⁺	27
901.55	901.52	32.7	PI	{36:2}	[M+Na] ⁺	3
901.71	901.73	-13.4	TAG	{54:6}	[M+Na] ⁺	
903.75	903.74	6.7	TAG	{54:5}	[M+Na] ⁺	
905.76	905.76	5.9	TAG	{54:4}	[M+Na] ⁺	3
907.54	907.53	7.8	PI	{38:5}	[M+Na] ⁺	3

907.78	907.77	5.1	TAG	{54:3}	[M+Na] ⁺	27
909.55	909.55	0.1	PI	{38:4}	[M+Na] ⁺	27
912.58	912.57	2.8	PS	{44:7}	[M+Na] ⁺	
917.70	917.70	1.1	TAG	{54:6}	[M+K] ⁺	
919.72	919.72	7.2	TAG	{54:5}	[M+K] ⁺	27
921.49	921.49	2.2	PI	{38:6}	[M+K] ⁺	3,27
921.72	921.73	-7.4	TAG	{54:4}	[M+K] ⁺	27
923.51	923.50	1.4	PI	{38:5}	[M+K] ⁺	27
923.75	923.75	-1.3	TAG	{54:3}	[M+K] ⁺	27
925.52	925.52	0.5	PI	{38:4}	[M+K] ⁺	27
926.54	926.53	10.8	PS	{44:8}	[M+K] ⁺	27
927.74	927.78	-36.9	TAG	{54:1}	[M+K] ⁺	27
929.76	929.76	1.6	TAG	{56:6}	[M+Na] ⁺	3
931.53	931.53	3.5	PI	{40:7}	[M+Na] ⁺	
931.77	931.77	-5.9	TAG	{56:5}	[M+Na] ⁺	
933.78	933.79	-6.7	TAG	{56:4}	[M+Na] ⁺	
935.80	935.80	-0.7	TAG	{56:3}	[M+Na] ⁺	
945.72	945.73	-11.0	TAG	{56:6}	[M+K] ⁺	27
947.50	947.50	-2.6	PI	{40:7}	[M+K] ⁺	
947.76	947.80	-46.0	TAG	{56:5}	[M+K] ⁺	27
1468.13	1468.14	-3.0	PC	{32:0}	2 [M+H] ⁺	28
1492.13	1492.15	-10.7	PE	{36:1}	[2M+H] ⁺	
1520.16	1520.17	-4.1	PC	{34:1}	[2M+H] ⁺	28
1516.14	1516.14	-0.7	PC	{34:2}	[2M+H] ⁺	3
1544.17	1544.16	7.4	PE	{38:2}	[2M+H] ⁺	3
1564.14	1564.14	-0.5	PC	{36:4}	[2M+H] ⁺	3
1566.16	1566.15	1.9	PC	{34:1}+{38:6}	[M+H] ⁺	28
1568.16	1568.13	18.0	PC	{36:3}	[2M+H] ⁺	3
1572.22	1572.20	13.0	PC	{36:2}	[2M+H] ⁺	3,30
1592.17	1592.17	-0.4	PE	{40:4}	[2M+H] ⁺	3,28
1616.16	1616.17	-6.1	PC	{38:5}	[2M+H] ⁺	3
1620.22	1620.20	12.0	PC	{38:4}	[2M+H]	3

References

1. Benabdellah, F. *et al.* Mass spectrometry imaging of rat brain sections: Nanomolar sensitivity with MALDI versus nanometer resolution by TOF-SIMS. *Anal. Bioanal. Chem.* (2010) doi:10.1007/s00216-009-3031-2.
2. Peggi A., J., S., S., B. & R., C. Enhanced Sensitivity for High Spatial Resolution Lipid Analysis by Negative Ion Mode MALDI Imaging Mass Spectrometry. *Changes* (2012).
3. Astigarraga, E. *et al.* Profiling and imaging of lipids on brain and liver tissue by matrix-assisted laser desorption/ionization mass spectrometry using 2-mercaptobenzothiazole as a matrix. *Anal. Chem.* (2008) doi:10.1021/ac801662n.
4. Sugiura, Y. *et al.* Visualization of the cell-selective distribution of PUFA-containing phosphatidylcholines in mouse brain by imaging mass spectrometry. *J. Lipid Res.* (2009) doi:10.1194/jlr.M900047-JLR200.
5. Wang, H. Y. J., Post, S. N. J. J. & Woods, A. S. A minimalist approach to MALDI imaging of glycerophospholipids and sphingolipids in rat brain sections. *Int. J. Mass Spectrom.* (2008) doi:10.1016/j.ijms.2008.04.005.
6. Cerruti, C. D., Benabdellah, F., Laprévotte, O., Touboul, D. & Brunelle, A. MALDI imaging and structural analysis of rat brain lipid negative ions with 9-aminoacridine matrix. *Anal. Chem.* (2012) doi:10.1021/ac2025317.
7. Cerruti, C. D. *et al.* MALDI imaging mass spectrometry of lipids by adding lithium salts to the matrix solution. *Anal. Bioanal. Chem.* (2011) doi:10.1007/s00216-011-4814-9.
8. Chan, K. *et al.* MALDI mass spectrometry imaging of gangliosides in mouse brain using ionic liquid matrix. *Anal. Chim. Acta* (2009) doi:10.1016/j.aca.2009.02.051.
9. Chen, Y. *et al.* Imaging MALDI mass spectrometry using an oscillating capillary nebulizer matrix coating system and its application to analysis of lipids in brain from a mouse model of Tay-Sachs/Sandhoff disease. *Anal. Chem.* (2008) doi:10.1021/ac702350g.
10. Colsch, B. & Woods, A. S. Localization and imaging of sialylated glycosphingolipids in brain tissue sections by MALDI mass spectrometry. *Glycobiology* (2010) doi:10.1093/glycob/cwq031.
11. Goto-Inoue, N. *et al.* The detection of glycosphingolipids in brain tissue sections by imaging mass spectrometry using gold nanoparticles. *J. Am. Soc. Mass Spectrom.* (2010) doi:10.1016/j.jasms.2010.08.002.
12. Meriaux, C., Franck, J., Wisztorski, M., Salzet, M. & Fournier, I. Liquid ionic matrixes for MALDI mass spectrometry imaging of lipids. *J. Proteomics* (2010) doi:10.1016/j.jprot.2010.02.010.
13. Shrivastava, K. *et al.* Ionic matrix for enhanced MALDI imaging mass spectrometry for identification of phospholipids in mouse liver and cerebellum tissue sections. *Anal. Chem.* (2010) doi:10.1021/ac102422b.
14. Schiller, J. *et al.* Lipid analysis by matrix-assisted laser desorption and ionization mass spectrometry: A methodological approach. *Anal. Biochem.* (1999) doi:10.1006/abio.1998.3001.
15. Källback, P., Nilsson, A., Shariatgorji, M. & Andrén, P. E. MslQuant - Quantitation Software for Mass Spectrometry Imaging Enabling Fast Access, Visualization, and Analysis of Large Data Sets. *Anal. Chem.* (2016) doi:10.1021/acs.analchem.5b04603.
16. Fahy, E., Sud, M., Cotter, D. & Subramaniam, S. LIPID MAPS online tools for lipid research. *Nucleic Acids Res.* (2007) doi:10.1093/nar/gkm324.
17. Cui, Q. *et al.* Metabolite identification via the Madison Metabolomics Consortium Database [3]. *Nature Biotechnology* (2008) doi:10.1038/nbt0208-162.
18. Al-Saad, K. A., Siems, W. F., Hill, H. H., Zabrouskov, V. & Knowles, N. R. Structural analysis of phosphatidylcholines by post-source decay matrix-assisted laser desorption/ionization time-of-flight mass spectrometry. *J. Am. Soc. Mass Spectrom.* (2003) doi:10.1016/S1044-0305(03)00068-0.

19. Domingues, M. R. M. *et al.* Do charge-remote fragmentations occur under matrix-assisted laser desorption ionization post-source decompositions and matrix-assisted laser desorption ionization collisionally activated decompositions? *J. Am. Soc. Mass Spectrom.* (1999) doi:10.1016/S1044-0305(98)00144-5.
20. Wang, T., Hao, Y. & Chen, S. Uncover the Interference of Lipid Fragments on the Qualification and Quantification of Serum Metabolites in MALDI-TOF MS Analysis. *Rapid Commun. Mass Spectrom.* e9293 (2022) doi:10.1002/rcm.9293.
21. Vance, J. E. & Steenbergen, R. Metabolism and functions of phosphatidylserine. *Prog. Lipid Res.* **44**, 207–234 (2005).
22. Canuto, R. A., Muzio, G., Maggiora, M., Biocca, M. E. & Dianzani, M. U. Glutathione-S-transferase, alcohol dehydrogenase and aldehyde reductase activities during diethylnitrosamine-carcinogenesis in rat liver. *Cancer Lett.* (1993) doi:10.1016/0304-3835(93)90144-X.
23. Wang, J., Kalt, W. & Sporns, P. Comparison between HPLC and MALDI-TOF MS Analysis of Anthocyanins in Highbush Blueberries. *J. Agric. Food Chem.* **48**, 3330–3335 (2000).
24. Wiesman, Z. & Chapagain, B. P. Determination of fatty acid profiles and TAGs in vegetable oils by MALDI-TOF/MS fingerprinting. *Methods Mol. Biol.* **579**, 315–336 (2009).
25. Asbury, G. R., Al-Saad, K. A., Siems, W. F., Hannan, R. M. & Hill, H. H. Analysis of triacylglycerols and whole oils by matrix-assisted laser desorption/ionization time of flight mass spectrometry. *J. Am. Soc. Mass Spectrom.* **10**, 983–991 (1999).

REVIEWERS' COMMENTS

Reviewer #5 (Remarks to the Author):

The authors have sufficiently addressed all of my queries.